# Making Non-Stochastic Control (Almost) as Easy as Stochastic

**Max Simchowitz**
EECS Department
UC Berkeley
Berkeley, CA 94720
msimchow@berkeley.edu

## Abstract

Recent literature has made much progress in understanding *online LQR*: a modern learning-theoretic take on the classical control problem where a learner attempts to optimally control an unknown linear dynamical system with fully observed state, perturbed by i.i.d. Gaussian noise. The optimal regret over time horizon $T$ against the optimal control law scales as $\widetilde{\Theta}(\sqrt{T})$. In this paper, we show that the same regret rate (against a suitable benchmark) is attainable even in the considerably more general non-stochastic control model, where the system is driven by *arbitrary adversarial* noise [3]. We attain the optimal $\widetilde{\mathcal{O}}(\sqrt{T})$ regret when the dynamics are unknown to the learner, and $\mathrm{poly}(\log T)$ regret when known, provided that the cost functions are strongly convex (as in LQR). Our algorithm is based on a novel variant of online Newton step [19], which adapts to the geometry induced by adversarial disturbances, and our analysis hinges on generic regret bounds for certain structured losses in the OCO-with-memory framework [6].

## 1 Introduction

In control tasks, a learning agent seeks to minimize cumulative loss in a dynamic environment which responds to its actions. While dynamics make control problems immensely expressive, they also pose a significant challenge: the learner's past decisions affect future losses incurred.

This paper focuses on the widely-studied setting of linear control, where the the learner's environment is described by a continuous state, and evolves according to a linear system of equations, perturbed by *process noise*, and guided by inputs chosen by the learner. Many of the first learning-theoretic results for linear control focused on *online LQR* [1, 13, 12, 25], an online variant of the classical *Linear Quadratic Regulator (LQR)* [21]. In online LQR, the agent aims to control an unknown linear dynamical system driven by independent, identically distributed Gaussian process noise. Performance is measured by regret against the optimal LQR control law on a time horizon $T$, for which the optimal regret rate is $\widetilde{\Theta}(\sqrt{T})$ [12, 25, 26, 9]. Theoretical guarantees for LQR rely heavily on the strong stochastic modeling assumptions for the noise, and may be far-from-optimal if these assumptions break. A complementary line of work considers *non-stochastic control*, replacing stochastic process noise with adversarial disturbances to the dynamics [3, 28]. Here, performance is measured by *regret*: performance relative to the best (dynamic) linear control policy in hindsight, given full knowledge of the adversarial perturbations.

Though many works have proposed efficient algorithms which attain sublinear regret for non-stochastic control, they either lag behind optimal guarantees for the stochastic LQR problem, or require partial stochasticity assumptions to ensure their regret. And while there is a host of literature demonstrating that, in many online learning problems without dynamics, the worst-case rates of regret for the adversarial and stochastic settings are the same [8, 31, 19], whether this is true

in control is far from clear. Past decisions affect future losses in control settings, and this may be fundamentally more challenging when perturbations are adversarial and unpredictable. Despite this challenge, we propose an efficient algorithm that matches the optimal $\sqrt{T}$ regret bound attainable the stochastic LQR problem, but under arbitrary, non-stochastic disturbance sequences and arbitrary strongly convex costs. Thus, from the perspective of regret with respect to a benchmark of linear controllers, we show that the optimal rate for non-stochastic control matches the stochastic setting.

**Our Setting**   Generalizing LQR, we consider partially-observed linear dynamics :

$$\mathbf{x}_{t+1} = A_\star \mathbf{x}_t + B_\star \mathbf{u}_t + \mathbf{w}_t, \quad \mathbf{y}_t = C_\star \mathbf{x}_t + \mathbf{e}_t \tag{1.1}$$

Here, the state $\mathbf{x}_t$ and process noise $\mathbf{w}_t$ lie in $\mathbb{R}^{d_x}$, the observation $\mathbf{y}_t$ and observation noise $\mathbf{e}_t$ lie in $\mathbb{R}^{d_y}$, and the input $\mathbf{u}_t \in \mathbb{R}^{d_u}$ is elected by the learner, and $A_\star, B_\star, C_\star$ are matrices of appropriate dimensions. We call the $(\mathbf{w}_t, \mathbf{e}_t)$ the *disturbances*, and let $(\mathbf{w}, \mathbf{e})$ denote the entire disturbance sequence. Unlike LQR, we assume that the disturbances are selected by an oblivious[1] adversary, rather than from a mean zero stochastic process, and the learner observes the outputs $\mathbf{y}_t$, but not the full state $\mathbf{x}_t$. Appendix C describes how our setting strictly generalizes the online LQR problem, and relates to its partially observed analogue LQG. A *policy* $\pi$ is a (possibly randomized) sequence of mappings $\mathbf{u}_t := \pi_t(\mathbf{y}_{1:t}, \mathbf{u}_{1:t-1})$. We denote by $\mathbf{y}_t^\pi$ and $\mathbf{u}_t^\pi$ sequence the realized sequence of outputs and inputs produced by policy $\pi$ and the noise sequence $(\mathbf{w}, \mathbf{e})$. At each time $t$, a convex cost $\ell_t : \mathbb{R}^{d_y \times d_u} \to \mathbb{R}$ is revealed, and the learner observes the current $\mathbf{y}_t$, and suffers loss $\ell_t(\mathbf{y}_t, \mathbf{u}_t)$. The *cost functional* of a policy $\pi$ is

$$J_T(\pi) := \sum_{t=1}^T \ell_t(\mathbf{y}_t^\pi, \mathbf{y}_t^\pi),$$

measuring the cumulative losses evaluated on the outputs and inputs induced by the realization of the disturbances $(\mathbf{w}, \mathbf{e})$. The learner's policy alg, is chosen to attain low *control regret* with respect to a pre-specified benchmark class $\Pi$ of reference policies,

$$\text{ControlReg}_T(\mathsf{alg}; \Pi) := J_T(\mathsf{alg}) - \inf_{\pi \in \Pi} J_T(\pi), \tag{1.2}$$

which measures the performance of alg (on the realized losses/disturbances) compared to the best policy $\pi \in \Pi$ in hindsight (chosen with knowledge of losses and disturbances). We consider a restricted a benchmark class $\Pi$ consisting of linear, dynamic controllers, formalized in Definition 3.1. While this class encompasses optimal control laws for many classical settings [28], in general it *does not* include the optimal control law for a given realization of noise. This is unavoidable: even in the simplest settings, it is impossible to attain sublinear regret with respect to the optimal control law [24]. We assume that the losses $\ell_t(\cdot)$ are $\alpha$-strongly convex, and grow at most quadratically:

**Assumption 1.** We suppose that all $\ell_t : \mathbb{R}^{d_y + d_u} \to \mathbb{R}$ are *L-subquadratic:* $0 \le \ell(v) \le L \max\{1, \|v\|_2^2\}$, and $\|\nabla \ell(v)\|_2 \le L \max\{1, \|v\|\}$. We also assume that $\ell_t$ are twice-continuously differentiable, and $\alpha$-strong convex ($\nabla^2 \ell_t \succeq \alpha I$). For simplicity, we assume $L \ge \max\{1, \alpha\}$.

This assumption is motivated by classical LQR/LQG, where the loss is a strongly convex quadratic of the form $\ell(y, u) = y^\top R y + u^\top Q u$ for $R, Q \succ 0$. The central technical challenge of this work is that, unlike standard online learning settings, *the strong convexity of the losses does not directly yield fast rates* [4, 16].

## 1.1 Our Contributions

For the above setting, we propose *Disturbance Reponse Control via Online Newton Step*, or DRC-ONS - an adaptive control policy which attains fast rates previously only known for settings with stochastic or semi-stochastic noise [25, 28, 12, 4]. Our algorithm combines the DRC controller parametrization [28] with Semi-ONS, a novel second-order online learning algorithm tailored to our setting. We show that DRC-ONS achieves logarithmic regret when the learner knows the dynamics:

**Theorem 3.1** (informal)   *When the agent knows the dynamics* (1.1) *(but does not have foreknowledge of disturbances nor the costs $\ell_t$),* DRC-ONS *has* $\text{ControlReg}_T = \mathcal{O}(\frac{L^2}{\alpha} \cdot \text{poly}(\log T))$.

This is the first bound to guarantee logarithmic regret with general strongly convex losses and non-stochastic noise. Past work required stochastic or semi-stochastic noise [4, 28], or was limited to fixed quadratic costs [16]. For unknown dynamics, we find:

**Theorem 3.2** (informal) *When the dyamics are unknown,* DRC-ONS *with an initial estimation phase attains* $\mathrm{ControlReg}_T = \widetilde{\mathcal{O}}(\frac{L^2}{\alpha}\sqrt{T})$.

This bound matches the optimal $\sqrt{T}$-scaling for *stochastic* online LQR [26]. Thus, from the perspective of regret minimization with respect to the benchmark $\Pi$, non-stochastic control is *almost* as easy as stochastic. This is not without many caveats, which are left to the discussion in Appendix B.1.

**Technical Contributions**   While our main results are control theoretic, our major technical insights pertain to online convex optimization (OCO). Our control algorithm leverages a known reduction [3] to the online convex optimization with memory (OCOM) framework [6], which modifies OCO by allowing losses to depend on past iterates. Past OCOM analyses required bounds on both the standard OCO regret and total Euclidean variation of the iterates produced (Section 2.4). But for the the losses that arise in our setting, Theorem 2.3 shows that there is a significant tradeoff between the two, obviating sharp upper bounds.   To overcome this , we show that online control enjoys additional structure we call OCO *with affine memory*, or OCOAM. We propose a novel second order method, Semi-ONS, based on online Newton step (ONS, [19]), tailored to this structure. Under a key technical condition satisfied by online control, we establish logarithmic regret.

**Theorem 2.1** (informal)   *Under the aforementioned assumption (Definition 2.2), the* Semi-ONS *algorithm attains* $\mathcal{O}\left(\frac{1}{\alpha}\log T\right)$ *regret in the* OCOAM *setting.*

The above bound directly translates to logarithmic control regret for known systems, via the control-to-OCOAM reduction spelled out in Section 3. For control of unknown systems, the undergirding OCOAM bound is quadratic sensitivity to $\epsilon$-approximate losses:

**Theorem 2.2** (informal)   *Consider the* OCOAM *setting with $\epsilon$-approximate losses (in the sense of Assumption 2). Then,* Semi-ONS *has regret* $\mathcal{O}\left(\frac{1}{\alpha}\log T \cdot T\epsilon^2\right)$.

Quadratic sensitivity to errors in the gradients was previously demonstrated for strongly convex stochastic optimization [15], and subsequently for strongly convex OCO [28]. Extending this guarantee to Semi-ONS is the most intricate technical undertaking of this paper.

## 1.2   Prior Work

In the interest of brevity, we restrict our attention to previous works regarding online control with a regret benchmark; for a survey of the decades old field of adaptive control, see e.g. [29]. Much work has focused on obtaining low regret in online LQR with unknown dynamics [1, 13, 25, 12], a setting we formally detail in Appendix C.1. Recent algorithms [25, 12] attain $\sqrt{T}$ regret for this setting, with polynomial runtime and polynomial regret dependence on relevant problem parameters. This was recently demonstrated to be optimal [26, 9], with Cassel et al. [9] showing that logarithmic regret is possible the partial system knowledge. In the related LQG setting (partial-observation, stochastic process and observation noise, Appendix C.2), Mania et al. [25] present perturbation bounds which suggest $T^{2/3}$ regret, improve to $\sqrt{T}$ by Lale et al. [23], matching the optimal rate for LQG. For LQG with both non-denegerate process and observation noise, Lale et al. [22] attain $\mathrm{poly}(\log T)$ regret, demonstrating that in the presence of observation, LQG is in fact *easier* than LQR (with no observation noise) in terms of regret; see Appendix B.1 for further discussion.

Recent work first departed from online LQR by considered adversarially chosen costs under known stochastic or noiseless dynamics [2, 11]. Agarwal et al. [4] obtain logarithmic regret for fully observed systems, stochastic noise and adversarially chosen, strongly convex costs. The non-stochastic control setting we consider in this paper was established in Agarwal et al. [3], who obtain $\sqrt{T}$-regret for convex, Lipschitz (not strongly convex) cost functions and known dynamics. Hazan et al. [20] attains $T^{2/3}$ regret for the same setting with unknown dynamics. Simchowitz et al. [28] generalizes both guarantees to partial observation, and generalize the optimal rate of logarithmic and $\sqrt{T}$ for known and unknown systems, respectively to strongly convex losses and a 'semi-stochastic' noise model. This assumption requires the noise to have a well-conditioned, stochastic component; in contrast, our methods allow *truly adversarial* noise sequences. Lastly, for the known system setting, Foster and Simchowitz [16] propose a different paradigm which yields logarithmic regret with

truly adversarial noise, but fixed quadratic cost functions and with full observation. In contrast, our algorithm accomodates both partial observation and arbitrary, changing costs, and its analysis and presentation are considerably simpler. Our work also pertains to the broader literature of on-line optimization with policy regret and loss functions with memory [7, 6], and our lower bound (Theorem 2.3) draws on the learning-with-switching-costs literature [5, 10, 14].

### 1.3 Organization and Notation

Section 2 formulates the general OCoAM setting, describes our Semi-ONS algorithm, and states its guarantees (Theorems 2.1 and 2.2), and the regret-movement tradeoff that hindered past approaches (Theorem 2.3). Section 3 turns to the control setting, describing the reduction to OCoAM, the DRC-ONS algorithm, and stating our main results (Theorems 3.1 and 3.2). Discussion of our results is deferred to Appendix B.1. All proofs are deferred to our appendix, whose organization of the appendix is detailed in Appendix A. Throughout, let $a \lesssim b$ denote that $a \leq Cb$, where $C$ is a universal constant independent of problem parameters. We use $\Omega(\cdot), \mathcal{O}(\cdot)$ as informal asymptotic notation. We let $a \vee b$ denote $\max\{a, b\}$, and $a \wedge b$ to denote $\min\{a, b\}$. For vectors $x$ and $\Lambda \succeq 0$, we denote $\|x\|_\Lambda := \sqrt{x^\top \Lambda x}$, and use $\|x\|$ and $\|x\|_2$ interchangeably for Euclidean norm. We let $\|A\|_{\mathrm{op}}$ denote the operator norm, and given a sequence of matrices $G = (G^{[i]})_{i \geq 0}$, we define $\|G\|_{\ell_1, \mathrm{op}} := \sum_{i \geq 0} \|G^{[i]}\|_{\mathrm{op}}$. We use $[(\cdot); (\cdot)]$ to denote vertical concatenation of vectors and matrices. Finally, non-bold arguments (e.g. $z$) denote function arguments, and bold (e.g. $\mathbf{z}_t$) denote online iterates.

## 2 Fast Rates for OCO with Affine Memory

Building on past work [28, 3], our results for control proceed via a reduction to online convex optimization (OCO) with memory, proposed by Anava et al. [6], and denoted by OCoM in this work. Our lower bound in Section 2.5 explains why this past strategy is insufficient. Thus, we consider a structured special case, OCoAM , which arises in control, present a second-order algorithm for this setting, Semi-ONS, and state its main guarantees.

**OCoM preliminaries** Let $\mathcal{C} \subset \mathbb{R}^d$ be a convex constraint set. OCoM is an online learning game where, at each time $t$, the learner plays an input $\mathbf{z}_t \in \mathcal{C}$, nature reveals an $h + 1$-argument loss $F_t : \mathcal{C}^{h+1} \to \mathbb{R}$, and the learner suffers loss $F_t(\mathbf{z}_t, \mathbf{z}_{t-1}, \ldots, \mathbf{z}_{t-h})$, abbreviated as $F_t(\mathbf{z}_{t:t-h})$. For each $F_t$, we define its *unary specialization* $f_t(z) := F_t(z, \ldots, z)$. The learner's performance is measured by what we term *memory-regret*:[2]

$$\mathrm{MemoryReg}_T := \sum_{t=1}^T F_t(\mathbf{z}_{t:t-h}) - \inf_{z \in \mathcal{C}} \sum_{t=1}^T f_t(z). \tag{2.1}$$

Because the learner's loss is evaluated on a history of past actions, OCoM encodes learning problems with dynamics, such as our control setting. This is in contrast to the standard OCo setting, which measures regret evaluated on the unary $f_t$: $\mathrm{OCoReg}_T := \sum_{t=1}^T f_t(\mathbf{z}_t) - \inf_{z \in \mathcal{C}} \sum_{t=1}^T f_t(z)$. Our goal is to attain logarithmic memory-regret, and quadratic sensitivity to structured errors (in a sense formalized below).

### 2.1 OCO with Affine Memory

While we desire logarithmic memory regret, Theorem 2.3 shows that existing analyses cannot yield better rates than $\Omega(T^{1/3})$. Luckily, the control setting gives us more structure. Let us sketch this with a toy setting, and defer the full reduction to Section 3. Consider a nilpotent, fully observed system: $\mathbf{y}_t \equiv \mathbf{x}_t$, and $A_\star^h = 0$. Defining $G^{[i]} := [A_\star^{i-1} B_\star; I \cdot \mathbb{I}_{i=0}]$, the linear dynamics give $[\mathbf{x}_t; \mathbf{u}_t] := \sum_{i=0}^h G^{[i]} \mathbf{u}_{t-i} + [\mathbf{x}_{t,0}; 0]$, where $\mathbf{x}_{t,0} = \sum_{i=0}^h A_\star^i \mathbf{w}_{t-i}$ . For simple policies parametrized by $\mathbf{u}_t^z = z \cdot \mathbf{w}_t, z \in \mathbb{R}$, the loss incured under iterates $z_{t:t-h}$, $\ell_t([\mathbf{x}_{t,0}; 0] + \sum_{i=0}^h G^{[i]} \mathbf{w}_{t-i} z_{t-i}) =: F_t(z_{t:t-h})$, exhibits *affine* dependence on the past. Generalizing the above, the OCo with affine memory (OCoAM) setting is as follows. Fix $G = (G^{[i]})_{i \geq 0} \in (\mathbb{R}^{p \times d_{\mathrm{in}}})^{\mathbb{N}}$ across rounds. At each $t \geq 1$, the learner selects $\mathbf{z}_t \in \mathcal{C} \subset \mathbb{R}^d$, and the adversary reveals a convex cost $\ell_t : \mathbb{R}^p \to \mathbb{R}$,

an offset vector $\mathbf{v}_t \in \mathbb{R}^p$, and a matrix $\mathbf{Y}_t \in \mathbb{R}^{d_{\mathrm{in}} \times d}$. The learner suffers loss with-memory loss $F_t(\mathbf{z}_{t:t-h})$, given by $F_t(z_{t:t-h}) := \ell_t(\mathbf{v}_t + \sum_{i=0}^{h} G^{[i]} \mathbf{Y}_{t-i} z_{t-i})$. The induced unary losses are

$$f_t(z) := \ell_t(\mathbf{v}_t + \mathbf{H}_t z), \quad \text{where } \mathbf{H}_t := \sum_{i=0}^{h} G^{[i]} \mathbf{Y}_{t-i}. \tag{2.2}$$

We consider two settings for OCOAM. In the *exact* setting, $G$ is known to the learner, and $\ell_t, \mathbf{v}_t, \mathbf{Y}_t$ are revealed at each $t$. Thus $f_t$ and $\mathbf{H}_t$ can be computed after each round. The *approximate* setting, the learner knows only an approximation $\widehat{G}$ of $G$, and recieves an estimate $\widehat{\mathbf{v}}_t$ of $\mathbf{v}_t$ ($\mathbf{Y}_t$ and $\ell_t$ remain exact). Our algorithm uses approximate unary losses:

$$\widehat{f}_t(z) := \ell_t(\widehat{\mathbf{v}}_t + \widehat{\mathbf{H}}_t z), \quad \text{where } \widehat{\mathbf{H}}_t := \sum_{i=0}^{h} \widehat{G}^{[i]} \mathbf{Y}_{t-i}. \tag{2.3}$$

We desire low sensitivity to the approximation errors of $\widehat{G}$ and $\widehat{\mathbf{v}}$, translating to low estimation error sensitivity for control of an unknown system. For both exact and approximate losses, memory regret is evaluated on the *exact* losses $F_t, f_t$, consistent with OCOM.

## 2.2 The Semi-ONS algorithm

The standard algorithmic template for OCOM is to run an online optimization procedure on the unary losses $f_t$, otherwise disregarding $F_t$ (but accounting for the discrepancy between the two in the analysis) [6]. We take this approach here, but with a tailored second order method. Let $\mathbf{z}_{t-h+1}, \ldots, \mathbf{z}_0 \in \mathcal{C}$ be arbitrary initial parameters. For step size and regularization parameters $\eta > 0$ and $\lambda > 0$, and setting $\nabla_t := \nabla f_t(\mathbf{z}_t)$, the Semi-ONS(Algorithm 1) iterates are:

$$\tilde{\mathbf{z}}_{t+1} \leftarrow \mathbf{z}_t - \eta \Lambda_t^{-1} \nabla_t, \ \mathbf{z}_{t+1} \leftarrow \arg\min_{z \in \mathcal{C}} \|\Lambda^{1/2}(\tilde{\mathbf{z}}_{t+1} - z)\|, \ \Lambda_t := \lambda I + \sum_{s=1}^{t} \mathbf{H}^\top \mathbf{H}_t, \quad (2.4)$$

The updates are nearly identical to online Newton step (ONS) [19], but whereas the ONS uses preconditioner $\Lambda_{t,\mathrm{ONS}} := \lambda I + \sum_{s=1}^{t} \nabla f_t(\mathbf{z}_t) \nabla f_t(\mathbf{z}_t)^\top$, Semi-ONS uses outer products of $\mathbf{H}_t$. This decision is explained in the paragraph concluding Section 2.4. In the approximate setting Semi-ONS proceeds using the following approximations, with $\widehat{\nabla}_t := \nabla \widehat{f}_t(\mathbf{z}_t)$

$$\tilde{\mathbf{z}}_{t+1} \leftarrow \mathbf{z}_t - \eta \widehat{\Lambda}_t^{-1} \widehat{\nabla}_t, \ \mathbf{z}_{t+1} \leftarrow \arg\min_{z \in \mathcal{C}} \|\widehat{\Lambda}^{1/2}(\tilde{\mathbf{z}}_{t+1} - z)\|, \ \widehat{\Lambda}_t := \lambda I + \sum_{s=1}^{t} \widehat{\mathbf{H}}^\top \widehat{\mathbf{H}}_t, \quad (2.5)$$

defined using the quantities in Eq. (2.3). In other words, approximate Semi-ONS is equivalent to exact Semi-ONS, treating $(\widehat{f}_t, \widehat{\mathbf{H}}_t)$ like the true $(f_t, \mathbf{H}_t)$.

---

**parameters**: Learning rate $\eta > 0$, regularization parameter $\lambda > 0$, convex domain $\mathcal{C} \subset \mathbb{R}^d$.
**initialize**: $\Lambda_0 = \lambda \cdot I_d$, $\mathbf{z}_1 \leftarrow \mathbf{0}_d$
**for** $t = 1, 2, \ldots$: **do**
    **recieve** triple $(\ell_t, \mathbf{v}_t, \mathbf{H}_t)$. % For approximate setting, replace $(\mathbf{v}_t, \mathbf{H}_t) \leftarrow (\widehat{\mathbf{v}}_t, \widehat{\mathbf{H}}_t)$
    $\nabla_t \leftarrow \nabla f_t(\mathbf{z}_t)$, where $f_t(z) = \ell_t(\mathbf{v}_t + \mathbf{H}_t z)$.
    $\Lambda_t \leftarrow \Lambda_{t-1} + \mathbf{H}_t^\top \mathbf{H}_t$ .
    $\widetilde{\mathbf{z}}_{t+1} \leftarrow \mathbf{z}_t - \eta \Lambda_t^{-1} \nabla_t$.
    $\mathbf{z}_{t+1} \leftarrow \arg\min_{z \in \mathcal{C}} \|\Lambda_t^{1/2}(z - \widetilde{\mathbf{z}}_{t+1})\|_2^2$.

**Algorithm 1:** Online Semi-Newton Step - Semi-ONS$(\lambda, \eta, \mathcal{C})$

---

## 2.3 Guarantees for Semi-ONS

To state our guarantees, we assume the $\alpha$-strong convexity and $L$-subquadratic assumption of Assumption 1. We assume various upper bounds on relevant quantities:

**Definition 2.1** (Bounds on Relevant Parameters). We assume $\mathcal{C}$ contains the origin. Further, we define the diameter $D := \max\{\|z - z'\| : z, z' \in \mathcal{C}\}$, $Y$-radius $R_Y := \max_t \|\mathbf{Y}_t\|_{\mathrm{op}}$, and $R_{Y,\mathcal{C}} := \max_t \max_{z \in \mathcal{C}} \|\mathbf{Y}_t z\|$; In the *exact* setting, we define the radii $R_v := \max_t \max\{\|\mathbf{v}_t\|_2\}$ and $R_G := \max\{1, \|G\|_{\ell_1,\mathrm{op}}\}$. In the *approximate* setting, $R_v := \max_t \max\{\|\mathbf{v}_t\|_2, \|\widehat{\mathbf{v}}_t\|_2\}$, $R_G := \max\{1, \|G\|_{\ell_1,\mathrm{op}}, \|\widehat{G}\|_{\ell_1,\mathrm{op}}\}$; For settings, we define the $H$-radius $R_H = R_G R_Y$, and define the effective Lipschitz constant $L_{\mathrm{eff}} := L \max\{1, R_v + R_G R_{Y,\mathcal{C}}\}$.

Lastly, our analysis requires that the smallest singular value of $G$, viewed as linear operator acting by convolution with sequences $(u_1, u_2, \dots) \in (\mathbb{R}^{d_{\text{in}}})^{\mathbb{N}}$, is bounded below:

**Definition 2.2.** We define the *convolution invertibility-modulus* as $\kappa(G) := 1 \wedge \inf_{(u_0, u_1, \dots)}$ $\{\sum_{n \geq 0} \|\sum_{i=0}^{n} G^{[i]} u_{n-i}\|_2^2 : \sum_t \|u_t\|_2^2 = 1\}$, and the *decay-function* $\psi_G(n) := \sum_{i \geq n} \|G^{[i]}\|_{\text{op}}$.

A Fourier-analytic argument (Lemma 3.1) demonstrates that $\kappa(G) > 0$ when expressing reducing our control setting to OCOAM (Section 3), and stability of our control parametrization ensures $\psi_G(n)$ decays exponentially; the reader should have in mind the scalings $\kappa(G) = \Omega(1)$ and $\psi_G(n) = \exp(-\Omega(n))$. For the exact setting, we have the following guarantee:

**Theorem 2.1** (Semi-ONS regret, exact case). *Suppose $\kappa = \kappa(G) > 0$, Assumption 1 holds, and consider the update rule Eq. (2.4) with parameters $\eta = \frac{1}{\alpha}$, $\lambda := 6hR_Y^2 R_G^2$. Suppose in addition that $h$ is large enough to satisfy $\psi_G(h+1)^2 \leq R_G^2/T$. Then, we have* $\text{MemoryReg}_T \leq 3\alpha h D^2 R_H^2 + \frac{3dh^2 L_{\text{eff}}^2 R_G}{\alpha \kappa^{1/2}} \log(1+T)$.

The above regret mirrors fast rates for strongly convex rates OCOM and exp-concave standard OCO. Its proof departs significantly from those of existing OCOM bounds, and is sketched in Section 2.4, and formalized in Appendix F. For the approximate setting, we assume

**Assumption 2** (Approximate Semi-ONS assumptions). We assume that $\|\widehat{G} - G_\star\|_{\ell_1, \text{op}} \leq \epsilon_G$, $\max_{t \geq 1} \|\mathbf{v}_t - \widehat{\mathbf{v}}_t\|_2 \leq c_v \epsilon_G$ for some $c_v > 0$, and that $\widehat{G}^{[i]} = 0$ for all $i > h$.

For simplicity, the following theorem considers $\epsilon_G^2 \geq 1/\sqrt{T}$, which arises in our estimation-exploitation tradeoff for control of unknown linear systems. It shows that Semi-ONS exhibits a quadratic sensitivity to the estimation error $\epsilon_G$, with $\text{MemoryReg}_T$ scaling as $\frac{1}{\alpha} \log T \cdot T \epsilon_G^2$.

**Theorem 2.2** (Semi-ONS regret, approximate case). *Suppose Assumptions 1 and 2 holds, and in addition $\nabla^2 \ell_t \preceq LI$ uniformly, and $\epsilon_G^2 \geq 1/\sqrt{T}$. Consider the update rule Eq. (2.5) with parameters $\eta = \frac{3}{\alpha}$ and $\lambda = (T\epsilon_G^2 + hR_G^2)$. Then* $\text{MemoryReg}_T \lesssim \log T \left( \frac{C_1}{\alpha \kappa^{1/2}} + C_2 \right)$ $\cdot \left( T\epsilon_G^2 + h^2(R_G^2 + R_Y) \right)$, *where* $C_1 := (1 + R_Y) R_G (h + d) L_{\text{eff}}^2$ *and* $C_2 := (L^2 c_v^2/\alpha + \alpha D^2)$.

The above mirrors the strongly convex setting, where online gradient descent with $\epsilon$-approximate gradients attains $\frac{1}{\alpha} T \epsilon^2$ regret [28]. In Appendix G we provide two stronger versions: The first (Theorem 2.2a) includes a certain negative regret term which is indispensible for the control setting, and accomodates misspecified $\lambda$. The second (Theorem G.1) allows for $\epsilon_G^2 \ll 1/\sqrt{T}$, establishing $(T\epsilon_G)^{2/3}$ regret for small $\epsilon_G$. Appendix G also details the proof of Theorem 2.2, which constitutes the main technical undertaking of the paper. The proof draws heavily on ideas from the proof of Theorem 2.1, which we presently sketch.

## 2.4 Proof Sketch for Exact Semi-ONS (Theorem 2.1)

Recall the with-memory and unary regret defined at the start of Section 2, and set $\nabla_t := \nabla f_t(\mathbf{z}_t)$. Following [6], our analysis begins with the following identity:

$$\text{MemoryReg}_T = \text{OCOReg}_T + \text{MoveDiff}_T, \quad \text{where MoveDiff}_T := \sum_{t=1}^{T} F(\mathbf{z}_{t:t-h}) - f(\mathbf{z}_t).$$

That is, $\text{MemoryReg}_T$ equals the standard regret on the $f_t$ sequence, plus the cumulative difference between $F_t$ (with memory) and $f_t$ (unary). The bound on $\text{OCOReg}_T$ for Semi-ONS mirros the analysis of standard ONS, using that $\nabla^2 f_t(\mathbf{z}_t) \succsim \mathbf{H}_t^\top \mathbf{H}_t \succsim \nabla_t \nabla_t^\top$ (Lemma F.2). To bound $\text{MoveDiff}_T$, past work on OCOM applies the triangle inequality and an $L$-Lipschitz condition on $F$ to bound the movement difference by movement in the Euclidean norm:

$$\text{MoveDiff}_T \leq \text{poly}(L, h) \cdot \text{EucCost}_T, \quad \text{where EucCost}_T := \sum_{t=1}^{T} \|\mathbf{z}_t - \mathbf{z}_{t-1}\|. \quad (2.6)$$

The standard approach is to run OGD on the unary losses [6] When doing so, the differences $\|\mathbf{z}_t - \mathbf{z}_{t-1}\|$ scale with Lipschitz constant $L$ and step sizes $\eta_t$. In particular, for the standard $\eta_t \propto \frac{1}{\alpha t}$ step size for $\alpha$-strongly convex losses, $\sum_{t=1}^{T} \|\mathbf{z}_t - \mathbf{z}_{t-1}\| = \mathcal{O}(\frac{1}{\alpha} \log T)$. Since OGD also has logarithmic unary regret, we obtain $\mathcal{O}(\frac{\text{poly}(L,h)}{a} \log T)$ memory regret. However, when $\ell_t$ are strongly convex,

the induced OCOAM losses $f_t$ need not be [16], and Theorem 2.3 shows that it is impossible to attain both logarithmic regret and logarithmic movement cost simultaneously. As a work around, we establish a refined movement bound in terms of $\mathbf{Y}_t$-sequence (see Lemma F.6):

$$\text{MoveDiff}_T \leq \text{poly}(L,h) \cdot \text{AdapCost}_T, \ \text{AdapCost}_T := \sum_{i=1}^{h}\sum_{t=1}^{T} \|\mathbf{Y}_t(\mathbf{z}_{t-i} - \mathbf{z}_{t-i-1})\|_2,$$

Via Lemma F.7, the Semi-ONS updates and an application of Cauchy-Schwartz yields:

$$\text{AdapCost}_T \leq \mathcal{O}\left(\text{poly}(L,h)\right) \cdot \Big( \underbrace{\textstyle\sum_{t=1}^{T} \nabla_t^\top \Lambda_t^{-1} \nabla_t}_{\nabla\text{-movement}} \Big)^{1/2} \cdot \Big( \underbrace{\textstyle\sum_{t=1}^{T} \mathbf{Y}_t^\top \Lambda_t^{-1} \mathbf{Y}_t}_{\mathbf{Y}\text{-movement}} \Big)^{1/2}. \quad (2.7)$$

Readers familiar with the analysis of ONS will recognize the $\nabla$-movement as the dominant term in its regret bound, and can be bounded in a similar fashion. To address the $\mathbf{Y}$-movement, we use the convolution-invertibility assumption (Definition 2.2). This assumption implies that convolution with $G = (G^{[i]})_{i \geq 0}$ is invertible, meaning that we can essentially invert the sequence $(\mathbf{H}_1, \mathbf{H}_2, \dots)$ defined by $\mathbf{H}_t := \sum_{i=0}^{h} G^{[i]} \mathbf{Y}_{t-i}$ so as to back out $(\mathbf{Y}_1, \mathbf{Y}_2, \dots)$. Linear algebraically, this implies (see Proposition F.8) $\Lambda_t - \lambda I = \sum_{s=1}^{t} \mathbf{H}_s^\top \mathbf{H}_s \succeq \frac{\kappa(G)}{2}\sum_{s=1}^{t} \mathbf{Y}_t^\top \mathbf{Y}_t - \mathcal{O}(1)$. In other words, up to an additive remainder term and multiplicative factor of $\kappa(G)$, the $\mathbf{H}_s$-covariance dominates that $\mathbf{Y}_s$-covariance. Hence, $\Lambda_t$ roughly dominates $\sum_{s=1}^{t-1} \mathbf{Y}_s^\top \mathbf{Y}_s + \lambda I$. Hence, $\mathbf{Y}$-movement is also $\mathcal{O}(d \log T)$ by an application of the log-determinant lemma (Lemma F.5). This yields a logarithmic upper bound on MoveDiff, and thus logarithmic memory regret.

**Semi-ONS v.s. ONS**  Standard ONS uses a preconditioner based on outer products of $\nabla_t$. However, the movement difference depends on gradients of the with-memory loss $F_t(\cdot, \dots, \cdot)$, which may not be aligned with direction of $\nabla_t$. Indeed, $\nabla_t \in \text{RowSpace}(\mathbf{Y}_t)$, but this is in general a strict inclusion; that is, $\mathbf{Y}_t$ accounts for more possible directions of movement that $\nabla_t$. Thus, Semi-ONS forms its preconditioner to ensure slower movement in all $\mathbf{Y}_t$-directions, using $\mathbf{H}_t$ as a proxy via the convolution-invertibility analysis.

### 2.5  The Regret-Movement Tradeoff

As described above, the standard analysis of OCOM bounds the sum of the unary regret and Euclidean total variation of the iterates. While this permits logarithmic regret when $f_t$ are strongly convex, OCOAM losses $f_t$ are not strongly convex even if $\ell_t$ are (see e.g. below). We now show that for a simple class of quadratic OCOAM losses, there is a nontrivial trade-off between the two terms. We lower bound $\mu\text{-Reg}_T := \text{OcoReg}_T + \mu\text{EucCost}_T = \sum_{t=1}^{T} f_t(\mathbf{z}_t) + \mu\|\mathbf{z}_t - \mathbf{z}_{t-1}\| - \inf_{z \in \mathcal{C}} \sum_{t=1}^{T} f_t(\mathbf{z}_t)$, which characterizes the Pareto curve between unary regret and Euclidean movement. We consider $d = 1$, $\mathcal{C} = [-1,1]$, $\ell(u) = u^2$, and the memory-1 OCOAM losses $f_t = \ell(\mathbf{v}_t - \epsilon z)$, where $\epsilon \in (0,1]$ is fixed and $\mathbf{v}_t \in \{-1,1\}$ are chosen by an adversary . On $\mathcal{C}$, $f_t$ are $\mathcal{O}(\epsilon)$-Lipschitz, and have Hessian $\epsilon^2$ (thus arbitrarily small strong convexity). Still, $\ell$ satisfies Assumption 1 with $\alpha = L = 1$. We prove the following in Appendix J.1:

**Theorem 2.3.** Let $c_1, \dots, c_4$ be constants. For $T \geq 1$ and $\mu \leq c_1 T$, there exists $\epsilon = \epsilon(\mu, T)$ and a joint distribution $\mathcal{D}$ over $\mathbf{v}_1, \dots, \mathbf{v}_T \in \{-1,1\}^T$ such that any proper (i.e. $\mathbf{z}_t \in \mathcal{C}$ for all $t$) possibly randomized algorithm alg suffers $\mathbb{E}[\mu\text{-Reg}_T] \geq c_2(T\mu^2)^{1/3}$. In particular, $\mathbb{E}[1\text{-Reg}_T] \geq c_2 T^{1/3}$, and if $\mathbb{E}[\text{OcoReg}_T] \leq R \leq c_3 T$, then, $\mathbb{E}[\text{EucCost}_T] \geq c_4\sqrt{T/R}$.

Hence, existing analyses based on Euclidean movement cannot ensure better than $T^{1/3}$ regret . Moreover, to ensure $\text{OcoReg}_T = \mathcal{O}(\log T)$, then one must suffer $\sqrt{T/\log T}$ movement. In Theorem J.1 in Appendix J.2, we show that standard ONS with an appropriately tuned regularization parameter attains this optimal tradeoff (up to logarithmic and dimension factors), even in the more general case of arbitrary exp-concave losses.

## 3  From OCOAM to Online Control

This sections proposes and analyzes the DRC-ONS algorithm via OCOAM. Recall the control setting with dynamics described by Eq. (1.1), and regret defined by Eq. (1.2). Throughout, we assume

that the losses satisfy the strong convexity and quadratic growth assumption of Assumption 1. Outputs $\mathbf{y}$ lie in $\mathbb{R}^{d_y}$, inputs $\mathbf{u}$ lie in $\mathbb{R}^{d_u}$. For the main text of this paper, we assume knowledge of a *stabilizing, static feedback* policy: that is a matrix $K \in \mathbb{R}^{d_u \times d_y}$ such that the policy $\mathbf{u}_t = K\mathbf{y}_t$ which is stabilizing ($\rho(A_\star + B_\star KC_\star) < 1$, where $\rho$ denotes the spectral radius). [3] For this stabilizing $K$, we select inputs $\mathbf{u}_t^{\text{alg}} := K\mathbf{y}_t^{\text{alg}} + \mathbf{u}_t^{\text{ex,alg}}$, where $\mathbf{u}_t^{\text{ex,alg}}$ is the *exogenous output* dictated by an online learning procedure. We let the *nominal iterates* $\mathbf{y}_t^K, \mathbf{u}_t^K$ denote the sequence of outputs and inputs that would occur by selecting $\mathbf{u}_t^{\text{alg}} = K\mathbf{y}_t^{\text{alg}}$, with no exogenous inputs. We exploit the superposition identity (using $[\cdot; \cdot]$ to denote vertical concatenation)

$$\left[\mathbf{y}_t^{\text{alg}}; \mathbf{u}_t^{\text{alg}}\right] = \left[\mathbf{y}_t^K; \mathbf{u}_t^K\right] + \sum_{i=0}^{t-1} G_K^{[i]} \mathbf{u}_{t-1}^{\text{ex}}, \tag{3.1}$$

where $G_K^{[0]} = [0; I_{d_u}]$ and $G_K^{[i]} = [C_\star; KC_\star](A_\star + B_\star KC_\star)^{i-1}B_\star$ for $i \geq 1$. We call $G_K$ the *nominal Markov operator*. Since $K$ is stabilizing, we will assume that $G_K^{[i]}$ decays geometrically, and that the nominal iterates are bounded. For simplicity, we take $\mathbf{x}_1 = 0$.

**Assumption 3.** For some $c_K > 0$ and $\rho_K \in (0, 1)$ and all $n \geq 0$, $\|G_K^{[i]}\|_{\text{op}} \leq c_K \rho_K^n$.

**Assumption 4.** We assume that $(\mathbf{w}_t, \mathbf{e}_t)$ are bounded such that, for all $t \geq 1$, $\|(\mathbf{y}_t^K, \mathbf{u}_t^K)\|_2 \leq R_{\text{nat}}$

Assumption 3 is analogous to "strong stability" [11], and holds for any stabilizing $K$. Assumption 4 is analogous to the bounded assumption in Simchowitz et al. [28]: since $K$ is stabilizing, any bounded sequence of disturbances implies a uniform upper bound on $\|(\mathbf{y}_t^K, \mathbf{u}_t^K)\|_2$[4]

**Benchmark Class**    We compete with linear dynamical controllers (LDCs) $\pi \in \Pi_{\text{ldc}}$ whose *closed loop* iterates are denoted $(\mathbf{y}_t^\pi, \mathbf{u}_t^\pi, \mathbf{x}_t^\pi)$ (see Definition E.2 for further details). These policies include static feedback laws $\mathring{\mathbf{u}}_t^\pi = K\mathring{\mathbf{y}}_t^\pi$, but *are considerably more general due to the internal state*. We consider *stabilizing* $\pi$: for all bounded disturbance sequences $\max_{t \geq 1} \|\mathbf{w}_t\|, \|\mathbf{e}_t\| < \infty$, it holds that $\max_{t \geq 1} \|\mathbf{y}_t^\pi\|, \|\mathbf{u}_t^\pi\| < \infty$. These policies enjoy geometric decay, motivating the following parametrization of our benchmark class.

**Definition 3.1** (Policy Benchmark)**.** Fix parameters $\rho_\star \in (0, 1)$ and $c_\star > 0$. Our regret benchmark competes LDC's $\pi \in \Pi_\star := \Pi_{\text{stab}}(c_\star, \rho_\star)$, where we define $\Pi_{\text{stab}}(c, \rho) := \{\pi \in \Pi_{\text{ldc}} : (\|G_{\pi,\text{cl}}^{[i]}\|_{\text{op}} \leq c\rho^n, \forall n \geq 0\}$, where the Markov operator $G_{\pi,\text{cl}}$ is in Definition E.3.

**Known v.s. Unknown Dynamics**    We refer to the *known* dynamics setting as the setting where the learner knows the matrices $A_\star, B_\star, C_\star$ defining the dynamics in Eq. (1.1). In the *unknown dynamics* setting, the learner does not know these matrices (but knows $K$).

**The DRC parametrization**    Given radius $R_\mathcal{M} > 0$ and memory $m \in \mathbb{N}$, we adopt the DRC parametrization of memory-$m$ controllers $M \in \mathcal{M}$ [28] :

$$\mathcal{M} = M_{\text{drc}}(m, R_\mathcal{M}) := \{M = (M^{[i]})_{i=0}^{m-1} \in (\mathbb{R}^{d_y d_u})^m : \sum_{i=0}^{m-1} \|M\|_{\text{op}} \leq R_\mathcal{M}\}. \tag{3.2}$$

Controllers $M \in \mathcal{M}$ are then applied to estimates of the nominal outputs $\mathbf{y}_t^K$. When the dynamics are known, $\mathbf{y}_t^K$ and $\mathbf{u}_t^K$ are recovered exactly via Eq. (3.1). If $A_\star, B_\star, C_\star$ are not known, we use an estimate $\widehat{G}$ of $G_K$ to construct estimates $\widehat{\mathbf{y}}_{1:t}^K, \widehat{\mathbf{u}}_{1:t}^K$:

$$\left[\widehat{\mathbf{y}}_t^K; \widehat{\mathbf{u}}_t^K\right] = \left[\mathbf{y}_t^{\text{alg}}; K\mathbf{y}_t^{\text{alg}}\right] - \sum_{i=1}^{t-1} \widehat{G}^{[i]} \mathbf{u}_{t-i}^{\text{ex,alg}}. \tag{3.3}$$

Going forward, we use the more general $\widehat{\mathbf{y}}_{1:t}^K$ notation, noting that it specializes to $\mathbf{y}_{1:t}^K$ for known systems (i.e. when $\widehat{G} = G_K$). The DRC parametrization selects exogenous inputs as linear combinations of $\widehat{\mathbf{y}}_{1:t}^K$ under $M \in \mathcal{M}$: via $\mathbf{u}_t^{\text{ex}}(M \mid \widehat{\mathbf{y}}_{1:t}^K) := \sum_{i=0}^{m-1} M^{[i]} \widehat{\mathbf{y}}_{t-i}^K$.

### 3.1 Reducing DRC to OCOAM

Fixing the DRC length $m \geq 1$, let $d = d_y d_u m$, and $p = d_y + d_u$. Further, let $(\widehat{\mathbf{y}}_t^K, \widehat{\mathbf{u}}_t^K)_{t \geq 1}$ and $\widehat{G}$ denote estimates of $(\mathbf{y}_t^K, \mathbf{u}_t^K)_{t \geq 1}$ and $G_K$, respectively

**Definition 3.2** (OCOAM quantities for control). Let $\mathfrak{e}[\cdot]$ denote the natural embedding of $M \in \mathcal{M}$ into $\mathbb{R}^d$, and let $\mathfrak{e}^{-1}[\cdot]$ denote its inverse; Define the OCOAM matrices $\mathbf{Y}_t := \mathfrak{e}_y[\widehat{\mathbf{y}}_{t:t-m+1}^K]$, where $\mathfrak{e}_y$ is embedding satisfying $\mathbf{Y}_t z = \mathbf{u}_t^{\text{ex}}(M \mid \widehat{\mathbf{y}}_{1:t}^K)$ for all $z$ of the form $z = \mathfrak{e}[M]$; Define the offset $\mathbf{v}_t^K = (\mathbf{y}_t^K, \mathbf{u}_t^K) \in \mathbb{R}^p$, and its approximation $\widehat{\mathbf{v}}_t^K = (\widehat{\mathbf{y}}_t^K, \widehat{\mathbf{u}}_t^K) \in \mathbb{R}^p$; Define the constraint set $\mathcal{C} := \mathfrak{e}(\mathcal{M}) \subset \mathbb{R}^d$ (that is, embed the DRC set into $\mathbb{R}^d$).

We now define the relevant OCOAM losses as those consistent with the above notation.

**Definition 3.3** (OCOAM losses for control). Let $\mathbf{Y}_t, \mathbf{v}_t^K, \widehat{\mathbf{v}}_t^K$ be as above. For $h \in \mathbb{N}$, define the *exact* losses $F_t(z_{t:t-h}) := \ell_t(\mathbf{v}_t^K + \sum_{i=0}^h G_K^{[i]} \mathbf{Y}_{t-i} z_{t-i})$, and $f_t(z) := \ell_t(\mathbf{v}_t^K + \mathbf{H}_t z)$, where $\mathbf{H}_t := \sum_{i=0}^h G_K^{[i]} \mathbf{Y}_{t-i}$. Given an estimate $\widehat{G}$ of $G_K$, the *approximate* unary loss is $\widehat{f}_t(z) := \ell_t(\widehat{\mathbf{v}}_t^K + \widehat{\mathbf{H}}_t z)$ with $\widehat{\mathbf{H}}_t := \sum_{i=0}^h \widehat{G}^{[i]} \mathbf{Y}_{t-i}$.

We take $h = \Theta(\log T)$, since the exponential decay assumption (Assumption 3) ensures $G_K^{[i]} = \exp(-\Omega(h)) \approx 0$ for $i > h$. The resulting OCOAM problem is to produce a sequence of iterates $\mathbf{z}_t$ minimizing $\text{MemoryReg}_T$ on the sequence $(F_t, f_t)$. Since $\mathbf{z}_t$ are embeddings of controllers, this gives rise to a natural control algorithm: for each iterate $\mathbf{z}_t$, back out a DRC controller $\mathbf{M}_t = \mathfrak{e}^{-1}(\mathbf{z}_t)$, and applies exogenous input $\mathbf{u}_t^{\text{ex,alg}} := \mathbf{u}_t^{\text{ex}}(\mathbf{M}_t \mid \mathbf{y}_{1:t}^K)$. In Appendix E, we streamline past work [28] by providing black-box reductions bounding the control regret (Eq. (1.2)) of such an algorithm by its memory regret. Proposition E.5 addresses the known system case, and Proposition E.8 the unknown case. Because the latter is more intricate, we conclude the present discussion with an informal statement of the known system reduction:

**Proposition E.5 (informal).** *Let algorithm* alg *which produces iterates* $\mathbf{z}_t \in \mathbb{R}^d$. *Let* alg$'$ *denote the control algorithm which selects* $\mathbf{u}_t^{\text{ex,alg}} := \mathbf{u}_t^{\text{ex}}(\mathbf{M}_t \mid \mathbf{y}_{1:t}^K)$, *where* $\mathbf{M}_t = \mathfrak{e}^{-1}(\mathbf{z}_t)$. *Then, for* $m = \widetilde{\mathcal{O}}(1)$, *we have* $\text{ControlReg}_T(\text{alg}') \leq \text{MemoryReg}_T(\text{alg}) + \mathcal{O}(1)$.

**Remark 3.1** (Hat-accent notation). We use $\mathbf{Y}_t$ even when defined using the approximate $\widehat{\mathbf{y}}_{1:t}^K$. However, $G$ and $\mathbf{v}^K$ do recieve hat-accents when estimates are used. This is because, while OCOAM can account for the approximation error on $G$ and $\mathbf{v}^K$ (Theorem 2.2), the approximation error introduced by setting $\mathbf{Y}_t := \mathfrak{e}_y[\widehat{\mathbf{y}}_{t:t-m+1}^K]$ requires control specific arguments

## 3.2 The DRC-ONS algorithm and guarantees

Stating the DRC-ONS algorithm is now a matter of putting the pieces together. For known systems, the learner constructs the losses in Definition 3.3 with $\widehat{G} = G_K$, and runs Semi-ONS on $f_t$, and uses these to perscribe a DRC controller in accordance with the above discussion. For unknown systems, one constructs the estimate $\widehat{G}$ via least squares, and then runs Semi-ONS on $\widehat{f}_t$; formal pseudocode is given in Algorithms 2 and 3 in Appendix D.1. Our formal guarantees are

**Theorem 3.1** (Guarantee for Known System). *Suppose Assumptions 1, 3 and 4 holds, and for given* $\rho_\star \in (0,1), c_\star > 0$, *let* $\Pi_\star$ *be as in Definition 3.1. For simplicity, also assume* $c_\star \geq c_K, \rho_\star \geq \rho_K$. *Then, for a suitable choice of parameters,* DRC-ONS(*Algorithm 2*) *achieves the bound* $\text{ControlReg}_T(\text{alg}; \Pi_\star) \leq \log^4(1+T) \cdot \frac{c_\star^5(1+\|K\|_{\text{op}})^3}{(1-\rho_\star)^5} \cdot d_u d_y R_{\text{nat}}^2 \cdot \frac{L^2}{\alpha}$.

**Theorem 3.2** (Guarantee for Unknown System). *Suppose Assumptions 1, 3 and 4 holds, and for given* $\rho_\star \in (0,1), c_\star > 0$, *let* $\Pi_\star$ *be as in Definition 3.1. For simplicity, also assume* $c_\star \geq c_K, \rho_\star \geq \rho_K$. *In addition, assume* $\nabla^2 \ell_t \preceq LI$ *uniformly. Then, for any* $\delta \in (0, 1/T)$, DRC-ONS *with an initial estimation phase (Algorithm 3) for an appropriate choice of parameters has the following regret with probability* $1 - \delta$: $\text{ControlReg}_T(\text{alg}; \Pi_\star) \lesssim \sqrt{T} \log^3(1+T) \log \frac{1}{\delta} \cdot \frac{c_\star^8(1+\|K\|_{\text{op}})^5}{(1-\rho_\star)^{10}} \cdot d_y(d_u + d_y) R_{\text{nat}}^5 \cdot \frac{L^2}{\alpha}$.

Together, these bounds match the optimal regret bounds for known and unknown control, up to logarithmic factors [4, 26]. The above theorems are proven Appendix E, which also gives complete statements which specify the parameter choices Theorems 3.1a and 3.2a. In addition, Appendix D generalizes the algorithm by replacing *static* $K$ in the DRC algorithm with a dynamic nominal controller $\pi_0$, for which analogous guarantees are stated in Appendix E. Importantly, Appendix E.3 verifies that convolution-invertibility holds:

**Lemma 3.1.** *For* $\kappa$ *as in Definition 2.2, we have* $\kappa(G_K) \geq \frac{1}{4} \min\{1, \|K\|_{\text{op}}^{-2}\}$.

## Broader Impact

Though this paper is primarily theoretical in nature, we believe that the non-stochastic control setting is an important one. Historically, one of the greatest strengths of control theory is its ability to provide robust, mathematical guarantees on performance quality. As control theory merges with recent developments in reinforcement learning, we see novel applications in domains with little room for error: control algorithms in automated transportation, server cooling, and industrial robotics can wreak havoc when gone awry. These tasks may range from easy-to-model to wildly unpredictable, and purely stochastic models may not suffice to capture the full extent of the uncertainty in the task. On the other hand, traditional techniques from robust control may be overly conservative, and deem certain tasks infeasible from the outset.

While far from perfect, we believe that the non-stochastic control model inches us closer towards robustness to modeling assumptions, without succumbing to excessive pessimism. As such, we find it important to understand what, if any, challenges this more accomodating model poses to data-driven control. We hope that our central theoretical contribution - demonstrating that the uncertainty in the noise model is in fact not a significant barrier to achieving near optimal performance - may encourage practioners not to abandon considerations of robustness for fear of sacrificing performance. But there is still a long road ahead, and we recognize that non-stochastic control does not capture many important senses of robustness in the decades-old control literature. We also recognize that there are, and will continue to be, instances when performance *must* be sacrificed for robustness, and hope our work will contribute a small but helpful part in a broader dialogue about the tensions between safety and performance in data-driven control.

## Acknowledgements

MS is generously supported by an Open Philanthropy AI Fellowship. MS also thanks Dylan Foster and Elad Hazan for their helpful discussions.

## Footnotes

[1]The oblivious assumption is only necessary if the dynamics are unknown to the learner; if the dynamics are known, our guarantees hold against adaptive adversaries as well.

[2]Throughout, the initial iterates $(\mathbf{z}_s)_{s \leq 0}$ are arbitrary elements of $\mathcal{C}$. We note that Anava et al. [6] referred to $\mathrm{MemoryReg}_T$ as "policy regret", but this differs slightly from the policy regret proposed by Arora et al. [7]. To avoid confusion, we use "memory regret".

[3]This may be restrictive for partially observed systems [17], see Appendix D for generalizations.

[4]The assumed bound can be stated in terms of $\max_t \|\mathbf{w}_t, \mathbf{e}_t\|_2$. One may allow $R_{\text{nat}}$ to grow logarithmically (e.g. $R_{\text{nat}} = \mathcal{O}(\log^{1/2} T)$ for subguassian noise), by inflating logarithmic factors in the final bounds.

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
