[Supplementary Material]

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

[5]Intuitively, this is because with (semi-stochastic) noise, one can replace the infinite-horizon invertibility condition $\kappa(G)$ of Definition 2.2 with a finite-horizon analogue, $\kappa_{m,h}(G)$. It is shown that this analogue decays at most polynomially in $m, h$, even though $\kappa(G)$ may be zero. This translates into a polynomial dependence on $m, h$ in the final bound, which contributes only logarithmic factors for the typical choice $m, h = \mathcal{O}(\log T)$.

[6]Empirically, one can just verify whether the LS problem is well conditioned

[7] $R_G \geq 1$ by Definition 2.1, and $\kappa \leq 1$ by Definition 2.2

[8]Note the folklore results that the expectation average of $k$ Rademacher random variables scales as $\sqrt{k}$

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

# Contents

# A Organization of the Appendix and Notation

The appendix is organized as follows:

- **Appendix B** provides further discussion, describing how our work serves to characterize the relative difficulty of adversarial noise in online control settings when compared to stochastic.

- **Appendix C** provides an in-depth comparison with the classic LQR and LQG settings, together with an in-depth discussion in Appendix B.1 about the extent to which stochasticity affects the optimal regret rates in online control.

- **Appendix D** provides the full statement of the algorithm DRC-ONS algorithm for the known and unknown settings, and describes the more general DRC-ONS-DYN algorithm for use with a non-static internal controller.

- **Appendix E** provides full statements and proofs of our main regret bounds for the control setting, Theorems 3.1 and 3.2. In particular, we provide the full analogues with the full parameter settings required for the regret bounds, Theorems 3.2b and G.1. We also provide generalizations of our DRC-ONS-DYN algorithm , Theorems 3.1b and 3.2b.

- **Appendix F** gives the full proof of the logarithmic regret bound for Semi-ONS, Theorem 2.1, and Appendix H provides the omitted proofs.

- **Appendix G** gives the full proof of the quadratic error sensitivity of Semi-ONS, Theorem 2.2, and Appendix I provides the omitted proofs.

- **Appendix J** gives the proof of Theorem 2.3 , and then demonstrates the standard online Newton step matches the tradeoff (Theorem J.1)

**Notation:** We use $a = \mathcal{O}(b)$ and $a \lesssim b$ interchangably to denote that $a \leq Cb$, where $C$ is a universal constant independent of problem parameters. We also use $a \vee b$ to denote $\max\{a, b\}$, and $a \wedge b$ to denote $\min\{a, b\}$. Notation relevant to the control problem is reviewed where-necessary in Appendices C and D. In what follows, we review notation relevant to the generic analyses of Semi-ONS.

In Semi-ONS, we have the with-memory loss functions

$$F_t(z_t, \ldots, z_{t-h}) := \ell_t(\mathbf{v}_t + \sum_{i=0}^{h} G^{[i]} \mathbf{Y}_{t-i} z_{t-i}),$$

and their unary specializations

$$f_t(z) := F_t(z, \ldots, z) = \ell_t(\mathbf{v}_t + \mathbf{H}_t z), \quad \mathbf{H}_t := \sum_{i=0}^{h} G^{[i]} \mathbf{Y}_{t-i}.$$

Here the losses $\ell_t, \mathbf{v}_t, \mathbf{Y}_t$ change at each round, and $G = (G^{[i]})_{i \geq 0}$ is regarded as part of an infinite-length Markov operator which is fixed throughout.

For unknown systems, we are use approximate losses, where $\widehat{\mathbf{v}}_t \approx \mathbf{v}_t$, $\widehat{G} \approx G$,

$$\widehat{f}_t(z) := \widehat{F}_t(z, \ldots, z) = \ell_t(\widehat{\mathbf{v}}_t + \widehat{\mathbf{H}}_t z), \quad \widehat{\mathbf{H}}_t := \sum_{i=0}^{h} \widehat{G}^{[i]} \mathbf{Y}_{t-i}.$$

Throughout, we use bold $\mathbf{z}_t$ to refer to the iterates of the algorithm.

# B Further Discussion

## B.1 Discussion of Results

In this work, we demonstrate that fast rates for online control, and in particular, the optimal $\sqrt{T}$ regret rate [26] for the online LQR setting, are achievable with non-stochastic noise. Interestingly, simultaneous work by Lale et al. [22] shows that the presence of observation noise implies that the

optimal regret for purely stochastic LQG is in fact *polylogarithmic*. At first this seems puzzling because, on face, LQG appears to be a strict generalization of LQR. However, $\text{poly}(\log T)$ regret occurs when LQG has a strictly non-degenerate stochastic observation noise $\mathbf{e}_t$, which is *not* the case in LQR. This faster rate is achievable because the noise on the observation provides continuous exploration, allowing the learner to continue to learn with dynamics while simultanously exploiting near-optimal policies. Alternatively, this observation noise can be understood as making the baseline comparator *easier* (i.e. $\min_{\pi \in \Pi} K_T(\pi)$ is larger), because the underlying control problem is more difficult.

Since we are not guaranteed this observation noise in purely non-stochastic control (indeed, there may be no observation noise at all), $\sqrt{T}$ is still the optimal rate in our setting. Thus, our regret guarantees contribute to the following surprising characterization of regret (with respect to linear dyanic policies) in linear control:

- For known system dynamics, non-stochastic control is just easy as stochastic (Theorem 3.1). There is no substantial price to pay for past mistakes, even under potentially unpredictable, non-stochastic disturbances.

- For unknown system dynamics, stochastic process noise confers little advantage over adversarial noise; both have quadratic sensitivity to error (Theorem 3.2).

- However, there *is* an advantage to having non-degenerate observation noise. But this is due to continual *exploration* induced by stochastic noise, and *not* because stochastic reduces sensitivity to error.

As mentioned in the introduction, competing with *arbitrary* policies (e.g. the optimal control law given the noise) requires regret which is linear in $T$ [24]. Understanding the optimal competive ratio, or further assumptions which allow sublinear regret with respect to the optimal control law, remain an interesting direction for future work.

## B.2 Conclusion

In this work, we demonstrate that fast rates for online control, and in particular, the optimal $\sqrt{T}$ regret rate [26] for the online LQR setting, are achievable with non-stochastic noise.

**Future Work**    It is an interesting direction for future research to determine if non-degenerate observation noise can be used to attain polylogarithmic regret for unknown systems in the *semi-stochastic* regime considered by Simchowitz et al. [28]. This regime interpolates between purely stochastic non-degenerate noise, and arbitrary adversarial noise considered in this setting.

Furthermore, it may be possible that $\sqrt{T}$ regret for unknown systems is attainable even *without* strongly convex cost function; currently, the state of the art in this setting is $T^{2/3}$ [28, 20].

Finally, we hope future work will take up a more ambitious direction of inquiry, investigating whether these techniques can be applied beyond linear time invariant systems with bound noise. Such directions understanding slowly-varying dynamics, robustness to non-linearities, and model-predictive control.

**Open Question: System Stability and Fast Rates**    Lastly, an open question that remains is the extent to which stability of the dynamics affects the extent to which stochastic control is easier than non-stochastic. For example, the guarantees in Lale et al. [22] assume that the dynamics of the system are internally stable, which presumbaly simplifies the system identification procedure. On the other hand, our work assumes only that our system can be stabilized by a static feedback controller, which holds without loss of generality for fully observed systems.

As discussed in Appendix D, there are many partially observed systems which cannot be stabilized even by static feedback, but can be stabilized by more general linear control laws. For such systems, our guarantees do extend, but under the opaque technical assumption on the dynamics induced by this more general stabilizing controller have the invertibility property of Definition 2.2. Recall that for the simple case of static feedback, this invertible property is proven to hold in Lemma 3.1.

On the other hand, Simchowitz et al. [28] show that for semi-stochastic disturbances (disturbances with a non-degenerate stochastic component), one can still achieve fast rates for *any* any linear sta-

bilizing scheme.[5] This seems to suggest that for controller parametrizations based on more powerful stabilizing controllers, stochasticity may in fact be beneficial. It is an interesting direction for future work to understand whether these more general stabilizing controllers admit fast regret rates for non-stochastic control.

# Part I

# Appendices for Control

## C  Past Work and Classical Settings

In this section, we describe in detail how our non-stochastic control setting compares with other control settings considered in the literature. At the end of the section, we conclude with a more thorough discussion of the separations (and lack thereof) between stochastic and non-stochastic control. Recall that our linear system is described by the dynamic equations

$$\mathbf{x}_{t+1} = A_\star \mathbf{x}_t + B_\star \mathbf{u}_t + \mathbf{w}_t, \quad \mathbf{y}_t = C_\star \mathbf{x}_t + \mathbf{e}_t, \tag{C.1}$$

Of special interest are the *fully observed* settings, where $\mathbf{y}_t = \mathbf{x}_t$. We may also imagine an intermediate, *full-rank observation* setting, where $d_y = d_x$, and $\sigma_{\min}(C_\star) > 0$. Note that this latter setting allows for observation noise $\mathbf{e}_t$, while the former does not. Finally, in full generality $C_\star \in \mathbb{R}^{d_y d_x}$ may have rank $\mathrm{rank}(C_\star) < d_x$, and thus states cannot in general be recovered from observations.

### C.1  Online LQR

The linear quadratic regularity, or LQR, corresponds to the setting where the state is fully observed $\mathbf{x}_t = \mathbf{y}_t$, and the noise $\mathbf{w}_t$ is selected from a mean-zero, light-tailed stochastic process - typically i.i.d. Gaussian. Crucially, the noise $\mathbf{w}_t$ is assumed to have some non-degenerate covariance: e.g., $\mathbf{w}_t \overset{\text{i.i.d}}{\sim} \mathcal{N}(0, \Sigma)$ for some $\Sigma \succ 0$. One then considers quadratic cost functions which do not vary with time:

$$\ell_t(x, u) = \ell(x, u) = x^\top R x + u^\top Q u,$$

where $R$ and $Q$ are positive definite matrices. In particular, $\ell(x, u)$ is a strong-convex function, and thus the LQR setting is subsumed by our present work.

For the above setting, the optimal control policy (in the limit as $T \to \infty$) is described by a static feedback law $\mathbf{u}_t = K_\star \mathbf{x}_t$, where $K_\star$ solves the Discrete Algebraic Riccati Euqation, or DARE; we denote the corresponding control policy $\pi^{K_\star}$. Note that this is in fact the optimal *unrestricted* control policy (say, over any policy which executes inputs as functions of present and past observations), despite having the simple static feedback form.

Results for online LQR consider a regret benchmark typically considered performance with respect to this benchmark (see e.g. [1, 13, 25, 12])

$$\overline{R}_T(\mathsf{alg}) := J_T(\mathsf{alg}) - T \lim_{n \to \infty} \frac{1}{n} \mathbb{E}_\mathbf{w}[J_n(\pi^{K_\star})]$$

where the righthand term is the *infinite horizon* average cost induced by placing the optimal control law $K_\star$. One can show (e.g. [26]) $\mathbb{E}_\mathbf{w}[J_n(\pi^{K_\star})]$ is increasing in $n$. Thus, by Jensen's inequality, it

holds that for any $\Pi \subset \Pi_{\mathrm{ldc}}$ containing $\pi^{K_\star}$,

$$\mathbb{E}_{\mathbf{w}}[\overline{R}_T(\mathsf{alg})] = \mathbb{E}_{\mathbf{w}}[J_T(\mathsf{alg})] - T \lim_{n \to \infty} \frac{1}{n} \mathbb{E}_{\mathbf{w}}[J_n(\pi^{K_\star})]$$
$$\leq J_T(\mathsf{alg}) - \mathbb{E}_{\mathbf{w}}[J_T(\pi^{K_\star})]$$
$$= \mathbb{E}_{\mathbf{w}}[J_T(\mathsf{alg})] - \inf_{\pi \in \Pi} \mathbb{E}_{\mathbf{w}}[J_T(\pi)]$$
$$\leq \mathbb{E}_{\mathbf{w}}[J_T(\mathsf{alg})] - \mathbb{E}_{\mathbf{w}} \inf_{\pi \in \Pi} J_T(\pi)$$
$$\leq \mathbb{E}_{\mathbf{w}}[J_T(\mathsf{alg}) - \inf_{\pi \in \Pi} J_T(\pi)] := \mathbb{E}_{\mathbf{w}}[\mathrm{Regret}_T(\mathsf{alg}; \Pi)],$$

where $\mathrm{Regret}_T$ is our non-stochastic benchmark. Hence, we find that, in expectation, the standard benchmark for online LQR is weaker than ours. Nevertheless, the two benchmark typically concide up to lower order terms due to martingale concentration. Observe however a key conceptual difference: the LQR regret $\overline{R}_T$ can be defined with an *a prior* benchmark, because the dynamics are stochastic. On the other hand, the non-stochastic benchmark is defined *a posteriori*, after because the noises are selected by an adversary.

### C.2  Online LQG

In the LQG, or linear quadratic gaussian control, one typically assumes a partially observed dynamical system, inheriting the full generality of Eq. (C.1). Again, the cost function is typically taken to be quadratic function of input and output:

$$\ell_t(y, u) = \ell(y, u) = y^\top R y + y^\top Q y,$$

Again, $R, Q$ are assumed to be positive defined, and thus our assumption that $\ell_t$ are strongly convex subsumes the LQG setting. Typically, online LQG assumes that both the process noise $\mathbf{w}_t$ *and the observation noise* $\mathbf{e}_t$ are not only mean zero and stochastic, but also well conditioned. For example, $\mathbf{w}_t \overset{\mathrm{i.i.d}}{\sim} \mathcal{N}(0, \Sigma_w)$ and $\mathbf{e}_t \overset{\mathrm{i.i.d}}{\sim} \mathcal{N}(0, \Sigma_e)$, where $\Sigma_w, \Sigma_e \succ 0$.

Whereas the unconstrained optimal policy in LQR is an *static* feedback law, the optimal LQG policy is *dynamic* linear controller of the form considered in this work. This is true even if $C_\star = I$ but there is non-zero process noise $\mathbf{e}_t$; that is, $\mathbf{y}_t = \mathbf{x}_t + \mathbf{e}_t$.

## D  Pseudocode, and Dynamic Feedback Generalization

### D.1  Full Pseudocode for Static Feedback Parametrization

---
**parameters**:
  Newton parameters $\eta, \lambda > 0$
  DRC parameters radius $R_\mathcal{M} > 0$, DRC length $m \geq 1$, memory length $h \geq 0$
  closed-loop Markov operator estimate $\widehat{G}$.   % if known system, set $\widehat{G} \leftarrow G_K$
**initialize:**
  Constraint set $\mathcal{M} \leftarrow \mathcal{M}_{\mathrm{drc}}(h, R_\mathcal{M})$ (Eq. (3.2))
  Semi-ONS subroutine $\mathcal{A} \leftarrow \text{Semi-ONS}(\eta, \lambda, \mathfrak{c}(\mathcal{M}))$ (Algorithm 1)
  initial values $\widehat{\mathbf{y}}_0^K, \widehat{\mathbf{y}}_{-1}^K, \ldots, \widehat{\mathbf{y}}_{-(m+h)}^K \leftarrow 0$ **for** $t = 1, 2, \ldots$: **do**
    **recieve** $\mathbf{y}_t^{\mathsf{alg}}$ from environment, iterate $\mathbf{z}_t$ from $\mathcal{A}$, and set DRC parameter $\mathbf{M}_t \leftarrow \mathfrak{c}^{-1}[\mathbf{z}_t]$.
    Construct estimate $\widehat{\mathbf{v}}_t^K = (\widehat{\mathbf{y}}_t^K, \widehat{\mathbf{u}}_t^K)$ via Eq. (3.3)
    **play** input $\mathbf{u}_t^{\mathsf{alg}} \leftarrow K \mathbf{y}_t^{\mathsf{alg}} + \mathbf{u}_t^{\mathsf{ex}}(\mathbf{M}_t \mid \widehat{\mathbf{y}}_{1:t}^K)$.
    **suffer** loss $\ell_t(\mathbf{y}_t^{\mathsf{alg}}, \mathbf{u}_t^{\mathsf{alg}})$, and observe $\ell_t(\cdot)$ .
    **feed** $\mathcal{A}$ the pair $(\ell_t, \widehat{\mathbf{H}}_t, \widehat{\mathbf{v}}_t^K)$, defined in Eq. (2.3), and update $\mathcal{A}$.
---
**Algorithm 2:** Disturbance Response Control via Online Newton Step (DRC-ONS).

### D.2  Stabilizing with dynamic feedback

In general, a partially observed system can not be able to be stabilized by static feedback. To circumvent this, we describe stabilizing the system with an *dynamic feedback controller*, a parameterization

**Input:**

Newton parameters $\eta, \lambda > 0$

DRC parameters radius $R_\mathcal{M} > 0$, DRC length $m \geq 1$, memory length $h \geq 0$

Estimation Length $N \geq 0$     % $N \propto \sqrt{T}$

**Initialize** $\widehat{G}^{[0]} = \begin{bmatrix} 0_{d_u \times d_y} \\ I_{d_u} \end{bmatrix}$, and $\widehat{G}^{[i]} = 0$ for $i > h$.

**for** $t = 1, 2, \ldots, N$ **do**

    **receive** $\mathbf{y}_t^{\mathsf{alg}}$

    **play** $\mathbf{u}_t^{\mathsf{alg}} = \mathbf{u}_t^{\mathsf{ex,alg}} + K\mathbf{y}_t^{\mathsf{alg}}$, where $\mathbf{u}_t^{\mathsf{ex,alg}} \overset{\text{i.i.d}}{\sim} \mathcal{N}(0, I_{d_u})$.

**estimate** $\widehat{G}^{[1:h]} = (\widehat{G}^{[i]})_{i \in [h]} \leftarrow \arg\min_{G^{[1:h]}} \sum_{t=h+1}^{N} \|\mathbf{v}_t^{\mathsf{alg}} - \sum_{i=1}^{h} G^{[i]} \mathbf{u}_{t-i}^{\mathsf{ex,alg}}\|_2^2$.

**run** Algorithm 2 for times $t = N+1, N+2, \ldots, T$, using $\widehat{G}$ as the Markov parameter estimate, and parameters $m, h, \lambda, \eta$.

**Algorithm 3:** Full DRC-ONS for Unknown System, with estimation

we refer to as DRC-DYN. The following exposition mirrors Simchowitz et al. [28], but is abridged considerably. Specifically, we assume that our algorithm maintains an internal state $\mathbf{s}_t^{\mathsf{alg}}$, which evolves according to the dynamical equations

$$\mathbf{s}_{t+1}^{\mathsf{alg}} = A_{\pi_0}\mathbf{s}_t^{\mathsf{alg}} + B_{\pi_0}\mathbf{y}_t^{\mathsf{alg}} + B_{\pi_0,u}\mathbf{u}_t^{\mathsf{ex}}, \tag{D.1}$$

and selects inputs as a combination of an exogenous input $\mathbf{u}_t^{\mathsf{ex}}$, and an endogenous input determined by the system:

$$\mathbf{u}_t^{\mathsf{alg}} = \mathbf{u}_t^{\mathsf{ex,alg}} + (C_{\pi_0}\mathbf{s}_t^{\mathsf{alg}} + D_{\pi_0}\mathbf{y}_t^{\mathsf{alg}}). \tag{D.2}$$

Lastly, the algorithmic prescribes an *control output*, denoted by $\boldsymbol{\omega}_t$, given by

$$\boldsymbol{\omega}_{t+1}^{\mathsf{alg}} = C_{\pi_0,\omega}\mathbf{s}_t^{\mathsf{alg}} + D_{\pi_0,\omega}\mathbf{y}_t^{\mathsf{alg}} \in \mathbb{R}^{d_\omega},$$

which we use to parameterize the controller. In the special case of static feedback, we take $C_{\pi_0,\omega} = 0$ and $D_{\pi_0,\omega} = I$, so that $\boldsymbol{\omega}_t^{\mathsf{alg}} = \mathbf{y}_t^{\mathsf{alg}}$. We assume that $\pi_0$ is *stabilizing*, meaning that, if we have $\max_t \|\mathbf{e}_t\|, \|\mathbf{w}_t\|, \|\mathbf{u}_t^{\mathsf{ex,alg}}\| < \infty$ are bounded, then with $\max_t \|\mathbf{u}_t^{\mathsf{alg}}\|, \|\mathbf{y}_t^{\mathsf{alg}}\|, \|\boldsymbol{\omega}_t^{\mathsf{alg}}\| < \infty$. As a consequence of the Youla parametrization [30], one can always construct a controller $\pi_0$ which has this property for sufficiently non-pathological systems.

Analogous to the sequence $\mathbf{y}_t^K, \mathbf{u}_t^K$, we consider a sequence that arises under no exogenous inputs:

**Definition D.1.** We define the 'Nature' sequence $\mathbf{y}_t^{\mathsf{nat}}, \mathbf{u}_t^{\mathsf{nat}}, \boldsymbol{\omega}_t^{\mathsf{nat}}$ as the sequence obtained by executing the stabilizing policy $\pi_0$ in the absence of $\mathbf{u}_t^{\mathsf{ex}} = 0$; we see $\mathbf{v}_t^{\mathsf{nat}} = (\mathbf{y}_t^{\mathsf{nat}}, \mathbf{u}_t^{\mathsf{nat}}) \in \mathbb{R}^{d_y + d_u}$. Each such sequence is determined uniquely by the disturbances $\mathbf{w}_t, \mathbf{e}_t$.

Moreover, the 'Nature' sequences can be related to the sequences visited by the algorithm via linear Markov operators

**Definition D.2.** We define the linear Markov operators $G_{\mathsf{ex} \to v}, G_{\mathsf{ex} \to \omega}$ as the operators for which

$$\boldsymbol{\omega}_t^{\mathsf{alg}} = \boldsymbol{\omega}_t^{\mathsf{nat}} + \sum_{i=1}^{t} G_{\mathsf{ex} \to \omega}^{[t-i]} \mathbf{u}_i^{\mathsf{ex}}, \quad \mathbf{v}_t^{\mathsf{alg}} = \mathbf{v}_t^{\mathsf{nat}} + \sum_{i=1}^{t} G_{\mathsf{ex} \to v}^{[t-i]} \mathbf{u}_i^{\mathsf{alg}}.$$

We note that $G_{\mathsf{ex} \to \omega}^{[0]} = 0_{d_\omega \times d_u}$ by construction.

Finally, we describe our controller parametrization:

**Definition D.3** (DRC with dynamic stabilizing controller). Generalizing Eq. (3.2), let $\mathcal{M}_{\mathrm{drc}}(m, R_\mathcal{M})$ denote $M \in \mathcal{G}^{d_u \times d_\omega}$ for which $\|M\|_{\ell_1,\mathrm{op}} \leq R_\mathcal{M}$, and $M^{[i]} = 0$ for all $i \geq m$. Given estimates $\widehat{\boldsymbol{\omega}}_{t-m+1}^{\mathsf{nat}}, \ldots, \widehat{\boldsymbol{\omega}}_t^{\mathsf{nat}}$, we select

$$\mathbf{u}_t^{\mathsf{ex}}(M \mid \widehat{\boldsymbol{\omega}}_{1:t}^{\mathsf{nat}}) := \sum_{i=0}^{m-1} M^{[i]} \widehat{\boldsymbol{\omega}}_{t-1}^{\mathsf{nat}}$$

We recover the static feedback setting in the following example:

**Example D.1** (Static Feedback). To recover the special case of static feedback, we make the following substitutions

- We set $\mathbf{s}_t^{\mathsf{alg}} = 0$ for all $t$, $C_{\pi_0} = 0$ and $D_{\pi_0} = K$.

- We set $C_{\pi_0,\omega} = 0$ and $D_{\pi_0,\omega} = I$, so that $\mathbf{u}_t^{\mathsf{alg}} = \mathbf{u}_t^{\mathsf{ex,alg}} + K\mathbf{y}_t^{\mathsf{alg}}$

- We set we set $C_{\pi_0,\omega} = 0$ and $D_{\pi_0,\omega} = I$, so that $\boldsymbol{\omega}_t^{\mathsf{alg}} = \mathbf{y}_t^{\mathsf{alg}}$ for all $t$.

- The quantities $\mathbf{y}_t^{\mathsf{nat}}$ and $\boldsymbol{\omega}_t^{\mathsf{nat}}$ both correspond to $\mathbf{y}_t^K$, and $\mathbf{u}_t^{\mathsf{nat}} = \mathbf{u}_t^K$, the operator $G_{\mathrm{ex}\to v}$ becomes the Markov operator $G_K$, and $G_{\mathrm{ex}\to\omega}$ becomes the top $d_y \times d_u$ block of $G_K$, capturing the response from $\mathbf{u}_t^{\mathrm{ex}} \to \mathbf{y}_t$.

- Thus, $\mathbf{u}_t^{\mathrm{ex}}(M \mid \widehat{\boldsymbol{\omega}}_{1:t}^{\mathsf{nat}})$ corresponds to $\mathbf{u}_t^{\mathrm{ex}}(M \mid \widehat{\mathbf{y}}_{1:t}^K)$.

### D.3   Full Algorithm under Dynamic Feedback

Let us now turn to the specific of the main algorithm with dynamic feedback, DRC-ONS-DYN. Throughout the algorithm, we maintain an internal state updated according to the nominal controller $\pi_0$ via Eq. (D.1). Moreover, all inputs are selected as $\mathbf{u}_t^{\mathsf{alg}} = \mathbf{u}_t^{\mathsf{ex,alg}} + (C_{\pi_0}\mathbf{s}_t^{\mathsf{alg}} + D_{\pi_0}\mathbf{y}_t^{\mathsf{alg}})$ in accordance with Eq. (D.2).

Next, we specify how we recover $\mathbf{v}_t^{\mathsf{alg}}$ and $\boldsymbol{\omega}_t^{\mathsf{alg}}$. Given estimates $\widehat{G}_{\mathrm{ex}\to(y,u)}, \widehat{G}_{\mathrm{ex}\to\omega}$, we parallel Eq. (3.3) in defining

$$\widehat{\mathbf{u}}_t^{\mathsf{nat}} := \begin{bmatrix} \widehat{\mathbf{y}}_t^{\mathsf{nat}} \\ \widehat{\mathbf{u}}_t^{\mathsf{nat}} \end{bmatrix} = \begin{bmatrix} \mathbf{y}_t^{\mathsf{alg}} \\ C_{\pi_0}\mathbf{s}_t^{\mathsf{alg}} + D_{\pi_0}\mathbf{y}_t^{\mathsf{alg}} \end{bmatrix} - \sum_{i=1}^{t-1} \widehat{G}_{\mathrm{ex}\to(y,u)}^{[i]} \mathbf{u}_{t-i}^{\mathsf{ex,alg}},$$

$$\widehat{\boldsymbol{\omega}}_t^{\mathsf{nat}} := \boldsymbol{\omega}_t^{\mathsf{alg}} - \sum_{i=1}^{t-1} \widehat{G}_{\mathrm{ex}\to\omega}^{[i]} \mathbf{u}_{t-i}^{\mathsf{ex,alg}}. \tag{D.3}$$

As in the static feedback case, the above exactly $\mathbf{v}_t^{\mathsf{nat}}, \boldsymbol{\omega}_t^{\mathsf{nat}}$ for exact estimates $\widehat{G}_{\mathrm{ex}\to(y,u)} = G_{\mathrm{ex}\to v}$ and $\widehat{G}_{\mathrm{ex}\to\omega} = G_{\mathrm{ex}\to\omega}$. We then contruct optimization losses as follows, mirroring Eq. (2.3):

$$\widehat{f}_t(z) := \ell_t(\widehat{\mathbf{v}}_t^K + \widehat{\mathbf{H}}_t z), \text{ where } \widehat{\mathbf{H}}_t := \sum_{i=0}^h \widehat{G}_{\mathrm{ex}\to(y,u)}^{[i]} \mathbf{Y}_{t-i}, \text{ and } \mathbf{Y}_s = \mathfrak{e}_\omega[\widehat{\boldsymbol{\omega}}_{s:s-m}^{\mathsf{nat}}], \tag{D.4}$$

where $\mathfrak{e}_\omega$ is an embedding map analogues to $\mathfrak{e}_y$.

With these estimates and definitions, Algorithms 4 and 5 provides the pseudocode generalizing Algorithms 2 and 3 to our setting. The main differences are

- Using $\widehat{\boldsymbol{\omega}}_t^{\mathsf{nat}}$ for the controller parameterization, rather than $\mathbf{y}_t^K$.
- Mainting the internal state $\mathbf{s}_t^{\mathsf{alg}}$
- Estimating two sets of Markov parameters, $\widehat{G}_{\mathrm{ex}\to\omega}$ and $\widehat{G}_{\mathrm{ex}\to(y,u)}$.

## E   Full Control Regret Bounds and Proofs

This section states and proves our main results for the control setting. We state and prove Theorems 3.1b and 3.2b for the general, dynamic-internal controllers described in Appendix D. We then derive the regret bounds Theorems 3.1 and 3.2 in the main text as consequences of the above theorems. In addition, we state variations of the main-text bounds which make explicit the parameter settings which attain the desired regret (Theorems 3.1a and 3.2a). The section is organized as follows:

- Appendix E.1 gives the requisite assumptions and conditions for the general setup of Appendix D, which replaces the static $K$ controller with dynamics internal controller.

**parameters**: Newton parameters $\eta, \lambda$, radius $R_{\mathcal{M}}$, DRC length $m$, memory $h$, closed-loop Markov operator estimate $\widehat{G}_{\mathrm{ex}\to\omega}, \widehat{G}_{\mathrm{ex}\to(y,u)}$, initial internal state $\mathbf{s}_1^{\mathrm{alg}}$

**initialize:**

    constraint set $\mathcal{M} \leftarrow \mathcal{M}_{\mathrm{drc}}(h, R_{\mathcal{M}})$ (Eq. (3.2)), with $\mathcal{C} \leftarrow \mathfrak{e}(\mathcal{M})$.

    optimization subroutine $\mathcal{A} \leftarrow \text{Semi-Ons}(\eta, \lambda, \mathcal{C})$ (Algorithm 1), with iterates $\mathbf{z}_k$

    initial values $\widehat{\boldsymbol{\omega}}_0^{\mathrm{nat}}, \widehat{\boldsymbol{\omega}}_{-1}^{\mathrm{nat}}, \ldots, \widehat{\boldsymbol{\omega}}_{-(m+h)}^{\mathrm{nat}} \leftarrow 0$

**for** $t = 1, 2, \ldots:$ **do**

    **recieve** $\mathbf{y}_t^{\mathrm{alg}}$ from environment

    Construct estimate $\widehat{\mathbf{u}}_t^{\mathrm{nat}} = (\widehat{\mathbf{y}}_t^{\mathrm{nat}}, \widehat{\mathbf{u}}_t^{\mathrm{nat}})$ and $\widehat{\boldsymbol{\omega}}_t^{\mathrm{nat}}$ via Eq. (D.3)

    Recieve iterate $\mathbf{z}_t$ from $\mathcal{A}$, and back out DRC parameter $\mathbf{M}_t \leftarrow \mathfrak{e}^{-1}[\mathbf{z}_t]$.

    **play** input $\mathbf{u}_t^{\mathrm{alg}} \leftarrow D_{\pi_0}\mathbf{y}_t^{\mathrm{alg}} + C_{\pi_0}\mathbf{s}_t^{\mathrm{alg}} + \mathbf{u}_t^{\mathrm{ex}}(\mathbf{M}_t \mid \widehat{\boldsymbol{\omega}}_{1:t}^{\mathrm{nat}})$. .

    **suffer** loss $\ell_t(\mathbf{y}_t^{\mathrm{alg}}, \mathbf{u}_t^{\mathrm{alg}})$, and observe $\ell_t(\cdot)$

    **feed** $\mathcal{A}$ the pair $(\widehat{f}_t, \widehat{\mathbf{H}}_t)$, defined in Eq. (D.4), and update $\mathcal{A}$

    **update** internal state $\mathbf{s}_{t+1}^{\mathrm{alg}}$ according to Eq. (D.1).

**Algorithm 4:** DRC-ONS-DYN from Markov Parameter Estimates

---

**Input:** Number of samples $N$, system length $h$, DRC length $m$, learning parameters $\eta, \lambda$.

**Initialize** $\widehat{G}_{\mathrm{ex}\to(y,u)}^{[0]} = \begin{bmatrix} 0_{d_u \times d_y} \\ I_{d_u} \end{bmatrix}$, and $\widehat{G}_{\mathrm{ex}\to(y,u)}^{[i]} = 0$ for $i > h$, and $\widehat{G}_{\mathrm{ex}\to\omega}^{[i]} = 0$ for $i = 0$ and for

$i > h$, $\mathbf{s}_1^{\mathrm{alg}} = 0$ **for** $t = 1, 2, \ldots, N$ **do**

    **draw** $\mathbf{u}_t^{\mathrm{ex,alg}} \sim \mathcal{N}(0, I_{d_u})$

    **receive** $\mathbf{v}_t^{\mathrm{alg}} = (\mathbf{y}_t^{\mathrm{alg}}, \mathbf{u}_t^{\mathrm{alg}})$ and $\boldsymbol{\omega}_t^{\mathrm{alg}}$.

    **play** $\mathbf{u}_t^{\mathrm{alg}} = \mathbf{u}_t^{\mathrm{ex,alg}} + (C_{\pi_0}\mathbf{s}_t^{\mathrm{alg}} + D_{\pi_0}\mathbf{y}_t^{\mathrm{alg}})$

    **update** internal state $\mathbf{s}_{t+1}^{\mathrm{alg}}$ according to Eq. (D.1).

**estimate** $\widehat{G}^{[1:h]}$ via

$$\widehat{G}_{\mathrm{ex}\to(y,u)}^{[1:h]} \leftarrow \underset{G^{[1:h]}}{\arg\min} \sum_{t=h+1}^{N} \|\mathbf{v}_t^{\mathrm{alg}} - \sum_{i=1}^{h} G^{[i]}\mathbf{u}_{t-i}^{\mathrm{ex,alg}}\|_2^2$$

$$\widehat{G}_{\mathrm{ex}\to\omega}^{[1:h]} \leftarrow \underset{G^{[1:h]}}{\arg\min} \sum_{t=h+1}^{N} \|\boldsymbol{\omega}_t^{\mathrm{alg}} - \sum_{i=1}^{h} G^{[i]}\mathbf{u}_{t-i}^{\mathrm{ex,alg}}\|_2^2$$

**run** Algorithm 4 for times $t = N+1, N+2, \ldots, T$, using $\widehat{G}_{\mathrm{ex}\to\omega}, \widehat{G}_{\mathrm{ex}\to(y,u)}$ as the Markov parameter estimates, and parameters $m, h, \lambda, \eta$, and state $\mathbf{s}_{t+1}^{\mathrm{alg}}$.

**Algorithm 5:** Full DRC-ONS-DYN for Unknown System (with estimation)

- Appendix E.2 states the general regret guarantees Theorems 3.1b and 3.2b for the dynamic-internal-controller setup. It also states Theorems 3.1a and 3.2a - the complete regret bounds for static feedback with parameter settings made explicit. The static regret bounds are derived in Appendix E.2.1.

- Appendix E.3 proves the bound on the invertibility modulus $\kappa(G_K)$, Lemma 3.1. It also provides discussion regarding the invertibility modulus in the dynamically-stabilized setting (see Remark E.2.

- Appendix E.4 proves the dynamically-stabilized setting guarantee for the known system, Theorem 3.1b. The proof combines the regret decomposition from Simchowitz et al. [28] with our policy regret bound, Theorem 2.1.

- Appendix E.5 proves the dynamically-stabilized setting guarantee for the unknown system, Theorem 3.1b. Again, we combine the existing regret decompositions with the policy regret bound Theorem 2.2.

The arguments that follow essentially reuse lemmas from [28] to port over our policy regret bounds for Semi-ONS to the control setting. We state formal reductions for the known and unknown system settings in Propositions E.5 and E.8, which may be useful in future works applying the DRC parameterization.

The only significant technical difference from [28] is in the analysis of the unknown system, where we use an intermediate step in their handling of one of the approximation errors. This yields an offset in the $\mathbf{Y}_t$-geometry (see Proposition E.8), which is explained further in Appendix E.5.

**Asymptotic Notation:** Throughout, we will use $\mathcal{O}_{\mathrm{cnst}(b)}$ to denote a quantity $a$ which is at most $Cb$, where $C$ is a universal constant independent of problem parameters. Equivalently, $a = \mathcal{O}_{\mathrm{cnst}(b)}$ if and only if $a \lesssim b$. We use both notations interchangably, and $\mathcal{O}_{\mathrm{cnst}(\cdot)}$ affords convenience.

## E.1 Preliminaries and Assumptions for Dynamic Feedback

While the main theorems in the main body of the main text assume explicity geometric decay, the results in this result will be established with a more abstract, yet theoretically more streamlined construction called a *decay function*:

**Definition E.1** (Decay Function). For a Markov operator $G = (G^{[i]})_{i \geq 0}$, we define the decay function as $\psi_G(n) := \sum_{i \geq n} \|G^{[i]}\|_{\mathrm{op}}$. We say that $G$ is *stable* if $\psi_G(0) < \infty$, which implies that $\lim_{n \to \infty} \psi_G(n) = 0$. In general, we say that $\psi$ is a proper, stable decay function if $\psi(n)$ is non-negative, non-increasing, and $\psi(0) < \infty$.

**Assumption 3b** (Stability). We assume that $R_{\pi_0} := \max\{\|G_{\mathrm{ex} \to v}\|_{\ell_1, \mathrm{op}}, \|G_{\mathrm{ex} \to \omega}\|_{\ell_1, \mathrm{op}}\} < \infty$. We further assume that the *decay function* of $G_{\mathrm{ex} \to v}$ and $G_{\mathrm{ex} \to \omega}$ are upper bounded by a proper, stable decay function $\psi_{\pi_0}$. Note that, when the static analogue Assumption 3b holds, we can take

$$R_{\pi_0} = \frac{c_K}{1 - \rho_K}, \quad \psi_{\pi_0}(n) = R_{\pi_0} \rho_K^n.$$

For any stabilizing $\pi_0$, Assumption 3b always holds, and in fact $\psi_{\pi_0}$ will have geometric decay. In the special case of static feedback $K$, Assumption 3 implies that

$$\psi_K(n) \leq \frac{c_K \rho_K^n}{1 - \rho_K}. \tag{E.1}$$

Again, since $\pi_0$ is stabilizing, we also may also assume that the iterates $\mathbf{y}_t^K, \mathbf{e}_t^K$ are bounded for all $t$:

**Assumption 4b** (Bounded Nature's-iterates). We assume that $(\mathbf{w}_t, \mathbf{e}_t)$ are bounded such that, for all $t \geq 1$, $\|\mathbf{v}^{\mathrm{nat}}\|, \|\boldsymbol{\omega}^{\mathrm{nat}}\| \leq R_{\mathrm{nat}}$. This is equivalent to Assumption 4 in when $\pi_0$ corresponds to static feedback $K$.

### E.1.1 Policy Benchmarks

**Definition E.2** (Linear Dynamic Controller). An LDC is specified by a linear dynamical system $(A_\pi, B_\pi, C_\pi, D_\pi)$, with internal state $\mathring{\mathbf{s}}_t^\pi \in \mathbb{R}^{d_\pi}$, equipped with the internal dynamical equations $\mathring{\mathbf{s}}_{t+1}^\pi = A_\pi \mathring{\mathbf{s}}_t^\pi + B_\pi \mathring{\mathbf{y}}_t^\pi$ and $\mathring{\mathbf{u}}_t^\pi := C_\pi \mathring{\mathbf{s}}_t^\pi + D_\pi \mathring{\mathbf{y}}_t^\pi$. We let $\Pi_{\mathrm{ldc}}$ denote the set of all LDC's $\pi$. These policies include static fedback laws $\mathring{\mathbf{u}}_t^\pi = K \mathring{\mathbf{y}}_t^\pi$, but *are considerably more general due to the internal state*. The *closed loop* iterates $(\mathbf{y}_t^\pi, \mathbf{u}_t^\pi, \mathbf{x}_t^\pi, \mathbf{s}_t^\pi)$ denotes the unique sequence consistent with Eq. (1.1), the above internal dynamics, and the equalities $\mathring{\mathbf{u}}_t^\pi = \mathbf{u}_t, \mathring{\mathbf{y}}_t^\pi = \mathbf{y}_t$. The sequence $(\mathbf{y}_t^K, \mathbf{u}_t^K)$ is a special case with $D_\pi = K$ and $C_\pi = 0$.

**Dynamic Policy Benchmark** Lastly, let us quantitatively define our policy benmark, from [28].

**Definition 3.1b** (Policy Benchmark). We define a $\pi_0 \to \pi$ as a Markov operator $G_{\pi_0 \to \pi}$ such that the inputs $\mathbf{u}_t^{\mathrm{ex}, \pi_0 \to \pi} := \sum_{i=1}^t G_{\pi_0 \to \pi}^{[t-i]} \boldsymbol{\omega}_i^{\mathrm{nat}}$ satisfies the following for all $t$:

$$\begin{bmatrix} \mathbf{y}_t^\pi \\ \mathbf{u}_t^\pi \end{bmatrix} = \begin{bmatrix} \mathbf{y}_t^{\mathrm{nat}} \\ \mathbf{u}_t^{\mathrm{nat}} \end{bmatrix} + \sum_{i=1}^t G_{\mathrm{ex} \to v}^{[t-i]} \mathbf{u}_i^{\mathrm{ex}, \pi_0 \to \pi}.$$

where $(\mathbf{y}_t^\pi, \mathbf{u}_t^\pi)$ is the sequence obtained by executed LDC $\pi$. We define the comparator class $\Pi_\star := \Pi_{\mathrm{stab}, \pi_0}(R_\star, \psi_\star)$, where $\Pi_{\mathrm{stab}, \pi_0}(R, \psi) := \{\pi \in \Pi_{\mathrm{ldc}} : \|G_{\pi_0 \to \pi}\|_{\ell_1, \mathrm{op}} \leq R, \psi_{G_{\pi_0 \to \pi}}(n) \leq \psi(n), \forall n\}$.

Exact expressions for conversion operators are detailed in Simchowitz et al. [28, Appendix C].

**Static Policy Benchmark**

**Definition E.3** (Static Feedback Operator). Let $G_{\pi,\mathrm{cl}}$ denote the Markov operator $G_{\pi,\mathrm{cl}}^{[i]} = D_{\pi,\mathrm{cl}}\mathbb{I}_{i=0} + C_{\pi,\mathrm{cl}}A_{\pi,\mathrm{cl}}^{i-1}B_{\pi,\mathrm{cl}}\mathbb{I}_{i>0}$, where we define

$$A_{\pi,\mathrm{cl}} := \begin{bmatrix} A_\star + B_\star D_\pi C_\star & B_\star C_\pi \\ B_\pi C_\star & A_\pi \end{bmatrix}, \quad B_{\pi,\mathrm{cl}} = \begin{bmatrix} B_\star D_\pi & -B_\star \\ B_\pi & 0 \end{bmatrix}$$

$$C_{\pi,\mathrm{cl}} := [(D_\pi - D_{\pi_0})C_\star \quad C_\pi], D_{\pi,\mathrm{cl}} = [D_\pi \quad 0]$$

To specialize to the static-feedback setting described in the main text of the paper, we develop the following concrete expression:

**Lemma E.1** (Conversion operators for static feedback). *Consider the special case of the above, where $\pi_0$ is corresponds to static feedback with matrix $K$. Then, the following is a $K \to \pi$ conversion operator.*

$$G_{K\to\pi}^{[i]} = D_\pi \mathbb{I}_{i=0} + \mathbb{I}_{i>0}C_{\pi,\mathrm{cl}}A_{\pi,\mathrm{cl}}^{i-1}B_{\pi,\mathrm{cl}}\begin{bmatrix} I \\ K \end{bmatrix},$$

*Next, fix $c_\star > 0$, $\rho_\star \in (0,1)$, and recall the set $\Pi_{\mathrm{stab}}(c_\star, \rho_\star) := \{\pi : \forall n, \|G_{\pi,\mathrm{cl}}^{[n]}\|_{\mathrm{op}} \leq c_\star \rho_\star^n\}$. Then defining*

$$\psi_\star(n) := \frac{(1 + \|K\|_{\mathrm{op}})c_\star\rho_\star^n}{1 - \rho_\star}, \quad R_\star := \frac{(1 + \|K\|_{\mathrm{op}})c_\star}{1 - \rho_\star}. \tag{E.2}$$

*we have that $\pi \in \Pi_\star$, where $\Pi_\star = \Pi_{\pi_0,\mathrm{stab}}(R_\star, \psi_\star)$ as defined in Definition 3.1b. Lastly, in the special case where the target policy $\pi$ corresponds to another static feedback law $\mathbf{u}_t = K_\pi \mathbf{y}_t$, then*

$$G_{K\to\pi}^{[i]} = \mathbb{I}_{i=0}K_\pi + (K_\pi - K)C_\star(A_\star + B_\star KC_\star)^{i-1}B_\star(K_\pi - K) \tag{E.3}$$

*Proof.* The first and third statements are a special case of Simchowitz et al. [28, Proposition 1], taking $D_{\pi_0} = K$, and $A_{\pi_0}, B_{\pi_0}, C_{\pi_0}$ identically zero. For the second statement follows from the fact that $\|G_{K\to\pi}^{[i]}\|_{\mathrm{op}} \leq (1 + \|K\|_{\mathrm{op}})\|G_{\pi,\mathrm{cl}}^{[i]}\|_{\mathrm{op}}$. $\square$

## E.2 Complete Statement of Regret Bounds for control setting

Here, we state our main regret bounds for both general dynamical internal controllers (Theorems 3.1b and 3.2b), and specialization for static controllers, Theorems 3.1a and 3.2a. The main theorems in the text Theorems 3.1 and 3.2 are special cases of the latter. Proofs of specialization to static controllers are provided in Appendix E.2.1 below.

**Assumption 5** (Invertibility Modulus). For the setting setting, where the system is stabilized by a possibility non-static nominal controller $\pi_0$, we assuch that the Markov operator $G_{\mathrm{ex}\to v}$ satisfies $\kappa(G_{\mathrm{ex}\to v}) > 0$.

**Remark E.1** (Conditions under which Assumption 5 holds). From Lemma 3.1, we note that Assumption 5 holds whenever $\pi_0$ corresponds to stabilizing the system with a static controller. In general, it is more opaque when Assumption 5 assumption holds. We discuss this in more detail in the Appendix E.3.

With our general setting and notation in place, we are ready to state our general bound. Throughout, we consider a comparator class

$$\Pi_\star := \Pi_{\mathrm{stab},\pi_0}(R_\star, \psi_\star), \text{ where}$$
$$\Pi_{\mathrm{stab},\pi_0}(R, \psi) := \{\pi \in \Pi_{\mathrm{ldc}} : \|G_{\pi_0\to\pi}\|_{\ell_1,\mathrm{op}} \leq R, \psi_{G_{\pi_0\to\pi}}(n) \leq \psi(n), \forall n\},$$

as defined in Definition 3.1b.

**Theorem 3.1b** (Main Regret Guarantee of DRC-ONS-DYN: Known System). *Suppose that 1,3b,4b, 5 hold. Moreover, choose $\lambda = 6hR_{\mathrm{nat}}^2 R_{\pi_0}^2$, $\eta = 1/\alpha$, and suppose that $m, h$ are selected so that that $\psi_{\pi_0}(h+1) \leq R_{\pi_0}/T$, $\psi_\star(m) \leq cR_\star/T$, and $R_\mathcal{M} \geq R_\star$. Then, the DRC-ONS-DYN algorithm (Algorithm 4) enjoys the following regret bound:*

$$\mathrm{ControlReg}_T(\mathsf{alg}; \Pi_\star) \lesssim (\alpha\sqrt{\kappa})^{-1}mh^2 d_u d_\omega R_{\pi_0}^3 R_{\mathrm{nat}}^2 R_\mathcal{M}^2 L^2 \log(1+T),$$

*The above guarantee is also inherited by DRC-ONS (Algorithm 2) as a special case.*

The above theorem is proven in Appendix E.4. For static stabilizing controllers, we obtain the following specialization.

**Theorem 3.1a** (Main Regret Guarantee of DRC-ONS: Known System, with Explicit Parameters). *Suppose Assumptions 1, 3 and 4 holds, and for given $\rho_\star \in (0,1), c_\star > 0$, let $\Pi_\star$ be as in Definition 3.1. Select parameters*

- $h = \lceil \frac{\log T}{1-\rho_K} \rceil$

- $m = \lceil \frac{\log T}{1-\rho_\star} \rceil$

- $R_\mathcal{M} = R_\star = (1 + \|K\|_{\mathrm{op}}) \frac{c_\star}{1-\rho_\star}$

- $\eta = 1/\alpha$, *and* $\lambda = 6hR_{\mathrm{nat}}^2 c_K^2 (1-\rho_K)^2$

*Then,*

$$\mathrm{ControlReg}_T(\mathsf{alg};\Pi_\star) \lesssim \frac{c_K^3 c_\star^2 (1+\|K\|_{\mathrm{op}})^3}{(1-\rho_K)^5 (1-\rho_\star)^3} \cdot d_u d_y R_{\mathrm{nat}}^2 \cdot \frac{L^2}{\alpha} \log^4(1+T)$$

For unknown systems, the following guarantees $\widetilde{\mathcal{O}}\left(\sqrt{T}\right)$ regret:

**Theorem 3.2b** (Main Regret Guarantee of DRC-ONS-DYN: Unknown System). *Suppose that Assumptions 1,3b,4b, 5 hold, and that $\ell_t$ are $L$-smooth ($\nabla^2 \ell_t \preceq L$). Lastly, fix $\delta \in (0, 1/T)$. Then, when the unknown-system variant of DRC-ONS-DYN with estimation (Algorithm 5) is run with the following choice of parameters*

- $\lambda = R_{\mathrm{nat}}^2 \log(1/\delta) \sqrt{T} + hR_{\pi_0}^2$ *and* $\eta = 3/\alpha$

- $N = h^2 \sqrt{T} \max\{d_\omega, d_y + d_u\}$

- $\sqrt{T} \geq 4 \cdot 1764 h^2 R_\mathcal{M}^2 R_{\pi_0}^2 + c_0 h^2 d_u^2$, *where $c_0$ is a universal constant arising from conditioning of the least squraes problem[6].*

- $m \geq m_\star + 2h$ *and* $R_\mathcal{M} \geq 2R_\star$.

- $\psi_{\pi_0}(h+1) \leq R_{\pi_0}/T$, $\psi_\star(m) \leq R_\star/T$

*Then, the following regret bound holds with probability $1 - \delta$:*

$$\mathrm{ControlReg}_T(\mathsf{alg};\Pi_\star) \lesssim \log(1+T) \frac{(d_\omega + d_y)(d_y + d_u)mhL^2 R_{\pi_0}^4 R_{\mathrm{nat}}^5 R_\mathcal{M}^4 \sqrt{T} \log(1/\delta)}{\alpha \kappa^{1/2}}.$$

*The same guarantee also holds for the static analgoue Algorithm 3).*

The following specializes to static control:

**Theorem 3.2a** (Main Regret Guarantee of DRC-ONS: Unknown System, with Explicit Parameters). *Suppose that 1,3b,4b, 5 hold, and that $\ell_t$ are $L$-smooth ($\nabla^2 \ell_t \preceq L$). For simplicity, further select comparator parameters $\rho_\star \geq \rho_K$, $c_\star \geq c_K$. Finally, fix $\delta \in (0, 1/T)$. Then, when the unknown-system variant of DRC-ONS-DYN with estimation (Algorithm 5) is run with the following choice of parameters*

- $h = \lceil (1-\rho_\star)^{-1} \log T \rceil$, $m = 3h$, $R_\mathcal{M} = 2 \frac{(1+\|K\|_{\mathrm{op}})c_\star}{1-\rho_\star}$.

- $\lambda = R_{\mathrm{nat}}^2 \log(1/\delta) \sqrt{T} + hc_K^2/(1-\rho_K)^2$ *and* $\eta = 3/\alpha$

- $N = h^2 \sqrt{T}(d_y + d_u)$

- $\sqrt{T} \geq c \log^2 T((1-\rho_\star)^{-6} c_\star^4 (1+\|K\|_{\mathrm{op}})^2 + (1-\rho_\star)^{-2} d_u^2)$ *for some universal constant $c$ (satisfied for $T = \widetilde{\mathcal{O}}(1)$).*

*Then, the following regret bound holds with probability $1 - \delta$:*

$$\mathrm{ControlReg}_T(\mathsf{alg}; \Pi_\star) \lesssim \sqrt{T} \cdot \frac{c_K^4 c_\star^4 (1 + \|K\|_{\mathrm{op}})^5}{(1 - \rho_K)^4 (1 - \rho_\star)^6} \cdot \frac{L^2 R_{\mathrm{nat}}^5}{\alpha} \cdot \log^3(1 + T) \log(1/\delta)$$

*The same guarantee also holds for the static analgoue Algorithm 3).*

### E.2.1 Specializing Dynamic Stabilizing Controller to Static

*Proof of Theorems 3.1 and 3.1a .* For the static case, as noted in Assumptions 3b and 4b, Assumption 4 implies Assumption 4b, and Assumption 3 implies Assumption 3b with

$$R_{\pi_0} = \frac{c_K}{1 - \rho_K}, \quad \psi_{\pi_0}(n) = R_{\pi_0} \rho_K^n.$$

Moreover, recall that our benchmark is $\pi \in \Pi_{\mathrm{stab}}(c_\star, \rho_\star)$, as defined in Definition 3.1. from Lemma E.1, this benchmark is subsumed by the benchmark $\Pi_\star$ for the choice of $\psi_\star, R_\star$, as in Eq. (E.2):

$$R_\star := \frac{(1 + \|K\|_{\mathrm{op}}) c_\star}{1 - \rho_\star}, \quad \psi_\star(n) \leq R_\star \rho_\star^n.$$

Let us now use the following technical claim:

**Fact E.2.** *Let $\rho \in (0, 1)$. Then $\rho^n \leq 1/T$ for $n \geq \frac{\log T}{1 - \rho}$*

*Proof of Fact E.2.* We have $\rho^n \leq 1/T$ for $n \geq \log(T)/\log(1/\rho)$. But $\log(1/\rho) \leq \frac{1}{\rho} - 1 = \frac{1 - \rho}{\rho}$, so it suffices to select $n \geq \log(T)(\rho/1 - \rho) \geq \log(T)/(1 - \rho)$. $\square$

Thus, our conditions $\psi_{\pi_0}(h + 1) \leq R_{\pi_0}/T$, $\psi_\star(m) \leq c R_\star/T$, and $R_\mathcal{M} \geq R_\star$ hold as soon as

$$h \geq \frac{\log T}{1 - \rho_K}, \quad \geq \frac{\log T}{1 - \rho_\star}.$$

Thus, setting $h = \lceil \frac{\log T}{1 - \rho_K} \rceil$, $m = \lceil \frac{\log T}{1 - \rho_\star} \rceil$, and $R_\mathcal{M} = R_\star = (1 + \|K\|_{\mathrm{op}}) \frac{c_\star}{1 - \rho_\star}$, and $\kappa(G_K) \geq \frac{1}{4} \min\{1, \|K\|_{\mathrm{op}}^{-2}\} \gtrsim (1 + \|K\|_{\mathrm{op}})^{-2}$, we obtain

$$\mathrm{ControlReg}_T(\mathsf{alg}; \Pi_\star) \lesssim \frac{c_K^3 c_\star^2 (1 + \|K\|_{\mathrm{op}})^3}{(1 - \rho_K)^5 (1 - \rho_\star)^3} \cdot d_u d_y R_{\mathrm{nat}}^2 \cdot \frac{L^2}{\alpha} \log^4(1 + T)$$

This requires the step size choice of $\eta = 1/\alpha$ and $\lambda = 6h R_{\mathrm{nat}}^2 c_K^2 (1 - \rho_K)^2$. $\square$

*Theorem 3.2a.* For static feedback, we have $d_\omega = d_y$. Thus, $(d_\omega + d_y)(d_y + d_u) = d_y(d_y + d_u)$. Next, we have $R_{\pi_0}^4 R_\mathcal{M}^4 = (1 - \rho_K)^{-4} c_K^4 \cdot (1 + \|K\|_{\mathrm{op}})^4 (1 - \rho_\star)^{-4} c_\star^4$, and $h \leq m \lesssim (1 - \rho_\star)^{-1} \log(1 + T)$. This gives

$$(d_\omega + d_y)(d_y + d_u) m h R_{\pi_0}^4 R_\mathcal{M}^4 \lesssim d_y(d_y + d_u) \frac{c_K^4 c_\star^4 (1 + \|K\|_{\mathrm{op}})^4}{(1 - \rho_K)^4 (1 - \rho_\star)^6} \log^2(1 + T).$$

Using $1/\sqrt{\kappa} \lesssim (1 + \|K\|_{\mathrm{op}})$, we then get

$$\mathrm{ControlReg}_T(\mathsf{alg}; \Pi_\star) \lesssim \sqrt{T} \cdot \frac{c_K^4 c_\star^4 (1 + \|K\|_{\mathrm{op}})^5}{(1 - \rho_K)^4 (1 - \rho_\star)^6} \cdot \frac{R_{\mathrm{nat}}^5 L^2}{\alpha} \cdot \log^3(1 + T) \log(1/\delta)$$

The correctness of the various parameter settings can e checked analogously. $\square$

### E.3 Invertibility-Modulus and Proof of Lemma 3.1

In this section, we bound the condition-modulus $\kappa(G_K)$ defined in Definition 2.2, and generalize the notion to DRC-DYN parametrizations. To begin, we recall our desired bound:

**Lemma 3.1.** *For $\kappa$ as in Definition 2.2, we have $\kappa(G_K) \geq \frac{1}{4} \min\{1, \|K\|_{\mathrm{op}}^{-2}\}$.*

For general DRC-DYN parameters, the Z-transform yields a clean lower bound for the condition-modulus of $\check{G}_{\mathrm{ex}\to v}$ from Definition 2.2:

**Proposition E.3.** *Define the Z-transform $\check{G}_{\mathrm{ex}\to v} := \mathbb{C} \to \mathbb{C}^{(d_y+d_u)\times d_u}$ as the function*

$$\check{G}_{\mathrm{ex}\to v}(z) = \sum_{i=0}^{\infty} G_{\mathrm{ex}\to v}^{[i]} z^{-i}$$

*Then, we have the lower bound:*

$$\kappa(G_{\mathrm{ex}\to v}) \geq \min_{z\in\mathbb{T}} \sigma_{\min}(\check{G}_{\mathrm{ex}\to v}(z))^2,$$

*where $\kappa(G_{\mathrm{ex}\to v})$ is the condition-modulus of $G_{\mathrm{ex}\to v}$, as defined in Definition 2.2. In particular, if $G_{\mathrm{ex}\to v}$ takes the form*

$$G_{\mathrm{ex}\to v}^{[i]} = \mathbb{I}_{i=0} D_{\mathrm{ex}\to v} + \mathbb{I}_{i>0} C_{\mathrm{ex}\to v} A_{\mathrm{ex}\to v}^{i-1} B_{\mathrm{ex}\to v},$$

*then*

$$\kappa_{\pi_0} \geq \min_{z\in\mathbb{T}} \sigma_{\min}(D_{\mathrm{ex}\to v} + C_{\mathrm{ex}\to v}(zI - A_{\mathrm{ex}\to v})^{-1} B_{\mathrm{ex}\to v})^2.$$

*Proof of Proposition E.3.* Part 2 applies the well-known formula that the Z-transform of an LTI system with operator $G^{[i]} = D\mathbb{I}_{i=0} + CA^{i-1}B\mathbb{I}_{i>0}$, which can be computed via

$$\check{G}(z) = D + C\left(\sum_{i\geq 1} A^{i-1}z^{-i}\right) B$$

$$= D + C\left(z^{-1}\sum_{i\geq 0}(A/z)^i\right) B$$

$$= D + C\left(z^{-1}(I - A/z)^{-1}\right) B$$

$$= D + C\left(zI - A\right))^{-1} B,$$

where we use formal identity identity $\sum_{i\geq 0} X^i = (I - X)^{-1}$.

Let us turn to the first part of the proof. We adopt the argument from [28, Appendix F]. Fix $u_0, u_1, \dots$ with $\sum n = 0^{\infty}\|u_n\|^2 = 1$, and define a Markov-shaped vector $U = (U^{[i]})$, with $U^{[i]}$, and its Z-transform $\check{U}(z) := \sum_{i=0}^{n} U^{[i]} z^{-i}$. We have that

$$\sum_{n\geq 0}\left\|\sum_{i=0}^{n} G^{[i]} u_{n-i}\right\|_2^2 = \sum_{n\geq 0} \|(G*U)^{[n]}\|^2$$

where $*$ denotes the convolution operator. By Parseval's identity, we have that

$$\sum_{n\geq 0}\left\|(G*U)^{[n]}\right\|_2^2 = \frac{1}{2\pi}\int_0^{2\pi} \|\widetilde{(G*U)}(e^{\iota\theta})\|_2^2 \mathrm{d}\theta,$$

where $\widetilde{(G*U)}(z) = \sum_{i\geq 0}(G*U)^{[i]} z^{-i}$ is the Z-transform of $G*U$. Because convolutions become multiplications under the Z-transformation, we have that for the Z-transform of $U$,

$$\frac{1}{2\pi}\int_0^{2\pi} \|\widetilde{(G*U)}(e^{\iota\theta})\|_2^2 \mathrm{d}\theta = \frac{1}{2\pi}\int_0^{2\pi} \|\check{G}(e^{\iota\theta})\check{U}(e^{\iota\theta})\|_2^2 \mathrm{d}\theta.$$

This establishes the first equality of the claim. For the inequality, we have

$$\frac{1}{2\pi}\int_0^{2\pi}\|\check{G}(e^{\iota\theta})\check{U}(e^{\iota\theta})\|_2^2\mathrm{d}\theta \geq \frac{1}{2\pi}\int_0^{2\pi}\sigma_{\min}(\check{G}(e^{\iota\theta}))^2\|\check{U}(e^{\iota\theta})\|_2^2\mathrm{d}\theta$$

$$\geq \min_{z\in\mathbb{T}}\sigma_{\min}(z)^2\cdot\frac{1}{2\pi}\int_0^{2\pi}\|\check{U}(e^{\iota\theta})\|_2^2\mathrm{d}\theta.$$

To conclude, we note that by Parsevals identity, $\frac{1}{2\pi}\int_0^{2\pi}\|\check{U}(e^{\iota\theta})\|_2^2\mathrm{d}\theta. = \sum_{n\geq0}\|U^{[n]}\| = \sum_{n\geq0}\|u_n\|^2 = 1$, giving $\sum_{n\geq0}\left\|\sum_{i=0}^n G^{[i]}u_{n-i}\right\|_2^2 = \frac{1}{2\pi}\int_0^{2\pi}\|\check{G}(e^{\iota\theta})\check{U}(e^{\iota\theta})\|_2^2\mathrm{d}\theta \geq \min_{z\in\mathbb{T}}\sigma_{\min}(z)^2$, as needed.

$\square$

We now turn to giving an explicit lower for the static-feedback stabilized setting:

*Proof of Lemma 3.1.* For the special case of static feedback, we recall from Eq. (3.1) that

$$G_{\mathrm{ex}\to v}^{[i]} = G_K^{[i]} = \mathbb{I}_{i=0}\begin{bmatrix}0\\I\end{bmatrix} + \mathbb{I}_{i>0}\begin{bmatrix}C_\star\\KC_\star\end{bmatrix}(A_\star + B_\star KC_\star)^{i-1}B_\star,\ i\geq 1.$$

Thus, defining $\check{A}(z) := (zI - A_\star + B_\star KC_\star)^{-1}$, we have from Proposition E.3 that

$$\check{G}_{\mathrm{ex}\to v}(z) = \begin{bmatrix}C_\star\check{A}(z)B_\star\\I + KC_\star\check{A}(z)B_\star\end{bmatrix},$$

where the above holds for all $z\in\mathbb{T}$ since $K$ is stabilizing. We now invoke a simple linear algebraic fact:

**Claim E.4** (Lemma F.2 in [28]). *Consider a matrix of the form*

$$W = \begin{bmatrix}YZ\\I + XZ\end{bmatrix}\in\mathbb{R}^{(d_1+d)\times d},$$

*with* $Y\in\mathbb{R}^{d_1\times d_1}$, $X, Z^\top\in\mathbb{R}^{d\times d_1}$. *Then,* $\sigma_{\min}(W)\geq\frac{1}{2}\min\{1,\frac{\sigma_{\min}(Y)}{\|X\|_{\mathrm{op}}}\}$.

Applying the above claim with $Y = I$, $X = K$, and $W = C_\star\check{A}(z)B_\star$, we conclude that $\sigma_{\min}(\check{G}_{\mathrm{ex}\to v}(z)) \geq \frac{1}{2}\min\{1,\|K\|_{\mathrm{op}}^{-1}\}$ for all $z\in\mathbb{C}$. Thus, by Proposition E.3, $\kappa(G_K) \geq (\frac{1}{2}\min\{1,\|K\|_{\mathrm{op}}^{-1}\})^2 = \frac{1}{4}\min\{1,\|K\|_{\mathrm{op}}^{-2}\}$, as needed. $\square$

**Remark E.2** (Generic Bounds on Invertibility). In general, we do not have a generic lower bound on the invertibility modulus which is verifiably no-negative for all choices of stabilizing controllers. For one, it is not clear that our lower bound in Proposition E.3 is sharp, in part because we are working with real operators. However, there are certain conditions (e.g. Youla parametrization, where $A_\star$ has no eigenvalues $z\in\mathbb{T}$, Simchowitz et al. [28, F.2.3]) where we have $\min_{z\in\mathbb{T}}\sigma_{\min}(\check{G}_{\mathrm{ex}\to v}(z))^2$ is strictly positive.

### E.4 Control Proofs for Known System

We focus on the dynamic version of our algorithm, DRC-ONS-DYN, with stabilizing controller $\pi_0$. For known Markov operator, this algorithm specializes to DRC-ONS in the case of static feedback. The following theorem reduces to bounding the policy regret:

**Proposition E.5** (Reduction to policy regret for known dynamics). *Consider the* DRC-ONS-DYN *algorithm (Algorithm 4) initialized with the exact Markov operators* $\widehat{G}_{\mathrm{ex}\to(y,u)} = G_{\mathrm{ex}\to v}, \widehat{G}_{\mathrm{ex}\to\omega} = G_{\mathrm{ex}\to\omega}$, *and iterates* $\mathbf{M}_t$ *produced by an arbitrary black-box optimization procedure* $\mathcal{A}$. *Further, suppose that* $\psi_\star(m)\leq cR_\star/T, \psi_{\pi_0}(h+1)\leq c\psi_{\pi_0}(h+1)/R_{\pi_0}$ *for some* $c > 0$. *Then,*

$$\mathrm{ControlReg}_T(\mathsf{alg}) \leq \mathrm{MemoryReg}_T(\mathsf{alg}) + 12LcR_\mathcal{M}^2R_{\pi_0}^2R_{\mathrm{nat}}^2.$$

*where, for the* $F_t$, $f_t$ *losses in Definition 3.3b, we define*

$$\mathrm{MemoryReg}_T(\mathsf{alg}) := \sum_{t=1}^T F_t(\mathbf{z}_{t:t-h}\mid\boldsymbol{\omega}_{1:t}^{\mathrm{nat}}) - \inf_{z\in\mathcal{M}_\mathfrak{e}}\sum_{t=1}^T f_t(z\mid\boldsymbol{\omega}_{1:t}^{\mathrm{nat}})$$

*The same is true for Algorithm 2 (for static feedback).*

**Remark E.3.** . In the above, we allow a slack parameter $c$ on the choice of $m, h$. This means that our main theorems can be generalized slightly to accomodate when $m, h$ are chosen larger-than-needed.

Next, we bound the relevant parameters required:

**Lemma E.6** (Parameter Bounds). *Assume $R_{\text{nat}}, R_{\mathcal{M}} \geq 1$. The following bounds hold*

    (a) *We have $D = \max\{\|z - z'\| : z, z' \in \mathcal{M}_{\mathfrak{e}}\} \leq 2\sqrt{m}R_{\mathcal{M}}$.*

    (b) *We have $R_Y := \max_t \|\mathbf{Y}_t\|_{\text{op}} = \max_t \|\mathfrak{e}_\omega(\boldsymbol{\omega}_{1:t}^{\text{nat}})\|_{\text{op}} \leq R_{\text{nat}}$.*

    (c) *We have $R_{Y,\mathcal{C}} = \max_t \max_{z \in \mathcal{C}} \|\mathbf{Y}_t z\| \leq R_{\mathcal{M}} R_{\text{nat}}$.*

    (d) *For $G = G_{\text{ex} \to v}$, we have $R_G = \|G_{\text{ex} \to v}\|_{\ell_1, \text{op}} \leq R_{\pi_0}$, $\psi_G \leq \psi_{\pi_0}$, and $R_H \leq R_{\pi_0} R_{\text{nat}}$*

    (e) *We have $R_v \leq R_{\text{nat}}$, and $L_{\text{eff}} \leq 2L R_{\pi_0} R_{\mathcal{M}} R_{\text{nat}}$.*

*Moreover, $d = m d_u d_\omega$*

We are now ready to prove our general regret bound for the known system case, encompassing

*Proof of Theorem 3.1b.* From Theorem 2.1, we have the bound:

$$\text{MemoryReg}_T(\text{alg}) = \sum_{t=1}^{T} F_t(\mathbf{z}_{t:t-h}) - \min_{z \in \mathcal{C}} \sum_{t=1}^{T} f_t(z) \leq 3\alpha h D^2 R_H^2 + \frac{3 d h^2 L_{\text{eff}}^2 R_G}{\alpha \kappa^{1/2}} \log(1 + T),$$

Let us now specific the above constants using Lemma E.6. From this lemma, we have that $\alpha h D^2 R_H^2 = \alpha h m R_{\pi_0}^2 R_{\text{nat}}^2 R_{\mathcal{M}}^2$. Moreover, $d h^2 L_{\text{eff}}^2 R_G = 4 m h^2 d_u d_\omega L^2 R_{\pi_0}^3 R_{\mathcal{M}}^2 R_{\text{nat}}^2$. Thus, with $\lambda := 6 h R_{\text{nat}}^2 R_{\pi_0}^2$ and $\eta = 1/\alpha$, we get

$$\begin{aligned}
\text{MemoryReg}_T(\text{alg}) &\lesssim m h^2 R_{\pi_0}^2 R_{\text{nat}}^2 R_{\mathcal{M}}^2 (\alpha + (\alpha\sqrt{\kappa})^{-1} L^2 R_{\pi_0} d_u d_\omega \log(1 + T)) \\
&\lesssim (\alpha\sqrt{\kappa})^{-1} m h^2 d_u d_\omega R_{\pi_0}^3 R_{\text{nat}}^2 R_{\mathcal{M}}^2 L^2 \log(1 + T),
\end{aligned}$$

where we used that $L^2/\alpha\sqrt{\kappa} \geq L^2/\alpha \geq \alpha$ by the assumption $\alpha \leq L$. Combining with Proposition E.5 and again using $L \leq L^2/\alpha\sqrt{\kappa}$ ensures that the total control regret $\text{ControlReg}_T$ suffers an additional constant $L$ in the bound, yielding at most

$$\text{ControlReg}_T(\text{alg}) \lesssim (\alpha\sqrt{\kappa})^{-1} m h^2 d_u d_\omega R_{\pi_0}^3 R_{\text{nat}}^2 R_{\mathcal{M}}^2 L^2 \log(1 + T),$$

as needed.

$\square$

### E.4.1   Proof of Proposition E.5

We follow the regret decomposition from [28], noting that our assumptions on the dynamics, magnitude bounds, and costs $c_t$ all align. To facilitate reuse of the technical material from [28], we introduce the following loss notation in the $M$-domain:

**Definition 3.3b** (Losses for the analysis). Generalizing Definition 3.3, we introduce the $z$-space losses,

$$F_t(z_{t:t-h} \mid \widehat{\boldsymbol{\omega}}_{1:t}^{\text{nat}}) := \ell_t(\mathbf{v}_t^{\text{nat}} + \sum_{i=0}^{h} G_{\text{ex} \to v}^{[i]} \mathbf{Y}_{t-i} z_{t-i}), \text{ where } \mathbf{Y}_s = \mathfrak{e}_\omega(\widehat{\boldsymbol{\omega}}_{1:s}^{\text{nat}}),$$

with unary specialization $F_t(z_{t:t-h} \mid \widehat{\boldsymbol{\omega}}_{1:t}^{\text{nat}}) := f_t(z, \ldots, z \mid \widehat{\boldsymbol{\omega}}_{1:t}^{\text{nat}})$. and their analogues in $M$-space

$$\bar{F}_t(M_{t:t-h} \mid \widehat{\boldsymbol{\omega}}_{1:t}^{\text{nat}}) := \ell_t(\mathbf{v}_t^{\text{nat}} + \sum_{i=0}^{h} G_{\text{ex} \to v}^{[i]} \mathbf{u}_t^{\text{ex}}(M \mid \widehat{\boldsymbol{\omega}}_{1:t}^{\text{nat}})),$$

and unary specialization $\bar{f}_t(M \mid \widehat{\boldsymbol{\omega}}_{1:t}^{\text{nat}}) := \bar{F}_t(M, \ldots, M \mid \widehat{\boldsymbol{\omega}}_{1:t}^{\text{nat}})$. Observe that, for $\mathbf{z}_s = \mathfrak{e}(\mathbf{M}_s)$ for $s \in [T]$, and $z = \mathfrak{e}(M)$, then

$$F_t(\mathbf{z}_{t:t-h} \mid \widehat{\boldsymbol{\omega}}_{1:t}^{\text{nat}}) = \bar{F}_t(\mathbf{M}_{t:t-h} \mid \widehat{\boldsymbol{\omega}}_{1:t}^{\text{nat}}), \quad \text{and} \quad f_t(z \mid \widehat{\boldsymbol{\omega}}_{1:t}^{\text{nat}}) = \bar{f}_t(M \mid \widehat{\boldsymbol{\omega}}_{1:t}^{\text{nat}}). \quad \text{(E.4)}$$

Moving forward, let $(\mathbf{y}^M, \mathbf{u}^M)$ denote the sequence produced by selecting input $\mathbf{u}_t^{\mathrm{ex}}(M \mid \boldsymbol{\omega}_{1:t}^{\mathrm{nat}})$ at each $i$. We then have

$\mathrm{ControlReg}_T(\mathsf{alg}; \Pi_\star)$

$$= \sum_{t=1}^T \ell_t(\mathbf{y}_t^{\mathsf{alg}}, \mathbf{u}_t^{\mathsf{alg}}) - \inf_{\pi \in \Pi_\star} \sum_{t=1}^T \ell_t(\mathbf{y}_t^\pi, \mathbf{u}_t^\pi)$$

$$\leq \underbrace{\sum_{t=1}^T \left| \ell_t(\mathbf{y}_t^{\mathsf{alg}}, \mathbf{u}_t^{\mathsf{alg}}) - \bar{F}_t(\mathbf{M}_{t:t-h} \mid \boldsymbol{\omega}_{1:t}^{\mathrm{nat}}) \right| }_{(i.a)} + \underbrace{\sum_{t=1}^T \bar{F}_t(\mathbf{M}_{t:t-h} \mid \boldsymbol{\omega}_{1:t}^{\mathrm{nat}}) - \inf_{M \in \mathcal{M}} \sum_{t=1}^T \bar{f}_t(M \mid \boldsymbol{\omega}_{1:t}^{\mathrm{nat}})}_{(ii)}$$

$$+ \underbrace{\max_{M \in \mathcal{M}} \sum_{t=1}^T \left| \bar{f}_t(M \mid \boldsymbol{\omega}_{1:t}^{\mathrm{nat}}) - \ell_t(\mathbf{y}_t^M, \mathbf{u}_t^M) \right|}_{(i.b)} + \underbrace{\left| \inf_{M \in \mathcal{M}} \sum_{t=1}^T \ell_t(\mathbf{y}_t^M, \mathbf{u}_t^M) - \inf_{\pi \in \Pi_\star} \sum_{t=1}^T \ell_t(\mathbf{y}_t^\pi, \mathbf{u}_t^\pi) \right|}_{(iii)}.$$

Let's proceed term by term. From Simchowitz et al. [28, Lemma 5.3] (replacing their notation $R_{G_\star}, \psi_{G_\star}$ with our notation $R_{\pi_0}, \psi_{\pi_0}$),

$$(i.a) + (i.b) \leq 4LTR_{\pi_0}R_{\mathcal{M}}^2 R_{\mathrm{nat}}^2 \psi_{\pi_0}(h+1). \tag{E.5}$$

Secondly, from Eq. (E.4), we have

$$(ii) = \sum_{t=1}^T F_t(\mathbf{z}_{t:t-h} \mid \boldsymbol{\omega}_{1:t}^{\mathrm{nat}}) - \inf_{z \in \mathcal{M}_\mathfrak{e}} \sum_{t=1}^T f_t(z \mid \boldsymbol{\omega}_{1:t}^{\mathrm{nat}}) := \mathrm{MemoryReg}_T(\mathsf{alg}). \tag{E.6}$$

Finally, from Simchowitz et al. [28, Theorem 1b], we have that for $R_{\mathcal{M}} \geq R_\star$,

$$(iii) \leq 2LTR_\star R_{\pi_0}^2 R_{\mathrm{nat}}^2 \psi(m) \tag{E.7}$$

Thus, we obtain

$\mathrm{ControlReg}_T(\mathsf{alg}; \Pi_\star) \leq (i.a) + (i.b) + (ii) + (iii)$

$$\leq \mathrm{MemoryReg}_T(\mathsf{alg}) + 4LTR_{\mathcal{M}}^2 R_{\pi_0}^2 R_{\mathrm{nat}}^2 \left( \frac{\psi_\star(m)}{R_\star} + \frac{2\psi_{\pi_0}(h+1)}{R_{\pi_0}} \right),$$

Finally, bound $\psi_\star(m) \leq cR_\star/T$ and $2\psi_{\pi_0}(h+1) \leq cR_{\pi_0}/T$ concludes. $\qquad\square$

### E.4.2 Proof of Lemma E.6

We go term by term:

(a) We have $D \leq 2\max\{\|z\| : z \in \mathcal{M}_\mathfrak{e}\}$. For $z = \mathfrak{e}(M)$, have that $\|z\| = \|M\|_{\mathrm{F}} \leq \sqrt{m}\|M\|_{\ell_1,\mathrm{op}} \leq \sqrt{m}R_{\mathcal{M}}$ by Simchowitz et al. [28, Lemma D.1]

(b) Each matrix $\mathbf{Y}_t$ can be represented as a block diagonal, with blocks as rows corresponding to $\boldsymbol{\omega}_s^{\mathrm{nat}}$ for $s \in \{t, t-1, \ldots, t-m+1\}$. This matrix has operator norm as most $\max\{\|\boldsymbol{\omega}_s^{\mathrm{nat}}\| : s \in \{t, t-1, \ldots, t-m+1\}\} \leq R_{\mathrm{nat}}$.

(c) We have that $\mathbf{Y}_t z = \mathbf{u}_t^{\mathrm{ex}}(M \mid \boldsymbol{\omega}_{1:t}^{\mathrm{nat}}) \leq \sum_{i=0}^{m-1} \|M^{[i]}\|_{\mathrm{op}} \|\boldsymbol{\omega}_{t-i}^{\mathrm{nat}}\|_{\mathrm{op}} \leq R_{\mathcal{M}} R_{\mathrm{nat}}$ by Holder's inequality.

(d) These bounds followly directly from our definitions.

(e) We have $R_v \leq R_{\mathrm{nat}}$ by assumption, and $L_{\mathrm{eff}} := 2LR_{\pi_0}R_{\mathcal{M}}R_{\mathrm{nat}}$ follows from the definition $L_{\mathrm{eff}} = L\max\{R_v + R_G R_{Y,\mathcal{C}}\}$, and the assumption s $R_{\mathcal{M}}, R_{\mathrm{nat}} \geq 1$, and $R_{\pi_0} \geq 1$ by definition ($R_{\pi_0} = \|G_{\mathrm{ex}\to v}\|_{\ell_1,\mathrm{op}}$, and $G_{\mathrm{ex}\to v}^{[i]} = \begin{bmatrix} 0 \\ I \end{bmatrix}$).

$\qquad\square$

### E.5 Unknown Systen

We begin by stating guarantees for the estimation procedures Algorithm 3 and Algorithm 5, which follow directly past work:

**Lemma E.7** ( Theorem 6b in Simchowitz et al. [28]). *Let $\delta \in (e^{-T}, T^{-1})$, $N, d_u \leq T$, and $\psi_{G_\star}(h+1) \leq \frac{1}{\sqrt{N}}$. Define $d_{\max} = \max\{d_y + d_u, d_\omega\}$, and set*

$$\epsilon_G(N, \delta) = \frac{h^2 R_{\mathrm{nat}}}{\sqrt{N}} C_\delta, \quad \text{where } C_\delta := 14\sqrt{d_u + d_{\max} + \log\tfrac{1}{\delta}}, \quad \text{and } R_{\mathbf{u},\mathrm{est}} := 3\sqrt{d_u + \log(1/\delta)}.$$

*and suppose that $N \geq h^4 C_\delta^2 R_{\mathbf{u},\mathrm{est}}^2 R_{\mathcal{M}}^2 R_{\pi_0}^2 + c_0 h^2 d_u^2$ for an appropriately large $c_0$, which can be satisfied by taking*

$$N \geq 1764(d_{\max} + d_u + \log(1/\delta))^2 h^4 R_{\mathcal{M}}^2 R_{\pi_0}^2 + c_0 h^2 d_u^2.$$

*Then with probability $1 - \delta - N^{-\log^2 N}$, Algorithm 5 satisfies the following bounds*

1. *$\epsilon_G \leq 1/\max\{R_{\mathbf{u},\mathrm{est}}, R_{\mathcal{M}} R_{\pi_0}\}$.*

2. *For all $t \in [N]$, $\|\mathbf{u}_t\| \leq R_{\mathbf{u},\mathrm{est}} := 3\sqrt{d_u + \log(1/\delta)}$*

3. *For estimation error is bounded as*

$$\|\widehat{G}_{\mathrm{ex}\to\omega} - G_{\mathrm{ex}\to\omega}\|_{\ell_1,\mathrm{op}} \leq \|\widehat{G}_{\mathrm{ex}\to\omega}^{[0:h]} - G_{\mathrm{ex}\to\omega}^{[0:h]}\|_{\ell_1,\mathrm{op}} + R_{\mathbf{u},\mathrm{est}}\psi_{G_\star}(h+1) \leq \epsilon_G$$

$$\|\widehat{G}_{\mathrm{ex}\to(y,u)} - G_{\mathrm{ex}\to v}\|_{\ell_1,\mathrm{op}} \leq \|\widehat{G}_{\mathrm{ex}\to(y,u)}^{[1:h]} - G_{\mathrm{ex}\to v}^{[1:h]}\|_{\ell_1,\mathrm{op}} + R_{\mathbf{u},\mathrm{est}}\psi_{G_\star}(h+1) \leq \epsilon_G.$$

*Moreover, Algorithm 3 also satisfies the above for $\widehat{G}_{\mathrm{ex}\to(y,u)} = \widehat{G}$ and $G_{\mathrm{ex}\to v} = G_K$.*

The above bounds are in turn a consequence of Simchowitz et al. [27]. We denote the event of Lemma E.7 as $\mathcal{E}^{\mathrm{est}}$, and the following exposition assumpt it holds.

Next, we state a blackbox reduction to the DRC online controller framework. This reduction crucially uses the fact that we have *over-parameterized* the set $\mathcal{M}$. Specifically, over comparator set is

$$\mathcal{M}_\star := M_{\mathrm{drc}}(m_\star, R_\star),$$

whereas the algorithm uses the over-parametrized set

$$\mathcal{M} := M_{\mathrm{drc}}(m, R_{\mathcal{M}}), \text{ with } R_{\mathcal{M}} \geq 2R_\star \text{ and } m \geq 2m_\star + h. \tag{E.8}$$

By over-parametrizing the controller set as above, we obtain the following guarantee:

**Proposition E.8** (Reduction to policy regret for known dynamics). . *Suppose that Eq. (E.8) holds, and that $\psi_{\pi_0}(h+1) \leq cR_{\pi_0}/T$ and $\psi_\star(m) \leq cR_\star/T$ for some $c > 1$, and that $N \geq m + h$. Consider the DRC-ONS-DYN algorithm with estimation (Algorithm 5) initialized with the exact Markov operators $\widehat{G}_{\mathrm{ex}\to(y,u)} = G_{\mathrm{ex}\to v}, \widehat{G}_{\mathrm{ex}\to\omega} = G_{\mathrm{ex}\to\omega}$, and iterates $\mathbf{M}_t$ produced by an arbitrary blackbox optimization procedure $\mathcal{A}$.*

$$\mathrm{ControlReg}_T(\mathrm{alg}; \Pi_\star) \leq \widehat{\mathrm{MemoryReg}}_T(z_\star) + \nu \sum_{t=N+m+2h+1}^{T} \|\mathbf{Y}_t(\mathbf{z}_t - z_\star)\|_2^2$$
$$+ \mathcal{O}_{\mathrm{cnst}(LR_{\pi_0}^3(N+cm))}\left(d_u + \log(1/\delta) + R_{\mathcal{M}}^4 R_{\mathrm{nat}}^2\right)$$
$$+ \mathcal{O}_{\mathrm{cnst}(LR_{\mathcal{M}}^3 R_{\pi_0}^2 R_{\mathrm{nat}}^2 T\epsilon_G^2)}\left(1 + \frac{LmR_{\pi_0}^2}{\nu}\right)$$

*where $\mathcal{O}_{\mathrm{cnst}(1)}$ hides a universal numerical constants. Here, for the $F_t, f_t$ losses in Definition 3.3b, we define the term:*

$$\widehat{\mathrm{MemoryReg}}_T(\mathrm{alg}; z_\star) := \sum_{t=N+m+2h+1}^{T} F_t(\mathbf{z}_{t:t-h} \mid \widehat{\boldsymbol{\omega}}_{1:t}^{\mathrm{nat}}) - \inf_{z \in \mathcal{M}_\mathfrak{e}} \sum_{t=N+m+2h+1}^{T} f_t(z \mid \widehat{\boldsymbol{\omega}}_{1:t}^{\mathrm{nat}}).$$

*Moreover, the same guarantee is also true of Algorithm 3.*

Again, we allow a slack parameter $c$ to allow for over-specifying $m, h$, demonstrating low sensitivity to imperfectly tuned algorithm parameters. Next, we translate the parameter bounds from the control setting to the ones required for the policy regret analysis of Semi-ONS:

**Lemma E.9** (Parameter Bounds for Unknown Setting). *Assume $R_{\mathrm{nat}} \geq 1$, and that $\cdot$. Then, for $t_0 := N + m + h + 1$, the following hold*

(a) *We have $D = \max\{\|z - z'\| : z, z' \in \mathcal{M}_{\mathfrak{c}}\} \leq \sqrt{m} R_{\mathcal{M}}$.*

(b) *We have $R_Y := \max_{t \geq t_0} \|\mathbf{Y}_t\|_{\mathrm{op}} \leq 2R_{\mathrm{nat}}$.*

(c) *We have $R_{Y,\mathcal{C}} = \max_{t \geq t_0} \max_{z \in \mathcal{C}} \|\mathbf{Y}_t z\| \leq 2R_{\mathcal{M}} R_{\mathrm{nat}}$.*

(d) *For $G = G_{\mathrm{ex} \to v}$, we have $R_G = |\widehat{G}_{\mathrm{ex} \to (y,u)}\|_{\ell_1,\mathrm{op}} \vee \|G_{\mathrm{ex} \to v}\|_{\ell_1,\mathrm{op}} \leq 2R_{\pi_0}, \psi_G \leq \psi_{\pi_0}$, and $R_H \leq 2R_{\pi_0} R_{\mathrm{nat}}$*

(e) *We have $R_v := \max_{t \geq t_0} \|\mathbf{v}_t^K\| \vee \|\widehat{\mathbf{v}}_t^K\| \leq 2R_{\mathrm{nat}}$, and $L_{\mathrm{eff}} := 8L R_{\pi_0} R_{\mathcal{M}} R_{\mathrm{nat}}$.*

(f) *We can take $c_v$ to be $3R_{\mathcal{M}} R_{\mathrm{nat}}$.*

*Moreover, $d = d_\omega d_y m$*

Finally, we are in place to prove our main theorem:

*Proof of Lemma E.9.* The bounds follow analogously to those in Lemma E.6, with the modification that, for $t \geq N + h$, we have $\|\widehat{\boldsymbol{\omega}}_t^{\mathrm{nat}}\| \leq 2R_{\mathrm{nat}}$ (by Simchowitz et al. [28, Lemma 6.1]), and that $\|\widehat{G}_{\mathrm{ex} \to (y,u)}\|_{\ell_1,\mathrm{op}} \leq 2R_{\pi_0}$ under $\mathcal{E}^{\mathrm{est}}$. Moreover, we can take the constant $c_v$ which bounds $\|\widehat{\mathbf{u}}_t^{\mathrm{nat}} - \mathbf{v}_t^{\mathrm{nat}}\|_2 \leq c_v \epsilon_G$ to be $3R_{\mathcal{M}} R_{\mathrm{nat}}$ by Simchowitz et al. [28, Lemma 6.4b]. $\square$

*Proof of Theorem 3.2b.* Let us prove the bound for the dynamic-controller variant Algorithm 5; the static-controller variant works similarly. Recall that we assume the following

- $\lambda = R_{\mathrm{nat}}^2 \log(1/\delta)\sqrt{T} + hR_{\pi_0}^2, \eta = 3/\alpha$

- $N = h^2\sqrt{T}d_{\max}$

- $\sqrt{T} \geq 4 \cdot 1764h^2 R_{\mathcal{M}}^2 R_{\pi_0}^2 + c_0 h^2 d_u^2$

- $m \geq m_\star + 2h, R_{\mathcal{M}} \geq 2R_\star$

- $\psi_{\pi_0}(h + 1) \leq R_{\pi_0}/T, \psi_\star(m) \leq R_\star/T$.

Let $\epsilon_G$ be an upper bound on the estimation error, which we will set to be greater than $\sqrt{T}$. By taking $\lambda \in [c_\lambda, 1](T\epsilon_G^2 + hR_H^2)$, and applying Theorem 2.2a, we can bound

$$\widehat{\mathrm{MemoryReg}}_T(z_\star) + \nu \sum_{t=N+m+2h+1}^{T} \|\mathbf{Y}_t(\mathbf{z}_t - z_\star)\|_2^2 \lesssim$$

$$c_\lambda^{-1} \log(1 + \frac{T}{c_\lambda}) \left( \frac{C_1}{\alpha\kappa^{1/2}} + C_2 \right) \left( T\epsilon_G^2 + h^2(R_G^2 + R_Y) \right),$$

where $C_1 := (1 + R_Y)R_G(h + d)L_{\mathrm{eff}}^2, C_2 := (L^2 c_v^2/\alpha + \alpha D^2)$, and $\nu_\star = \frac{\alpha\sqrt{\kappa}}{48(1+R_Y)}$ are constants which we must bound presently. Since $d = d_\omega d_y m \geq h, L \geq \alpha$, and $\kappa \leq 1$

$$C_1 \lesssim d_\omega d_y m R_{\mathrm{nat}} R_{\pi_0} L_{\mathrm{eff}}^2 \lesssim d_\omega d_y m L^2 R_{\pi_0}^3 R_{\mathrm{nat}}^3 R_{\mathcal{M}}^2$$

$$C_2 \lesssim L^2/\alpha R_{\mathrm{nat}}^2 R_{\mathcal{M}}^2 + mR_{\mathcal{M}}^2 \lesssim L^2/\alpha(mR_{\mathrm{nat}}^2 R_{\mathcal{M}}^2) \leq \frac{L^2}{\alpha\sqrt{\kappa}}(mR_{\mathrm{nat}}^2 R_{\mathcal{M}}^2).$$

Thus, we can bound

$$\left( \frac{C_1}{\alpha\kappa^{1/2}} + C_2 \right) \lesssim \frac{d_\omega d_y m L^2 R_{\pi_0}^3 R_{\mathrm{nat}}^3 R_{\mathcal{M}}^2}{\alpha\kappa^{1/2}}.$$

Thus, from Proposition E.8 with $\nu = \nu_\star$, taking $c = 1$, and bounding $R_G \lesssim R_{\pi_0}$, $R_Y \lesssim R_{\mathrm{nat}}$ from Lemma E.9

$$\mathrm{ControlReg}_T(\mathsf{alg}; \Pi_\star) \lesssim c_\lambda^{-1} \log(1 + \frac{T}{c_\lambda}) \frac{d_\omega d_y m L^2 R_{\pi_0}^3 R_{\mathrm{nat}}^3 R_{\mathcal{M}}^2}{\alpha \kappa^{1/2}} \left( T\epsilon_G^2 + h^2(R_{\pi_0}^2 + R_{\mathrm{nat}}) \right),.$$

$$+ LR_{\pi_0}^3 (N + m) \left( d_u + \log(1/\delta) + R_{\mathcal{M}}^4 R_{\mathrm{nat}}^2 \right) + LR_{\mathcal{M}}^3 R_{\pi_0}^2 R_{\mathrm{nat}}^2 T \epsilon_G^2 \left( 1 + \frac{Lm R_{\pi_0}^2}{\nu_\star} \right)$$

Using the above bounds we have $\nu_\star = \frac{\alpha\sqrt{\kappa}}{48(1+R_Y)} \gtrsim \alpha\sqrt{\kappa}/R_{\mathrm{nat}}$. Thus, for $L \geq \alpha$ and $\kappa \leq 1$, the term $\frac{Lm R_{\pi_0}^2}{\nu_\star}$ dominates 1, and we have

$$LR_{\mathcal{M}}^3 R_{\pi_0}^2 R_{\mathrm{nat}}^2 T \epsilon_G^2 \left( 1 + \frac{Lm R_{\pi_0}^2}{\nu_\star} \right) \lesssim \frac{L^2 R_{\mathcal{M}}^3 R_{\pi_0}^4 R_{\mathrm{nat}}^3 m}{\alpha\sqrt{\kappa}}$$

Moreover, using $N \geq m$ by assumption and aggregating terms and simplifying

$$\mathrm{ControlReg}_T(\mathsf{alg}; \Pi_\star) \lesssim c_\lambda^{-1} \log(1 + \frac{T}{c_\lambda}) \frac{d_\omega d_y m L^2 R_{\pi_0}^4 R_{\mathrm{nat}}^3 R_{\mathcal{M}}^3}{\alpha \kappa^{1/2}} \left( T\epsilon_G^2 + h^2(R_{\pi_0}^2 + R_Y) \right),.$$

$$+ LR_{\pi_0}^3 N \left( d_u + \log(1/\delta) + c R_{\mathcal{M}}^4 R_{\mathrm{nat}}^2 \right).$$

Next, recall $d_{\max} := \max\{d_u + d_y, d_\omega\}$, let us take $N = h^2\sqrt{T} d_{\max}$. From Lemma E.7, this yields $\epsilon_G^2 = \frac{h^4 R_{\mathrm{nat}}^2}{N} C_\delta^2 \approx \frac{h^4 R_{\mathrm{nat}}^2 (d_{\max} + \log(1/\delta))}{N} = R_{\mathrm{nat}}^2 \log(1/\delta)/\sqrt{T}$ and that $\epsilon_G^2 \geq \sqrt{T}$. This yields

$$\mathrm{ControlReg}_T(\mathsf{alg}; \Pi_\star) \lesssim c_\lambda^{-1} \log(1 + \frac{T}{c_\lambda}) \frac{d_\omega d_y m L^2 R_{\pi_0}^4 R_{\mathrm{nat}}^3 R_{\mathcal{M}}^3}{\alpha \kappa^{1/2}} \left( \sqrt{T} R_{\mathrm{nat}}^2 \log(1/\delta) + h^2(R_{\pi_0}^2 + R_{\mathrm{nat}}) \right),.$$

$$+ LR_{\pi_0}^3 h^2 \sqrt{T} d_{\max} \left( d_u + \log(1/\delta) + R_{\mathcal{M}}^4 R_{\mathrm{nat}}^2 \right)$$

Finally, we us bound $LR_{\pi_0}^3 h^2\sqrt{T} d_{\max} \left( d_u + \log(1/\delta) + R_{\mathcal{M}}^4 R_{\mathrm{nat}}^2 \right) \leq LR_{\mathcal{M}}^4 R_{\mathrm{nat}}^2 R_{\pi_0}^3 h^2 \log(1/\delta) d_u$, and take $L \leq L^2/\alpha \leq L^2/\alpha\sqrt{\kappa}$. Thus, we can bound the above by

$$\mathrm{ControlReg}_T(\mathsf{alg}; \Pi_\star) \lesssim c_\lambda^{-1} \log(1 + \frac{T}{c_\lambda}) \frac{d_\omega d_y (m + h^2) L^2 R_{\pi_0}^4 R_{\mathrm{nat}}^3 R_{\mathcal{M}}^4}{\alpha \kappa^{1/2}} \left( \sqrt{T} R_{\mathrm{nat}}^2 \log(1/\delta) + R_{\pi_0}^2 \right).$$

Finally, for $\lambda = R_{\mathrm{nat}}^2 \log(1/\delta)\sqrt{T} + hR_{\pi_0}^2$, we can take $c_\lambda \approx 1$. Together with $m + h^2 \leq mh$ under the present assumption, we conclude

$$\mathrm{ControlReg}_T(\mathsf{alg}; \Pi_\star) \lesssim \log(1 + T) \frac{(d_\omega d_y + d_{\max} d_u) mh L^2 R_{\pi_0}^4 R_{\mathrm{nat}}^3 R_{\mathcal{M}}^4}{\alpha \kappa^{1/2}} \left( \sqrt{T} R_{\mathrm{nat}}^2 \log(1/\delta) + R_{\pi_0}^2 \right).$$

Finally, we require $N \geq 1764(d_{\max} + d_u + \log(1/\delta))^2 h^4 R_{\mathcal{M}}^2 R_{\pi_0}^2 + c_0 h^2 d_u^2$., which means for our choice of $N = h^2\sqrt{T} d_{\max}$ and $d_{\max} \geq d_u$, our stipulation that $\sqrt{T} \geq 4 \cdot 1764 h^2 R_{\mathcal{M}}^2 R_{\pi_0}^2 + c_0 h^2 d_u^2$ suffices. This ensures in turn that $\sqrt{T} R_{\mathrm{nat}}^2 \log(1/\delta)$ dominates $R_{\pi_0}^2$, allowing us to drop the term from the final bound, ultimately yields

$$\mathrm{ControlReg}_T(\mathsf{alg}; \Pi_\star) \lesssim \log(1 + T) \frac{(d_\omega d_y + d_{\max} d_u) mh L^2 R_{\pi_0}^4 R_{\mathrm{nat}}^5 R_{\mathcal{M}}^4 \sqrt{T} \log(1/\delta)}{\alpha \kappa^{1/2}}.$$

Finally, using $d_{\max} = \max\{d_\omega, d_y + d_u\}$, we have $(d_\omega d_y + d_{\max} d_u) \leq d_\omega(d_y + d_u) + d_u(d_y + d_u) = (d_\omega + d_y)(d_y + d_u)$, concluding the bound.

$\square$

### E.5.1 Proof of Proposition E.5

Recall that $\bar{f}_t, \bar{F}_t$ losses from Definition 3.3b. In a fixed a comparator matrix $\overline{M} \in \mathcal{M}$, where we recall $\mathcal{M} = M_{\mathrm{drc}}(m, R_{\mathcal{M}})$, where $R_{\mathcal{M}} \geq 2R_\Pi$ and $m \geq 2m_\star - 1 + h$. $\overline{M}$ will be chosen towards

the proof in a careful way, and is not necessarily the best-in-hindsight parameter on the $\overline{M}$ sequence. Our regret decomposition is as follows:

$$\text{ControlReg}_T(\text{alg}; \Pi_\star) = \sum_{t=1}^{T} \ell_t(\mathbf{y}_t^{\text{alg}}, \mathbf{u}_t^{\text{alg}}) - \inf_{\pi \in \Pi_\star} \sum_{t=1}^{T} \ell_t(\mathbf{y}_t^{\pi}, \mathbf{u}_t^{\pi})$$

$$\leq \underbrace{\sum_{t=1}^{N+m+2h} \ell_t(\mathbf{y}_t^{\text{alg}}, \mathbf{u}_t^{\text{alg}})}_{(i)} + \underbrace{\sum_{t=N+m+2h+1}^{T} |\ell_t(\mathbf{y}_t^{\text{alg}}, \mathbf{u}_t^{\text{alg}}) - \bar{F}_t(\mathbf{M}_{t:t-h} \mid \widehat{\boldsymbol{\omega}}_{1:t}^{\text{nat}})|}_{(ii.a)}$$

$$+ \underbrace{\sum_{t=N+m+2h+1}^{T} \bar{F}_t(M_{t:t-h} \mid \widehat{\boldsymbol{\omega}}_{1:t}^{\text{nat}}) - \sum_{t=N+m+2h+1}^{T} \bar{f}_t(\overline{M} \mid \widehat{\boldsymbol{\omega}}_{1:t}^{\text{nat}})}_{(iii)}$$

$$+ \underbrace{\sum_{t=N+m+2h+1}^{T} f_t(\overline{M} \mid \widehat{\boldsymbol{\omega}}_{1:t}^{\text{nat}}) - \inf_{M' \in \mathcal{M}_\star} \sum_{t=N+m+2h+1}^{T} \bar{f}_t(M' \mid \boldsymbol{\omega}_{1:t}^{\text{nat}})}_{(iv)}$$

$$+ \underbrace{\max_{M \in \mathcal{M}_\star} |\sum_{t=1}^{T} \bar{f}_t(M \mid \boldsymbol{\omega}_{1:t}^{\text{nat}}) - \ell_t(\mathbf{y}_t^M, \mathbf{u}_t^M)|}_{(ii.b)} + \underbrace{\left| \inf_{M \in \mathcal{M}_\star} \sum_{t=1}^{T} \ell_t(\mathbf{y}_t^M, \mathbf{u}_t^M) - \inf_{\pi \in \Pi_\star} \sum_{t=1}^{T} \ell_t(\mathbf{y}_t^{\pi}, \mathbf{u}_t^{\pi}) \right|}_{(v)}.$$

Again, let us work term-by-term, starting with the terms which are most similar to the terms that arise in the known system. Together with $R_{\mathcal{M}} \geq R_\star$, the last two terms can be bounded via Eq. (E.5) and Eq. (E.7)

$$(ii.b) + (v) \lesssim LTR_{\mathcal{M}}^2 R_{\pi_0}^2 R_{\text{nat}}^2 \left( \frac{\psi_\star(m_\star)}{R_\star} + \frac{2\psi_{\pi_0}(h+1)}{R_{\pi_0}} \right).$$

Moreover, similar arguments can be used to bound $(ii.a) \lesssim$ RHS of Eq. (E.5) (specifically, one replaces the appearance of $\boldsymbol{\omega}_t^{\text{nat}}$ in the proof Simchowitz et al. [28, Lemma 5.3] with $\widehat{\boldsymbol{\omega}}_t^{\text{nat}}$, and uses the bound $\|\widehat{\boldsymbol{\omega}}_t^{\text{nat}}\| \leq 2R_{\text{nat}}$ by Simchowitz et al. [28, Lemma 6.1] ). Thus, we have so far

$$(ii.a) + (ii.b) + (v) \lesssim LTR_{\mathcal{M}}^2 R_{\pi_0}^2 R_{\text{nat}}^2 \left( \frac{\psi_\star(m_\star)}{R_\star} + \frac{2\psi_{\pi_0}(h+1)}{R_{\pi_0}} \right).$$

Next, analogously to Eq. (E.6), we recognize that

$$(iii) = \widehat{\text{MemoryReg}}_T(z_\star), \text{ for } z_\star := \mathfrak{e}(\overline{M}).$$

Furthermore, from Simchowitz et al. [28, Lemma 6.3] and the definition of the term $\overline{R}_{\mathbf{u}}$ in Simchowitz et al. [28, Lemma 6.1b], and with $N \geq m+2h$, we have $(i) \lesssim LNR_{\pi_0}^2(R_{\mathbf{u},\text{est}} + R_{\mathcal{M}}R_{\text{nat}})^2$. Thus, collecting what we have thus far, we obtain

$$\text{ControlReg}_T(\text{alg}; \Pi_\star)$$
$$\leq \widehat{\text{MemoryReg}}_T(z_\star) + (iv)$$
$$+ \mathcal{O}_{\text{cnst}(1)} \cdot LR_{\pi_0}^2 \cdot \left( N(R_{\mathbf{u},\text{est}} + R_{\mathcal{M}}R_{\text{nat}})^2 + TR_{\mathcal{M}}^2 R_{\text{nat}}^2 \left( \frac{\psi_\star(m_\star)}{R_\star} + \frac{\psi_{\pi_0}(h+1)}{R_{\pi_0}} \right) \right),$$

where $\mathcal{O}_{\text{cnst}(1)}$ supresses a universal constant. It remains to account for the term $(iv)$. In particular, for $\psi_{\pi_0}(h+1) \leq cR_{\pi_0}/T$ and $\psi_\star(m_\star) \leq cR_\star/T$, the above simplies to

$$\text{ControlReg}_T(\text{alg}; \Pi_\star) \leq \widehat{\text{MemoryReg}}_T(z_\star) + (iv)$$
$$+ \mathcal{O}_{\text{cnst}(L)} R_{\pi_0}^2 \cdot \left( (N+c)(R_{\mathbf{u},\text{est}}^2 + R_{\mathcal{M}}^2 R_{\text{nat}}^2) \right), \quad \text{(E.9)}$$

**Lemma E.10** (Slight Modification of Equation E.6 in Simchowitz et al. [28], altering numerical constants and allowing $c$ dependence). *Suppose that $\mathcal{E}^{\text{est}}$ holds, and that $\psi_{\pi_0}(h+1) \leq cR_{\pi_0}/T$.*

*Futher, assume $R_{\mathcal{M}} \geq 2R_{\star}$ and $m \geq 2m_{\star} + h$. Then, there exists an $\overline{M} \in \mathcal{M}$ such that, for all $\nu > 0$, we have*

$$\text{Term } (iv) \leq \mathcal{O}_{\text{cnst}(1)} \cdot LR_{\mathcal{M}}^3 R_{\pi_0}^2 R_{\text{nat}}^2 T\epsilon_G^2 \left(1 + \frac{LmR_{\pi_0}^2}{\nu}\right) \tag{E.10}$$

$$+ \mathcal{O}_{\text{cnst}(c)} LR_{\mathcal{M}}^2 R_{\pi_0}^2 R_{\text{nat}}((R_{\mathbf{u},\text{est}} + R_{\mathcal{M}} R_{\text{nat}})R_{\pi_0} + m)$$

$$+ \nu \sum_{t=N+m+2h+1}^{T} \left\| \mathbf{u}_j^{\text{ex}}(\mathbf{M}_j \mid \widehat{\boldsymbol{\omega}}_{1:j}^{\text{nat}}) - \mathbf{u}_j^{\text{ex}}(\overline{M} \mid \widehat{\boldsymbol{\omega}}_{1:j}^{\text{nat}}) \right\|_2^2. \tag{E.11}$$

Absorbing the first $h$ terms in the sum into the term on the first line (using arguments as in Lemma E.6, this contributes $\mathcal{O}_{\text{cnst}(R_{\mathcal{M}}^2 R_{\text{nat}}^2 h)} \leq \mathcal{O}_{\text{cnst}(R_{\mathcal{M}}^2 R_{\text{nat}}^2 m)}$ ), and translating back to our $\mathbf{Y}, z$-notation, we have that there exists a $z_{\star} \in \mathcal{M}_{\mathfrak{e}}$ such that

$$\text{Term } (iv) \leq \mathcal{O}_{\text{cnst}(1)} \cdot LR_{\mathcal{M}}^3 R_{\pi_0}^2 R_{\text{nat}}^2 T\epsilon_G^2 \left(1 + \frac{LmR_{\pi_0}^2}{\nu}\right)$$

$$+ \mathcal{O}_{\text{cnst}(c)} R_{\mathcal{M}}^2 R_{\pi_0}^2 R_{\text{nat}}((R_{\mathbf{u},\text{est}} + R_{\mathcal{M}} R_{\text{nat}})R_{\pi_0} + m)$$

$$+ \nu \sum_{t=N+m+h+1}^{T} \|\mathbf{Y}_t(\mathbf{z}_t - z_{\star})\|_2^2.$$

Putting things together with Eq. (E.9), we have the bound that for $\psi_{\pi_0}(h+1) \leq R_{\pi_0}/T$ and $\psi_{\star} \leq R_{\star}/T$, we find

$$\text{ControlReg}_T(\text{alg}; \Pi_{\star})$$

$$\leq \widehat{\text{MemoryReg}}_T(z_{\star}) + \nu \sum_{t=N+m+2h+1}^{T} \|\mathbf{Y}_t(\mathbf{z}_t - z_{\star})\|_2^2$$

$$+ \mathcal{O}_{\text{cnst}(1)} \cdot LR_{\mathcal{M}}^3 R_{\pi_0}^2 R_{\text{nat}}^2 T\epsilon_G^2 \left(1 + \frac{LmR_{\pi_0}^2}{\nu}\right)$$

$$+ \mathcal{O}_{\text{cnst}(L)} R_{\pi_0}^2 \cdot \left((N+c)(R_{\mathbf{u},\text{est}}^2 + R_{\mathcal{M}}^2 R_{\text{nat}}^2) + cR_{\mathcal{M}}^2 R_{\text{nat}}((R_{\mathbf{u},\text{est}} + R_{\mathcal{M}} R_{\text{nat}})R_{\pi_0} + m)\right)$$

Finally, since $N \geq m$, we bound

$$LR_{\pi_0}^2 \cdot \left(N(R_{\mathbf{u},\text{est}}^2 + cR_{\mathcal{M}}^2 R_{\text{nat}}^2) + R_{\mathcal{M}}^2 R_{\text{nat}}((R_{\mathbf{u},\text{est}} + R_{\mathcal{M}} R_{\text{nat}})R_{\pi_0} + m)\right)$$

$$\leq \mathcal{O}_{\text{cnst}(L)} R_{\pi_0}^3 \left((N+cm)(R_{\mathbf{u},\text{est}}^2 + R_{\mathcal{M}}^3 R_{\text{nat}}^2) + cmR_{\mathbf{u},\text{est}} R_{\mathcal{M}}^2 R_{\text{nat}}\right)$$

$$\leq \mathcal{O}_{\text{cnst}(L)} R_{\pi_0}^3 (N+cm)(R_{\mathbf{u},\text{est}}^2 + cR_{\mathcal{M}}^4 R_{\text{nat}}^2),$$

where the last step is by AM-GM. Thus,

$$\text{ControlReg}_T(\text{alg}; \Pi_{\star})$$

$$\leq \widehat{\text{MemoryReg}}_T(z_{\star}) + \nu \sum_{t=N+m+2h+1}^{T} \|\mathbf{Y}_t(\mathbf{z}_t - z_{\star})\|_2^2$$

$$+ \mathcal{O}_{\text{cnst}(LR_{\pi_0}^3(N+c))} \left(R_{\mathbf{u},\text{est}}^2 + R_{\mathcal{M}}^4 R_{\text{nat}}^2\right) + \mathcal{O}_{\text{cnst}(LR_{\mathcal{M}}^3 R_{\pi_0}^2 R_{\text{nat}}^2 T\epsilon_G^2)} \left(1 + \frac{LmR_{\pi_0}^2}{\nu}\right),$$

which after substituing in $R_{\mathbf{u},\text{est}}^2 \lesssim d_u + \log(1/\delta)$ (Lemma E.7), concludes the bound.

# Part II

# Appendices for OCoM

## F    Proof of Logarithmic Memory Regret (Theorem 2.1)

This section proves Theorem 2.1. We begin by bounding the standard (no-memory) regret in Appendix F.1, and then turn to agressing the contribution of memory in Appendix F.2. All ommitted proofs, as well as the proof of Proposition F.8, are given in Appendix I in numerical order.

### F.1 Bounding the (unary) OCO Regret

As a warmup, we establish a bound on the no-memory regret for Semi-ONS. Throughout, recall the parameters from Definition 2.1, which we assume to be finite.

**Proposition F.1.** *Suppose the the losses satisfy Assumption 1, and $\kappa_h := \kappa(G) > 0$. Then, for $\eta \geq \frac{1}{\alpha}$, Semi-ONS$(\lambda, \eta, \mathcal{C})$ fed pairs $(f_t, \mathbf{H}_t)$ satisfies the following:*

$$\mathrm{OcoReg}_T := \sum_{t=1}^{T} f_t(\mathbf{z}_t) - \min_{z \in \mathcal{C}} \sum_{t=1}^{T} f_t(z) \leq \frac{\eta d L_{\mathrm{eff}}^2}{2} \log \left( 1 + \frac{T R_H^2}{\lambda} \right) + \frac{\lambda D^2}{2\eta}.$$

This section proves the above proposition, and all ommited proofs in the proofs in this section are deferred to Appendix H.2. First, let us establish two simple structural properties of $f_t$:

**Lemma F.2.** *For all $z \in \mathcal{C}$*

1. $\nabla^2 f_t(z) \succeq \alpha \mathbf{H}_t^\top \mathbf{H}_t$

2. *There exists a function $g_t(z) \in \mathbb{R}^{\mathsf{d_v}}$ such that $\nabla f_t(z) = \mathbf{H}_t^\top g_t(z)$, and $\|g_t(z)\| \leq L_{\mathrm{eff}}$. In particular, $\nabla f_t(z) \nabla f_t(z)^\top \preceq L_{\mathrm{eff}}^2 \mathbf{H}_t^\top \mathbf{H}_t$.*

*Proof.* Point (1): By the chain rule and the fact that $\nabla^2(z \mapsto \mathbf{v}_t + \mathbf{H}z) = 0$, we have $\nabla^2 f(z) = \mathbf{H}_t^\top \nabla^2 \ell(\mathbf{v}_t + \mathbf{H}_t z) \mathbf{H}_t$. Since $\ell_t$ is strongly convex, $\nabla^2 \ell(\mathbf{v}_t + \mathbf{H}_t z) \succeq \alpha I$. Point (2): Again invoking the chain rule, $\nabla f_t(z) = \mathbf{H}_t^\top g_t(z)$, where $g_t(z) = \nabla \ell_t(\mathbf{v}_t + \mathbf{H}_t z)$. Since $\ell_t$ is $L$-subquadratic, $\|g_t(z)\| \leq L \max\{1, \|\mathbf{v}_t + \mathbf{H}_t z\|_2\} \leq L \max\{1, R_v + R_G \max_{t,z \in \mathcal{C}} \|\mathbf{Y}_t z\|_2\} = L \max\{1, R_v + R_G R_{Y,\mathcal{C}}\} = L_{\mathrm{eff}}$. $\qquad\square$

Next, we establish a simple quadratic lower bound, which mirrors the basic inequality in analysis of standard ONS:

**Lemma F.3** (Quadratic Lower Bound). *For all $z_1, z_2 \in \mathcal{C}$, we have*

$$f_t(z_1) \geq f_t(z_2) + \nabla f_t(z_2) + \frac{\alpha}{2} \|\mathbf{H}_t(z_1 - z_2)\|_2^2.$$

*Proof of Lemma F.3.* By Taylor's theorem, there exists a $z_3$ on the segment joining $z_1$ and $z_2$ for which $f_t(z_1) \geq f_t(z_2) + \nabla f_t(z_2) + \frac{1}{2} \|(z_1 - z_2)\|_{\nabla^2 f_t(z_3)}^2$. By Lemma F.2, $\nabla^2 f_t(z_3) \succeq \alpha \mathbf{H}_t \mathbf{H}_t^\top$. $\quad\square$

**Remark F.1.** Observe that Lemma F.3 uses the fact that $\nabla^2 f_t(z) \succeq \alpha \mathbf{H}_t \mathbf{H}_t^\top$ *globally*. Lemma F.3 may be *false* if instead one replaces $\mathbf{H}_t^\top \mathbf{H}_t$ in the definition with $\nabla^2 f_t(z_t)$, because the latter may be very large at a given point. This is why we use $\mathbf{H}^\top \mathbf{H}_t$ in the definition of $\Lambda_t$, as opposed to the full-Hessian. This is no longer an issue if one assume that $\nabla^2 f_t(z) \preceq \beta I$ globally, in which case one pays for the conditioning $\beta/\alpha$.

**Remark F.2** (Comparision to Cannonical Online Newton). Let us compare the above to the cannonical Online Newton Step algorithm [19]. This algorithm applies to exp-concave functions, which satisfy the bound $\nabla^2 f \succeq \alpha \nabla f (\nabla f)^\top$ globally. For these functions, the analogue of Lemma F.3, with $f_t(z_1) \geq f_t(z_2) + \nabla f_t(z_2) + \frac{\alpha}{2} \|\nabla f_t(z_2)(z_1 - z_2)\|_2^2$ *does in fact* hold, abeit due to a somewhat trickier argument [18, Lemma 4.3]. This enables the algorithm to use the preconditioner $\Lambda_t = \lambda I + \sum_{s=1}^{t} \nabla f (\nabla f)^\top$. Note however that this yields a smaller pre-conditioner $\Lambda_t$, for which Proposition F.8 may fail.

As a consequence, we obtain intermediate regret bound for Semi-ONS, which mirrors the standard analysis of online newton step (e.g. Hazan [18, Chapter 4]).

**Lemma F.4** (Online Semi-Newton Step Regret). *Suppose that $\eta \geq \frac{1}{\alpha}$. Then,*

$$\sum_{t=1}^{T} f_t(\mathbf{z}_t) - \inf_{z \in \mathcal{C}} \sum_{t=1}^{T} f_t(z) \leq \frac{\lambda D^2}{2\eta} + \frac{\eta}{2} \sum_{t=1}^{T} \nabla_t^\top \Lambda_t^{-1} \nabla_t,$$

Lastly, we recall a standard log-det potential lemma. To facillitate reuse, the lemma is stated for a slightly more general sequence of matrices $\widetilde{\Lambda}_t$:

**Lemma F.5** (Log-det potential). *Suppose that $\widetilde{\Lambda}_t \succeq c \sum_{t=1}^T \mathbf{H}_t^\top \mathbf{H}_t + \lambda_0$. Then,*

$$\sum_{t=1}^T \mathrm{tr}(\mathbf{H}_t \widetilde{\Lambda}_t^{-1} \mathbf{H}_t^\top) \leq \frac{d}{c} \log\left(1 + \frac{cTR_H^2}{\lambda_0}\right)$$

*Proof.* Define $\check{\Lambda}_t = \sum_{t=1}^T \mathbf{H}_t^\top \mathbf{H}_t + \frac{\lambda_0}{c}$. Then, $\sum_{t=1}^T \mathrm{tr}(\mathbf{H}_t \widetilde{\Lambda}_t^{-1} \mathbf{H}_t^\top) \leq \frac{1}{c} \sum_{t=1}^T \mathrm{tr}(\mathbf{H}_t \check{\Lambda}_t^{-1} \mathbf{H}_t^\top)$. The result now follows from the standard log-det potential lemma (see e.g. Hazan [18, Proof of Theorem 4.4]). $\square$

*Proof of Proposition F.1.* Begin with the unary bound:

$$\mathrm{OcoReg}_T := \sum_{t=1}^T f_t(\mathbf{z}_t) - \inf_{z \in \mathcal{C}} \sum_{t=1}^T f_t(z) \leq \frac{\lambda D^2}{2\eta} + \frac{\eta}{2} \sum_{t=1}^T \nabla_t^\top \Lambda_t^{-1} \nabla_t.$$

From Lemma F.2, we have $\nabla_t \nabla_t^\top \preceq L_{\mathrm{eff}}^2 \mathbf{H}_t^\top \mathbf{H}_t$. Since $\Lambda_t \succ 0$, this implies that $\nabla_t^\top \Lambda_t^{-1} \nabla_t = \langle \nabla_t \nabla_t, \Lambda_t^{-1} \rangle \leq L_{\mathrm{eff}}^2 \langle \mathbf{H}_t^\top \mathbf{H}_t, \Lambda_t^{-1} \rangle = L_{\mathrm{eff}}^2 \mathrm{tr}(\mathbf{H}_t \Lambda_t^{-1} \mathbf{H}_t^\top)$. Thus, by Lemma F.5,

$$\frac{\eta}{2} \sum_{t=1}^T \nabla_t^\top \Lambda_t^{-1} \nabla_t \leq \frac{\eta L_{\mathrm{eff}}^2}{2} \sum_{t=1}^T \mathrm{tr}(\mathbf{H}_t \Lambda_t^{-1} \mathbf{H}_t^\top) \leq \frac{d\eta L_{\mathrm{eff}}^2}{2} \log\left(1 + \frac{TR_H^2}{\lambda}\right). \tag{F.1}$$

$\square$

## F.2 Memory Regret for Known System

In this section, we adress movement costs, thereby proving Theorem 2.1. In what follows, we make the simplifying assumption that $\mathbf{z}_s = \mathbf{z}_1$ for $s \leq 1$. We will remove this assumption at the end of the proof. Our goal is to bound:

$$\begin{aligned}
\mathrm{MemoryReg}_T &:= \sum_{t=1}^T F_t(\mathbf{z}_t, \ldots, \mathbf{z}_{t-h}) - \min_{z \in \mathcal{C}} f_t(z) \\
&= \underbrace{\sum_{t=1}^T F_t(\mathbf{z}_t, \ldots, \mathbf{z}_{t-h}) - f_t(\mathbf{z}_t)}_{(\mathrm{MoveDiff}_T)} + \underbrace{\sum_{t=1}^T f_t(\mathbf{z}_t) - \min_{z \in \mathcal{C}} \sum_{t=1}^T f_t(z)}_{(\mathrm{OcoReg}_T)}.
\end{aligned}$$

The second term is bounded by direct application of Proposition F.1. For the first term, we begin with the following lemma, which shows that the relevant movement cost is only along the $\mathbf{Y}_{t-i}$ directions:

**Lemma F.6** (Movement Cost). *For all $t \geq 1$, we have*

$$|F_t(\mathbf{z}_t, \ldots, \mathbf{z}_{t-h}) - f_t(\mathbf{z}_t)| \leq L_{\mathrm{eff}} R_G \sum_{i=1}^h \|\mathbf{Y}_{t-i}(\mathbf{z}_t - \mathbf{z}_{t-i})\|_2.$$

*Therefore, by the triangle inequality, rearranging summations, and the assumption $\mathbf{z}_s = \mathbf{z}_1$ for $s \leq 1$,*

$$\mathrm{MoveDiff}_T \leq h L_{\mathrm{eff}} R_G \sum_{s=1-h}^T \sum_{i=1}^{h-1} \|\mathbf{Y}_s(\mathbf{z}_{s+i+1} - \mathbf{z}_{s+i})\|_2 \cdot \mathbb{I}_{1 \leq s+i \leq t-1}.$$

Next, let us develop a bound on $\|\mathbf{Y}_s(\mathbf{z}_{t+1} - \mathbf{z}_t)\|_2$:

**Lemma F.7.** *Adopt the convention $\Lambda_s = \Lambda_1$ for $s \leq 0$. Further, consider $s \leq t$, with $t \geq 1$ and $s$ possibly negative. Then, $\|\mathbf{Y}_s(\mathbf{z}_{t+1} - \mathbf{z}_t)\|_2 \leq \eta L_{\mathrm{eff}} \mathrm{tr}(\mathbf{Y}_s \Lambda_s^{-1} \mathbf{Y}_s)^{1/2} \mathrm{tr}(\mathbf{H}_t^\top \Lambda_t^{-1} \mathbf{H}_t))^{1/2}$. Therefore,*

$$\mathrm{MoveDiff}_T \leq \eta h^2 L_{\mathrm{eff}} R_G \cdot \sqrt{\sum_{t=1-h}^T \mathrm{tr}(\mathbf{Y}_t \Lambda_t^{-1} \mathbf{Y}_t)} \cdot \sqrt{\sum_{t=1}^T \mathrm{tr}(\nabla_t^\top \Lambda_t^{-1} \nabla_t)}.$$

Now, we already bounded the sum of the terms $\mathrm{tr}(\nabla_t^\top \Lambda_t^{-1} \nabla_t)$ in Eq. (F.1):

$$\sum_{s=1}^{T} \mathrm{tr}(\nabla_t \Lambda_t^{-1} \nabla_t) \leq dL_{\mathrm{eff}}^2 \log(1 + \frac{TR_H^2}{\lambda}). \tag{F.2}$$

The main technical challenge is to reason about the sum $\mathrm{tr}(\mathbf{Y}_t \Lambda_t^{-1} \mathbf{Y}_t)$. We bound this quantity using the following proposition:

**Proposition F.8.** *Suppose that* $\kappa(G) > 0$*, and define* $c_{\psi;t} := 1 \vee \frac{t\psi_G(h+1)^2}{hR_G^2}$*. Then, for any* $\mathbf{Y}_{1-h}, \mathbf{Y}_{2-h}, \dots, \mathbf{Y}_t$*, the matrices* $\mathbf{H}_s = \sum_{i=0}^{[h]} G^{[i]} \mathbf{Y}_{s-i}$ *satisfy*

$$\sum_{s=1}^{t} \mathbf{H}_s^\top \mathbf{H}_s \succeq \frac{\kappa(G)}{2} \cdot \left( \sum_{s=1-h}^{t} \mathbf{Y}_s^\top \mathbf{Y}_s \right) - 5hR_H^2 c_{\psi;t} I.$$

The above proposition is proved in Appendix H.1. Under the assumption of the theorem, we have $c_{\psi;t} \leq 1$, so $5hR_H^2 c_{\psi;t} \leq 5hR_H^2$. Thus, for $\lambda = 6hR_H^2$, we have $\Lambda_t \geq \frac{\lambda}{6}I + \kappa \sum_{s=1-h}^{t} \mathbf{Y}_s^\top \mathbf{Y}_s$. Note that this holds even for $t \leq 0$, with the above convention $\Lambda_t = \Lambda_1$ for negative $t$. Thus, Lemma F.5 and the simplifications $R_Y \leq R_H$, $\kappa \leq 1$ gives

$$\sum_{s=1-h}^{T} \mathrm{tr}(\mathbf{Y}_t \Lambda_t^{-1} \mathbf{Y}_t) \leq \frac{2d}{\kappa} \log\left( 1 + \frac{6\kappa TR_Y^2}{2\lambda} \right) \leq \frac{2d}{\kappa} \log\left( 1 + \frac{3R_H^2}{\lambda} \right). \tag{F.3}$$

We can now complete the proof of Theorem 2.1.

*Proof of Theorem 2.1.* Combining Lemma F.7, Eqs. (F.2) and (F.3), and finally the unary regret bound from Proposition F.1

$$\mathrm{MoveDiff}_T + \mathrm{OcoReg}_T$$

$$\leq \mathrm{OcoReg}_T + \eta h^2 L_{\mathrm{eff}}^2 R_G \cdot \sqrt{\sum_{t=1-h}^{T} \mathrm{tr}(\mathbf{Y}_t \Lambda_t^{-1} \mathbf{Y}_t)} \cdot \sqrt{\sum_{t=1}^{T} \mathrm{tr}(\mathbf{H}_t^\top \Lambda_t^{-1} \mathbf{H}_t)}.$$

$$\leq \mathrm{OcoReg}_T + \sqrt{\frac{2}{\kappa}} d\eta h^2 L_{\mathrm{eff}}^2 R_G \log(1 + \frac{3TR_H^2}{\lambda}).$$

Finally, since $\lambda = 6hR_H^2$, $\log(1 + \frac{3TR_H^2}{\lambda}) \leq \log(1 + T)$. Thus, combining with the unary regret bound from Proposition F.1,

$$\mathrm{MoveDiff}_T + \mathrm{OcoReg}_T \leq \frac{\lambda D^2}{\eta} + \eta L_{\mathrm{eff}}^2 d \left( \frac{1}{2} + h^2 R_G \sqrt{\frac{2}{\kappa}} \right) \log\left(1 + T\right),$$

To conclude, we use $\eta = \frac{1}{\alpha}$, so that with $\lambda = 6hR_H^2$, yields $\frac{\lambda D^2 R_H^2}{\eta} = 3\alpha R_H^2 D^2$. Moreover, noting $h^2 R_G \sqrt{\frac{1}{\kappa}} \geq 1$[7], we arrive at

$$\mathrm{MemoryReg}_T = \mathrm{MoveDiff}_T + \mathrm{OcoReg}_T \leq 3\alpha D^2 R_H^2 + \frac{2dh^2 L_{\mathrm{eff}}^2 R_G}{\alpha \kappa^{1/2}} \log\left(1 + T\right). \tag{F.4}$$

Recall that the above bound follows under the assumption that $\mathbf{z}_s = \mathbf{z}_1$ for $s \leq 1$. Let us remove this assumption presently. Observe that the iterates $\mathbf{z}_s$ for $s < 1$ *do not* alter the trajector of future iterates $\mathbf{z}_t$ for $t \geq 1$; they only appear in the memory regret bound via the with memory loss $F_t(\mathbf{z}_{t:t-h})$. Thus, introducing $\check{\mathbf{z}}_t := \mathbb{I}(t \geq 1)\mathbf{z}_t + \mathbb{I}(t < 1)\mathbf{z}_1$, imposing the above assumption ($\mathbf{z}_s = \mathbf{z}_1$ for $s \leq 1$) comes at the expense of regret at most

$$\sum_{t=1}^{T} |F_t(\check{\mathbf{z}}_{t:t-h}) - F(\mathbf{z}_{t:t-h})| = \sum_{t=1}^{h} |F_t(\check{\mathbf{z}}_{t:t-h}) - F(\mathbf{z}_{t:t-h})|.$$

With routine computations and the assumption that $L \geq 1$, each term in the above can be bounded by $L_{\mathrm{eff}} \sum_{i=0}^{h} G^{[i]} \|\mathbf{Y}_{t-i}\check{\mathbf{z}}_t - \mathbf{z}_t)\|_2 \leq L_{\mathrm{eff}} R_G R_{Y,\mathcal{C}} \leq L_{\mathrm{eff}}^2$. This contributes a total addition cost of $hL_{\mathrm{eff}}^2$, we which can be absored into the right-most term on Eq. (F.4) at the expense of replacing the constant 2 with a factor of 3. $\qquad\square$

# G   Regret with Quadratic Error Sensitivity (Theorem 2.2)

This section proves Theorem 2.2 and its generalizations. It is organized as follows:

- In Appendix G.1, we two bounds which make explicit a certain negative regret term. Theorem 2.2a gives the generaliztion of Theorem 2.2 in the $\epsilon_G^2 \geq \sqrt{T}$ regime (and allows for slight mis-specification of $\lambda$), and Theorem G.1 proves a guarantee that degrades as $(T\epsilon_G)^{2/3}$ for small $\epsilon_G$. We prove Theorem 2.2a from Theorem G.1 in Appendix G.1.1.
- The remainder of the section is dedicated to the proof of Theorem G.1. This begins with Appendix G.2, which introduces relevant preliminaries.
- Appendix G.3 provides a careful analysis of initial regret terms, and controlling the contribution of errors introduced by using the $\widehat{f}_t$ sequence rather than $f_t$.
- Appendix G.4 details our careful "blocking argument", which we use to offset the errors the terms $\sum_t \|\mathbf{Y}_t(\mathbf{z}_t - z_\star)\|$ from the gradients by a negative terms $\sum_t \|\mathbf{X}_t(\mathbf{z}_t - z_\star)\|_2^2$ that arise in the regret analysis.
- Appendix G.5 concludes the proof of Theorem G.1, bounding first the movement cost and then tuning relevant parameters in the analysis.

All ommitted proofs are provided in Appendix I, organized into subsections and presented in numerical order.

## G.1   Bounds for Unknown Systems with Negative Regret

Here, we provide bounds which explicitly account for an appropriate negative regret term, scaling with $\sum_{t=1}^{T} \|\mathbf{Y}_t(\mathbf{z}_t - z_\star)\|^2$. Specifically, for any fixed comparator $z_\star \in \mathcal{C}$, our goal is to bound

$$\overline{\mathrm{MemoryReg}}_T(\nu; z_\star) := \sum_{t=1}^{T} F_t(\mathbf{z}_{t:t-h}) - f_t(z_\star) + \nu \sum_{t=1}^{T} \|\mathbf{Y}_t(\mathbf{z}_t - z_\star)\|^2, \qquad \text{(G.1)}$$

which gives a negative regret term by re-arranging $\nu \sum_{t=1}^{T} \|\mathbf{Y}_t(\mathbf{z}_t - z_\star)\|^2$ to the right-hand side of the above display. Note that we prove this bound for *any* fixed comparator $z_\star$, not just the "best-in-hindsight" comparator. Moreover, proving this bound for the best-in-hinsight comparator does not imply the bound for all $z_\star \in \mathcal{C}$, because the terms $\delta_t$ in the negative-regret term differ as a function of $z_\star$.

To state our bound on $\overline{\mathrm{MemoryReg}}_T$, we recall the relevant parameter bounds:

**Definition 2.1** (Bounds on Relevant Parameters). We assume $\mathcal{C}$ contains the origin. Further, we define the diameter $D := \max\{\|z - z'\| : z, z' \in \mathcal{C}\}$, $Y$-radius $R_Y := \max_t \|\mathbf{Y}_t\|_{\mathrm{op}}$, and $R_{Y,\mathcal{C}} := \max_t \max_{z \in \mathcal{C}} \|\mathbf{Y}_t z\|$; In the *exact* setting, we define the radii $R_v := \max_t \max\{\|\mathbf{v}_t\|_2\}$ and $R_G := \max\{1, \|G\|_{\ell_1, \mathrm{op}}\}$. In the *approximate* setting, $R_v := \max_t \max\{\|\mathbf{v}_t\|_2, \|\widehat{\mathbf{v}}_t\|_2\}$, $R_G := \max\{1, \|G\|_{\ell_1, \mathrm{op}}, \|\widehat{G}\|_{\ell_1, \mathrm{op}}\}$; For settings, we define the $H$-radius $R_H = R_G R_Y$, and define the effective Lipschitz constant $L_{\mathrm{eff}} := L \max\{1, R_v + R_G R_{Y,\mathcal{C}}\}$.

Our main result in this section is as follows. We also allow $\lambda$ to be slightly under-specified. This show's relative insensitivity to the selection of $\lambda$, and is also useful when porting the bound over to the control setting:

**Theorem 2.2a.** *Consider the setting of Theorem 2.2, but where instead $\lambda \in (c_\lambda, 1] \cdot (T\epsilon_G^2 + hR_G^2)$ for $c_\lambda \in (0, 1]$. Equivalently, consider the setting of Theorem G.1 below, but with the additional conditions $\epsilon_G \geq \sqrt{T}$ and $\beta = L$. Then for any $z_\star \in \mathcal{C}$,*

$$c_\lambda \overline{\mathrm{MemoryReg}}_T(\nu_\star; z_\star) \lesssim \log(1 + \frac{T}{c_\lambda}) \left(\frac{C_1}{\alpha \kappa^{1/2}} + C_2\right) \left(T\epsilon_G^2 + h^2(R_G^2 + R_Y)\right),$$

*where $C_1 := (1 + R_Y)R_G(h + d)L_{\mathrm{eff}}^2$, $C_2 := (L^2 c_v^2/\alpha + \alpha D^2)$, and $\nu_\star = \frac{\alpha\sqrt{\kappa}}{48(1+R_Y)}$.*

Theorem 2.2 is an immediate conseuqnece of Theorem 2.2a. We prove the above guarantee from a more statement, which allows for $\epsilon_G^2 \leq \sqrt{T}$ as well.

**Granular Guarantee for** Semi-**ONS with errors**    To state our generic guarantee, we specify the following constants:

**Definition G.1** (Constants for Unknown $G$ Regret Analysis)**.** We define the constants We begin by establishing a slight generalization of Theorem 2.2a, accomodating arbitrarily small. To start, define the constants

$$C_{\mathrm{mid}} := (1 + \tfrac{\beta^2}{L^2})(1 + R_Y)hL_{\mathrm{eff}}^2 + \beta^2\sqrt{\kappa}c_v^2 + \alpha^2\sqrt{\kappa}D^2 \tag{G.2}$$

$$C_{\mathrm{hi}} := (1 + R_Y)L_{\mathrm{eff}}R_G^2 R_{Y,\mathcal{C}}(h+d) + \alpha D^2 \tag{G.3}$$

$$C_{\mathrm{low}} := (1 + R_Y)^2 R_G h^2 \cdot dL_{\mathrm{eff}}^2. \tag{G.4}$$

$$\nu_\star = \frac{\alpha\sqrt{\kappa}}{48(1 + R_Y)}\min\left\{4(1 + R_Y)(T\epsilon_G^4)^{1/3}, 1\right\} \tag{G.5}$$

Finally, we define a logarithmic factor

$$\mathfrak{L} := \log(1 + R_H^2 T/\lambda), \quad \text{with } \mathfrak{L} \le \log(1 + T) \text{ for } \lambda \ge R_H^2. \tag{G.6}$$

Our more granular result is the following:

**Theorem G.1** (Granular Regret Guarantee for Semi-ONS on an unknown system)**.** *Consider running* Semi-ONS *on the empirical loss sequence* $(\widehat{f}_t, \widehat{\mathbf{H}}_t)$. *Suppose that*

- *The losses* $\ell_t$ *are* $L$-*subquadratic and* $\alpha$-*strongly convex for* $L \ge 1 \vee \alpha$ *(Assumption 1), and are* $\beta$ *smooth* $(\nabla^2\ell_t \preceq \beta I)$

- *Suppose that* $\|\widehat{G} - G_\star\|_{\ell_1,\mathrm{op}} \le \epsilon_G$, $\widehat{G}^{[i]} = 0$ *for* $i > h$, *and* $\max_{t \ge 1}\|\mathbf{v}_t - \widehat{\mathbf{v}}_t\|_2 \le c_v\epsilon_G$ *for some constant* $c_v \ge 0$.

- *The step size is* $\eta = 3/\alpha$, *and* $\lambda$ *lies in* $\lambda \in [c_\lambda, 1]\left(T\epsilon_G^2 + (T\epsilon_G)^{2/3} + hR_G^2\right)$ *for some* $c_\lambda \in (0, 1]$.

- *All relevant quantities are bounded as in Definition 2.1*

*Then, the memory regret on the true loss sequence* $(f_t, \mathbf{H}_t)$ *is bounded by*

$$c_\lambda\overline{\mathrm{MemoryReg}}_T(\nu_\star; z_\star) \lesssim C_{\mathrm{hi}}(T\epsilon_G)^{2/3}\mathfrak{L} + \frac{C_{\mathrm{mid}}T\epsilon_G^2}{\alpha\sqrt{\kappa}} + \frac{C_{\mathrm{low}}\mathfrak{L}}{\alpha\sqrt{\kappa}} + \alpha hR_G^2 D^2.$$

Observe that, when $\epsilon_G^2 \ge \sqrt{T}$, the dominating term is $T\epsilon_G^2$. However, for $\epsilon \le \sqrt{T}$, the term $(T\epsilon_G)^{2/3}$ dominates.

### G.1.1    Proof of Theorem 2.2a from from Theorem G.1

Theorem 2.2a follows from the granular Theorem G.1 as a consequence of the following tedious simplifications. Recall that Theorem 2.2 adds the assumptions that $\epsilon_G^2 \ge \sqrt{T}$, and $\beta = L$. This enables the following simplifications. First, since $(T\epsilon_G^4)^{1/3} \ge 1$ we can take $\nu_\star = \frac{\alpha\sqrt{\kappa}}{48(1+R_Y)}$, which is precisely the value of $\nu$ used in the theorem. Second, we have $(T\epsilon_G)^{2/3}/T\epsilon_G^2 = 1/(T\epsilon_G^4)^{1/3} \le 1$. This means that the choice of $\lambda = c_\lambda(T\epsilon_G^2 + hR_G^2)$ is valid for Theorem G.1, up to rescaling $c_\lambda$ by a factor of 2. Thus, we have

$$c_\lambda\overline{\mathrm{MemoryReg}}_T\left(\tfrac{\alpha\sqrt{\kappa}}{48(1+R_Y)}; z_\star\right) \lesssim C_{\mathrm{hi}}(T\epsilon_G)^{2/3}\mathfrak{L} + \frac{C_{\mathrm{mid}}(T\epsilon_G^2)}{\alpha\sqrt{\kappa}} + \frac{C_{\mathrm{low}}\mathfrak{L}}{\alpha\sqrt{\kappa}} + \alpha hR_G^2 D^2$$

$$\lesssim \frac{\mathfrak{L}}{\alpha\sqrt{\kappa}}\left((T\epsilon_G^2)(C_{\mathrm{hi}}\alpha\sqrt{\kappa} + C_{\mathrm{mid}}) + C_{\mathrm{low}} + \alpha^2\sqrt{\kappa}hR_G^2 D^2\right).$$

First, let us simplify $C_{\mathrm{hi}}\alpha\sqrt{\kappa} + C_{\mathrm{mid}}$. Using the simplifying condition $\beta = L$, and using $R_G R_{Y,\mathcal{C}} \le L_{\mathrm{eff}}$ (again, $L \ge 1$), we have

$$C_{\mathrm{hi}}\alpha\sqrt{\kappa} + C_{\mathrm{mid}} \lesssim (1 + R_Y)(hL_{\mathrm{eff}}^2 + L_{\mathrm{eff}}R_G^2 R_{Y,\mathcal{C}}(h+d)) + L^2\sqrt{\kappa}c_v^2 + \alpha^2\sqrt{\kappa}D^2$$

$$\lesssim (1 + R_Y)R_G(h + d)L_{\mathrm{eff}}^2 + L^2\sqrt{\kappa}c_v^2 + \alpha^2\sqrt{\kappa}D^2.$$

Hence,

$$
\begin{aligned}
&(C_{\mathrm{hi}}\alpha\sqrt{\kappa} + C_{\mathrm{mid}})T\epsilon_G^2 + C_{\mathrm{low}} + \alpha^2\sqrt{\kappa}hR_G^2D^2 \\
&\lesssim (1+R_Y)R_G(h+d)L_{\mathrm{eff}}^2(T\epsilon_G + (1+R_Y)h^2) + (L^2\sqrt{\kappa}c_v^2 + \alpha^2\sqrt{\kappa}D^2)(T\epsilon_G^2 + hR_G^2) \\
&\lesssim C_1(T\epsilon_G^2 + (1+R_Y)h^2) + \alpha\sqrt{\kappa}C_2(T\epsilon_G^2 + hR_G^2) \\
&\lesssim (C_1\alpha\sqrt{\kappa}C_2)(T\epsilon_G^2 + (1+R_Y)h^2 + hR_G^2) \\
&\lesssim (C_1\alpha\sqrt{\kappa}C_2)(T\epsilon_G^2 + h^2R_Yh^2 + R_G^2)
\end{aligned}
$$

for $C_1 := (1+R_Y)R_G(h+d)L_{\mathrm{eff}}^2$ and $C_2 := (L^2\alpha^{-1}c_v^2 + \alpha D^2)$. Thus we conclude that

$$
\overline{\mathrm{MemoryReg}}_T\left(\tfrac{\alpha\sqrt{\kappa}}{48(1+R_Y)}; z_\star\right) \lesssim c_\lambda^{-1}\log(1+T)\left(\frac{C_1}{\alpha\kappa^{1/2}} + C_2\right)\left(T\epsilon_G^2 + h^2(R_G^2 + R_Y)\right),
$$

as needed.

## G.2 Preliminaries for Proof of Theorem G.1

**Notation:** Let us begin by introducing relevant notation. Set $\nabla_t = \nabla f_t(\mathbf{z}_t)$ to denote the gradients of the true counterfactual stationary counterfactual costs $f_t$, and let $\hat{\nabla}_t := \nabla\hat{f}_t(\mathbf{z}_t)$ denote the gradient of their approximations. Analogously, define the matrices

$$
\widehat{\Lambda}_t = \lambda I + \sum_{t=1}^{T}\widehat{\mathbf{H}}_t^\top\widehat{\mathbf{H}}_t, \quad \Lambda_t = \lambda I + \sum_{t=1}^{T}\mathbf{H}_t^\top\mathbf{H}_t
$$

For $t \le 1$, we will use the conventions $\Lambda_t = \Lambda_1$ and $\widehat{\Lambda}_t = \widehat{\Lambda}_1$. Throughout, we fix an *arbitrary* comparator $z_\star \in \mathcal{C}$, and further introduce the notation

$$
\boldsymbol{\delta}_t := \mathbf{z}_t - z_\star, \quad \mathbf{err}_t = \hat{\nabla}_t - \nabla_t
$$

to denote the difference of $\mathbf{z}_t$ from the comparator, and difference between gradients, respectively.

We recall that $\lambda, \eta$ are the algorithm parameters dictating the magnitude of the regularizer in $\Lambda_t$, and step size, respectively. We will also introduce a "blocking parameter" $\tau$, whose purposes is described at length in Appendix G.4. For simplicity, most of the proof will focuses on the unary regret analogue of $\overline{\mathrm{MemoryReg}}_T$, defined as follows:

$$
\overline{\mathrm{OcoReg}}_T(\nu; z_\star) := \sum_{t=1}^{T} f_t(\mathbf{z}_t) - f_t(z_\star) + \nu\sum_{t=1}^{T}\|\mathbf{Y}_t\boldsymbol{\delta}_t\|^2, \quad \boldsymbol{\delta}_t := \mathbf{z}_t - z_\star, \tag{G.7}
$$

We extend to memory regret in Appendix G.5. denote a logarithmic factor that will appear throughout.

**Reduction $\mathbf{z}_s = \mathbf{z}_1$ for $s \le 1$:** As in the proof of Theorem 2.1 in Appendix F.2, we can assume that $\mathbf{z}_s = \mathbf{z}_1$, at the expense of an additional factor of $hL_{\mathrm{eff}}^2$ in the regret. This term is dominated by the factor of $C_{\mathrm{low}}\mathfrak{L}$ in Theorem G.1, and can thus be disregarded in the following argument.

## G.3 Bounding Regret in Terms of Error

We begin with the following basic regret bound, controls the excess regret of using inexact gradients compared to standard bounds from online Newton.

**Lemma G.1.** *Let $\lambda \ge 1$. Then regret on measured on the $f_t^{\mathrm{pred}}$ sequence is bounded by*

$$
\sum_{t=1}^{T} f_t(\mathbf{z}_t) - f_t(z_\star) \le \sum_{t=1}^{T}\mathbf{err}_t^\top\boldsymbol{\delta}_t + \frac{1}{2\eta}\sum_{t=1}^{T}(\|\widehat{\mathbf{H}}_t\boldsymbol{\delta}_t\|^2 - \eta\alpha\|\mathbf{H}_t\boldsymbol{\delta}_t\|^2) + \widehat{\mathrm{Reg}}_T,
$$

*where $\widehat{\mathrm{Reg}}_T := \frac{\eta dL_{\mathrm{eff}}^2\mathfrak{L}}{2} + \frac{\lambda D^2}{2\eta}$ arises from the regret bound in Proposition F.1, and we recall $\mathfrak{L} := \log(e + TR_H^2)$.*

Next, let us turn to bounding the mismatch arising from the terms $\sum_{t=1}^{T}(\|\widehat{\mathbf{H}}_t\boldsymbol{\delta}_t\|^2 - \eta\alpha\|\mathbf{H}_t\boldsymbol{\delta}_t\|^2)$:

**Lemma G.2.** *For $\eta \geq \frac{3}{\alpha}$, we have $\|\widehat{\mathbf{H}}_t\boldsymbol{\delta}_t\|^2 - \eta\alpha\|\mathbf{H}_t\boldsymbol{\delta}_t\|^2 \leq -\|\mathbf{H}_t\boldsymbol{\delta}_t\|^2 + 8R_{Y,\mathcal{C}}^2\epsilon_G^2$. Hence, we have the regret bound:*

$$\sum_{t=1}^{T} f_t(\mathbf{z}_t) - f_t(z_\star) \leq \sum_{t=1}^{T}\mathbf{err}_t^\top\boldsymbol{\delta}_t - \frac{1}{2\eta}\sum_{t=1}^{T}\|\mathbf{H}_t\boldsymbol{\delta}_t\|^2 + \frac{4}{\eta}TR_{Y,\mathcal{C}}^2\epsilon_G^2 + \widehat{\mathrm{Reg}}_T.$$

### G.3.1 Controlling the error contributions

Next, we turn to bounding the contribution of the error in estimating the gradient:

**Lemma G.3.** *There exists $g_{1,t}$ and $g_{2,t}$ with $\|g_{1,t}\|_2 \leq L_{\mathrm{eff}}$ and $\|g_{2,t}\| \leq \beta\epsilon_G(c_v + 2R_{Y,\mathcal{C}})$ such that*

$$\mathbf{err}_t = (\widehat{\mathbf{H}}_t - \mathbf{H}_t)^\top g_{1,t} + \mathbf{H}_t^\top g_{2,t}.$$

By leveraring the specific structure of $\mathbf{err}_t$, we obtain:

**Lemma G.4.** *For $\eta \geq \frac{3}{\alpha}$, the following regret bound holds for all $z_\star \in \mathcal{C}$ and all $\nu > 0$:*

$$\sum_{t=1}^{T} f_t(\mathbf{z}_t) - f_t(z_\star) \leq \frac{1}{4\eta}\sum_{t=1}^{T}\left(\frac{\nu}{h+1}\sum_{i=0}^{h}\|\mathbf{Y}_{t-i}\boldsymbol{\delta}_t\|^2 - \|\mathbf{H}_t\boldsymbol{\delta}_t\|^2\right) + T\epsilon_G^2 \cdot \mathrm{ERR}(\nu) + \widehat{\mathrm{Reg}}_T,$$

$$(\mathrm{G.8})$$

*where* $\mathrm{ERR}(\nu) := \left(\frac{\eta(h+1)L_{\mathrm{eff}}^2}{\nu} + \eta\beta^2(c_v + 2R_{Y,\mathcal{C}})^2 + \frac{4R_{Y,\mathcal{C}}}{\eta}\right)$.

As a consequence, we have

$$\overline{\mathrm{OcoReg}}_T\left(\frac{\nu}{4\eta}; z_\star\right) \leq \frac{1}{4\eta}\sum_{t=1}^{T}\left(\nu\|\mathbf{Y}_t\boldsymbol{\delta}_t\| + \frac{\nu}{h+1}\sum_{i=0}^{h}\|\mathbf{Y}_{t-i}\boldsymbol{\delta}_t\|^2 - \|\mathbf{H}_t\boldsymbol{\delta}_t\|^2\right) + T\epsilon_G^2\,\mathrm{ERR}(\nu) + \widehat{\mathrm{Reg}}_T,$$

$$(\mathrm{G.9})$$

## G.4 The 'blocking argument'

A this stage of the proof, the main challenge is to show that for some small constant $\nu$, the terms $\|\widehat{\mathbf{Y}}_{t-i}\boldsymbol{\delta}_t\|^2$ in Eq. (G.9) are offset by $\|\mathbf{H}_t\boldsymbol{\delta}_t\|^2$ on aggregate. We do this by dividing times into "blocks" of size $\tau = \Theta(\sqrt{T})$, centering at the terms $\boldsymbol{\delta}_t$ at times $t = k_j + 1$, for indices $k_j$ defined below. We define $j_{\max} := \lfloor T/\tau \rfloor$ as the number of blocks. We then argue that, within any block

$$\sum_{t \text{ in block } j}\|\mathbf{H}_t\boldsymbol{\delta}_t\|^2 \gtrsim \sum_{i=0}^{h}\sum_{t \text{ in block } j}\frac{1}{\nu}\|\widehat{\mathbf{Y}}_{t-i}\boldsymbol{\delta}_t\|^2 + \mathcal{O}(1) \tag{G.10}$$

for appropriate $\nu$ and block size $\tau$. The reason we should expect an inequality of the above form to holds is that, from adapting Proposition F.8, we have the inequality that

$$\sum_{t \text{ in block } j}\mathbf{H}_t\mathbf{H}^\top \gtrsim \sum_{t \text{ in block } j}\mathbf{Y}_t\mathbf{Y}_t^\top - \mathcal{O}(1) \cdot I, \tag{G.11}$$

However, Eq. (G.11) does not directly imply a bound of the form Eq. (G.10), beacuse the vectors $\boldsymbol{\delta}_t$ differ for each $t$. Instead, we 're-center' the $\boldsymbol{\delta}_t$ terms in the sum $\boldsymbol{\delta}_t = \boldsymbol{\delta}_{k_j+1}$, and at argue

$$\sum_{t \text{ in block } j}\|\mathbf{H}_t\boldsymbol{\delta}_{k_j+1}\|^2 \approx \sum_{i=0}^{h}\sum_{t \text{ in block } j}\frac{1}{\nu}\|\mathbf{Y}_{t-i}\boldsymbol{\delta}_{k_j+1}\|^2 - \mathcal{O}(1). \tag{G.12}$$

The above bound can be established from an estimate of the form Eq. (G.11). Summing this up across all $j_{\max}$ blocks, we see that the negative regret from the terms $\|\mathbf{H}_t\boldsymbol{\delta}_{k_j+1}\|^2$ cancels the regret from the terms $\|\mathbf{Y}_{t-i}\boldsymbol{\delta}_{k_j+1}\|^2$. Accounting for all $j_{\max} = \Theta(T/\tau)$ blocksgives

$$\sum_{j=1}^{j_{\max}}\sum_{t \text{ in block } j}\|\mathbf{H}_t\boldsymbol{\delta}_{k_j+1}\|^2 \approx \sum_{j=1}^{j_{\max}}\sum_{i=0}^{h}\sum_{t \text{ in block } j}\frac{1}{\nu}\|\mathbf{Y}_{t-i}\boldsymbol{\delta}_{k_j+1}\|^2 - \mathcal{O}(T/\tau). \tag{G.13}$$

incurring an additive factor of $T/\tau$, favoring larger block sizes $\tau$.

But, we must also argue that not too much is lost by approximating the statement Eq. (G.10) with the centered analogue Eq. (G.13). The cost of recentering will ultimatels as $\mathcal{O}(\tau)$, so trading off $\tau$ with the bound of yields $\sqrt{T}$ regret in the final bound.

Interestingly, the cost of recentering is intimately tied to bounding the movement of the iterates $\mathbf{z}_t$. Thus, we find that the same properties that allow Semi-ONS to attain logarithmic regret for the known system case are also indispensible in achieving low sensitivity to error in the unknown system case.

### G.4.1 Formalizing the blocking argument

Formally, the cost of the above re-centering argument is captured by the following lemma:

**Lemma G.5** (Blocking Argument). *Given parameter $\tau \in \mathbb{N}$, and introduce the $k_j = \tau(j-1)$, and $j_{\max} := \lfloor T/\tau \rfloor$. Then, with the understanding that $\mathbf{z}_s = 0$ for $s \le 1$, the following holds for all $i \in [h]$,*

$$\sum_{t=1}^{T} \|\mathbf{Y}_{t-i}\boldsymbol{\delta}_t\|_2^2 \le 4\tau R_{Y,\mathcal{C}} + \sum_{j=1}^{j_{\max}}\sum_{s=1}^{\tau} \|\mathbf{Y}_{k_j+s-i}\boldsymbol{\delta}_{k_j+1}\|_2^2 + 4R_{Y,\mathcal{C}}\sum_{t=1}^{T}\sum_{s=0}^{\tau-1}\|\mathbf{Y}_{t-i}(\mathbf{z}_{t-s}-\mathbf{z}_{t-s-1})\|_2.$$

$$\sum_{t=1}^{T} \|\mathbf{H}_{i-h}\boldsymbol{\delta}_t\|_2^2 \ge \sum_{j=1}^{j_{\max}}\sum_{s=1}^{\tau} \|\mathbf{H}_{k_j+s}\boldsymbol{\delta}_{k_j+1}\|_2^2 - 4R_{Y,\mathcal{C}}R_G\sum_{t=1}^{T}\sum_{s=0}^{\tau-1}\|\mathbf{H}_t(\mathbf{z}_{t-s}-\mathbf{z}_{t-s-1})\|_2,$$

Notice that, while the left-hand side depends on $\boldsymbol{\delta}_t$, the right hand side is 'centered' at $\boldsymbol{\delta}_{k_j+1}$ for $j \in [j_{\max}]$, at the expense of movement penalties on $\mathbf{z}_{t-s} - \mathbf{z}_{t-s-1}$. Let us re-write the above bound to give a useful regret decomposition. We introduce bounding terms $\mathrm{Reg}_{Y,\mathrm{move},i}$ and $\mathrm{Reg}_{H,\mathrm{move}}$ for the movement costs above associated with the centering argument, and $\mathrm{Reg}_{\mathrm{cancel}}$ associated with the offsetting argument described above. Formally,

$$\mathrm{Reg}_{Y,\mathrm{move},i} := \sum_{t=1}^{T}\sum_{s=0}^{\tau-1}\|\mathbf{Y}_{t-i}(\mathbf{z}_{t-s}-\mathbf{z}_{t-s-1})\|_2.$$

$$\mathrm{Reg}_{H,\mathrm{move}} := \sum_{t=1}^{T}\sum_{s=0}^{\tau-1}\|\mathbf{H}_t(\mathbf{z}_{t-s}-\mathbf{z}_{t-s-1})\|_2$$

$$\mathrm{Reg}_{\mathrm{cancel}} := \sum_{j=1}^{j_{\max}}\sum_{s=1}^{\tau}\left(\sum_{i=0}^{h}\left(\nu\left(\frac{1}{h+1}+\mathbb{I}_{i=0}\right)\|\mathbf{Y}_{k_j+s-i}\boldsymbol{\delta}_{k_j+1}\|_2^2\right) - \|\mathbf{H}_{k_j+s}\boldsymbol{\delta}_{k_j+1}\|_2^2\right).$$

Then, from Lemma G.5, the upper bound on $\overline{\mathrm{OcoReg}}_T$ in Eq. (G.9) can be expressed as

$$\overline{\mathrm{OcoReg}}_T\left(\frac{\nu}{4\eta};z_\star\right) \le \frac{1}{4\eta}\mathrm{Reg}_{\mathrm{block}} + T\epsilon_G^2\,\mathrm{ERR}(\nu) + \widehat{\mathrm{Reg}}_T, \tag{G.14}$$

where we define and bound

$$\mathrm{Reg}_{\mathrm{block}} := \sum_{t=1}^{T}\left(\nu\|\mathbf{Y}_t\boldsymbol{\delta}_t\| + \frac{\nu}{h+1}\sum_{i=0}^{h}\|\mathbf{Y}_{t-i}\boldsymbol{\delta}_t\|^2 - \|\mathbf{H}_t\boldsymbol{\delta}_t\|^2\right)$$

$$\le 8\tau \cdot \nu R_{Y,\mathcal{C}} + 8\nu R_{Y,\mathcal{C}}\left(\max_{i\in[h]}\mathrm{Reg}_{Y,\mathrm{move},i}\right) + 4R_{Y,\mathcal{C}}R_G \cdot \mathrm{Reg}_{H,\mathrm{move}} + \mathrm{Reg}_{\mathrm{cancel}}.$$
$$\tag{G.15}$$

Thus, we shall conclude our argument by developing bounds on $\mathrm{Reg}_{Y,\mathrm{move},i}$, $\mathrm{Reg}_{H,\mathrm{move}}$ and $\mathrm{Reg}_{\mathrm{cancel}}$.

**Movement Costs** Via Eq. (G.15) and the definitions of $\mathrm{Reg}_{Y,\mathrm{move},i}$ and $\mathrm{Reg}_{H,\mathrm{move}}$, the cost of the re-centering argument is given by a movement costs, which we bound presently. Since the

movement of the algorithm are small in the norms induced by the preconditioning matrices $\widehat{\Lambda}$, our main argument invokes steps of the form

$$\|\mathbf{H}_t(\mathbf{z}_{t-s} - \mathbf{z}_{t-s-1})\|_2 \leq \frac{\|\mathbf{H}_t^\top \widehat{\Lambda}_{t-s-1}^{-1} \mathbf{H}_t\|_{\mathrm{op}}^2}{2} + \frac{\|(\mathbf{z}_{t-s} - \mathbf{z}_{t-s-1})\|_{\widehat{\Lambda}_{t-s-1}}^2}{2},$$

much like the regret analysis in the known system case. Moreover, the contribuitons of the $\|(\mathbf{z}_{t-s} - \mathbf{z}_{t-s-1})\|_{\widehat{\Lambda}_{t-s-1}}^2$ can be bounded via an application of the log-det potential argument, as in [Proposition F.1](#).

However, we observe that the conditioning of the relevant movement costs is in terms of the $\widehat{\Lambda}$ matrix. To bound terms $\|\mathbf{H}_t^\top \widehat{\Lambda}_{t-s-1}^{-1} \mathbf{H}_t\|_{\mathrm{op}}^2$, we will need to relate the matrices $\widehat{\Lambda}_{t-s-1}$, constructed based on the estimated sequence $(\widehat{\mathbf{H}}_t)$, and with delays up to $(s+1) = \tau$, to the matrixes $\Lambda_t$, based on $(\mathbf{H}_t)$ and current time $t$. This is accomplished by the following lemma:

**Lemma G.6.** *For $c_\lambda \in (0,1]$, set $c_\Lambda(\tau) := 2(1 + R_Y) + 2c_\lambda^{-\frac{1}{2}} R_Y \sqrt{\frac{\tau R_G^2}{\lambda}}$. Then, for $\lambda \geq c_\lambda T \epsilon_G^2$, we have that for all $t \in [T]$,*

$$\widehat{\Lambda}_{t-\tau}^{-1} \preceq c_\Lambda(\tau)^2 \Lambda_t^{-1},$$

*where we adopt the convention $\widehat{\Lambda}_s = \widehat{\Lambda}_1$ and $\Lambda_s = \Lambda_1$ for $s \leq 1$.*

For our scalings of $\tau$ and $\lambda$, $c_\Lambda$ will be roughly constant in magnitude. With the above lemma in hand, we show that the movement terms from the blocking argument scale proportionally to $\tau$.

**Lemma G.7.** *Recall the logarithmic factor $\mathfrak{L} := \log(e + TR_H^2)$. If $\lambda$ is chosen such that $\lambda \geq \frac{c_\lambda}{h} T \epsilon_G^2 + c_\lambda h R_G^2$, then the movement terms admit the following bounds for $i \in \{0, \dots, h\}$:*

$$\mathrm{Reg}_{Y,\mathrm{move},i} := \sum_{t=1}^{T} \sum_{s=0}^{\tau-1} \|\mathbf{Y}_{t-i}(\mathbf{z}_{t-s} - \mathbf{z}_{t-s-1})\|_2 \leq \tau c_\Lambda c_\lambda^{-\frac{1}{2}} \cdot dL_{\mathrm{eff}} \sqrt{\frac{2(1 + 10R_Y^2)}{\kappa}} \mathfrak{L}.$$

$$\mathrm{Reg}_{H,\mathrm{move}} := \sum_{t=1}^{T} \sum_{s=0}^{\tau-1} \|\mathbf{H}_t(\mathbf{z}_{t-s} - \mathbf{z}_{t-s-1})\|_2 \leq \tau c_\Lambda c_\lambda^{-\frac{1}{2}} \cdot dL_{\mathrm{eff}} \mathfrak{L}$$

**Cancellation within blocks** Next, let us argue that the term $\mathrm{Reg}_{\mathrm{cancel}}$ is small, which leverages cancellation within blocks. As per the proof sketch at the beginning of the section, we show that the terms $\|\mathbf{Y}_{k_j+s-i}\boldsymbol{\delta}_{k_j+1}\|_2^2$ offset the terms $\|\mathbf{H}_{k_j+s}\boldsymbol{\delta}_{k_j+1}\|_2^2$ up to a $\mathcal{O}(1)$ factor for each $j$, incuring an error scaling as $j_{\max} \approx T/\tau$ (thereby inducing a trade-off on the parameter $\tau$):

**Lemma G.8.** *For $\nu \leq \frac{\kappa}{4}$, we have*

$$\mathrm{Reg}_{\mathrm{cancel}} \leq \frac{20T}{\tau} \cdot \nu h R_G^2 R_{Y,\mathcal{C}}^2 + 5T \epsilon_G^2 \cdot \kappa R_{Y,\mathcal{C}}^2.$$

### G.4.2 Summarizing the blocking argument

Grouping all the terms that have emerged thus far, we summarize the current state of our argument in the following lemma:

**Lemma G.9.** *Assuming $L_{\mathrm{eff}} \geq 1$, $\nu \leq \frac{\sqrt{\kappa}}{4(1+R_Y)}$, and $\lambda \geq c_\lambda(\frac{1}{h} T \epsilon_G^2 + h R_G^2 + \tau)$, we have that for all $z_\star \in \mathcal{C}$,*

$$c_\lambda \overline{\mathrm{OcoReg}}_T \left(\frac{\nu}{4\eta}; z_\star\right) \lesssim \frac{T \epsilon_G^2}{\alpha} \cdot \left(\frac{h L_{\mathrm{eff}}^2}{\nu} + \beta^2(c_v^2 + R_{Y,\mathcal{C}} + R_{Y,\mathcal{C}}^2)\right) + \widehat{\mathrm{Reg}}_T.$$

$$+ \frac{T\nu}{\tau} \cdot \left(\alpha h R_G^2 R_{Y,\mathcal{C}}^2\right) + \tau \cdot \left(\alpha(1 + R_Y) R_{Y,\mathcal{C}} R_G^2 \cdot dL_{\mathrm{eff}} \mathfrak{L}\right),$$

Let us take stock of what we have so far. The bound $\overline{\mathrm{OcoReg}}_T(\nu/4\eta; z_\star)$ has four components:

- $\widehat{\mathrm{Reg}}_T$, which accounts for the regret on the $\widehat{f}_t$ sequence.

- A term scaling with $T\epsilon_G^2$, which accounts for the sensitivity to error. This term also involves the offset $\nu$.
- A term scaling as $\frac{T\nu}{\tau}$, yielding a penalty for the number of blocks in the blocking argument.
- A term scaling as linearly in $\tau$, arising from the movement costs from the recentering argument.

The final regret bound will follow from carefully trading off the parameters $\nu$ and $\tau$ in the analysis, and from setting $\lambda$ appropriately. Before continuing, we first adress with "with-memory" portion of the bound, passing from unary regret to memory regret.

### G.5 Concluding the Bound

Before concluding the bound, we need to bound the movment cost that appears:

**Lemma G.10** (Movement Cost: Unknown System). *Under the conditions of Lemma G.9,*

$$\text{MoveDiff}_T := \sum_{t=1}^{T} F_t(\mathbf{z}_{t:t-h}) - f_t(\mathbf{z}_t) \le 9\eta\kappa^{-\frac{1}{2}}(1+R_Y)^2 R_G h^2 \cdot dL_{\text{eff}}^2 \mathfrak{L}$$

We are now ready to prove our main theorem:

*Proof of Theorem G.1.* Let us begin by unpacking

$$\widehat{\text{Reg}}_T + \text{MoveDiff}_T \le 9\eta\kappa^{-\frac{1}{2}}(1+R_Y)^2 R_G h^2 \cdot dL_{\text{eff}}^2 \mathfrak{L} + \frac{\eta dL_{\text{eff}}^2 \mathfrak{L}}{2} + \frac{\lambda D^2}{2\eta}$$

$$\lesssim \frac{1}{\alpha\sqrt{\kappa}}(1+R_Y)^2 R_G h^2 \cdot dL_{\text{eff}}^2 \mathfrak{L} + \alpha\lambda D^2,$$

where we use $\eta = \frac{3}{\alpha}$. Thus, from Lemma G.9, the term $\overline{\text{MemoryReg}}_T$ defined in Eq. (G.1) satisfies the following for any $z_\star \in \mathcal{C}$, provided that the conditions of Lemma G.9 hold:

$$c_\lambda \overline{\text{MemoryReg}}_T \left(\frac{\nu}{4\eta}; z_\star\right) \le c_\lambda \overline{\text{OcoReg}}_T \left(\frac{\nu}{4\eta}; z_\star\right) + \text{MoveDiff}_T$$

$$\lesssim \frac{T\epsilon_G^2}{\alpha} \cdot \left(\frac{hL_{\text{eff}}^2}{\nu} + \beta^2(c_v^2 + R_{Y,\mathcal{C}} + R_{Y,\mathcal{C}}^2)\right) + \frac{1}{\alpha\sqrt{\kappa}}(1+R_Y)^2 R_G h^2 \cdot dL_{\text{eff}}^2 \mathfrak{L} + \alpha\lambda D^2$$

$$+ \frac{T\nu}{\tau} \cdot \left(\alpha h R_G^2 R_{Y,\mathcal{C}}^2\right) + \tau \cdot \left(\alpha(1+R_Y)R_{Y,\mathcal{C}} R_G^2 \cdot dL_{\text{eff}}\mathfrak{L}\right),$$

where above we use $c_\lambda \le 1$. Let us now specialize parameters. As per our theorem, we take

$$\lambda = c_\lambda \left(T\epsilon_G^2 + c(T\epsilon_G)^{2/3} + hR_G^2\right), \quad \tau = (T\epsilon_G)^{2/3}, \quad c_\lambda \in (0,1)$$

which we verify satisfies the condition on $\lambda$ placed by Lemma G.9. For this choice of parameters, we have

$$\overline{\text{MemoryReg}}_T \left(\frac{\nu}{4\eta}; z_\star\right) \lesssim \frac{1}{\alpha\sqrt{\kappa}}(1+R_Y)^2 R_G h^2 \cdot dL_{\text{eff}}^2 \mathfrak{L} + \alpha h R_G^2 D^2$$

$$+ \frac{T\epsilon_G^2}{\alpha} \cdot \left(\beta^2(c_v^2 + R_{Y,\mathcal{C}} + R_{Y,\mathcal{C}}^2) + \alpha^2 D^2\right)$$

$$+ \alpha(T\epsilon_G)^{2/3} \cdot \left(D^2 + (1+R_Y)R_{Y,\mathcal{C}} R_G^2 \cdot dL_{\text{eff}}\mathfrak{L}\right)$$

$$+ \frac{T\epsilon_G^2}{\alpha} \cdot \frac{hL_{\text{eff}}^2}{\nu} + \frac{T\nu}{\tau} \cdot \left(\alpha h R_G^2 R_{Y,\mathcal{C}}^2\right).$$

Next, let's tune $\nu$. Define $\nu_0 := \frac{\sqrt{\kappa}}{4(1+R_Y)}$ to denote the upper bound on $\nu$ imposed by Lemma G.9. Moreover, let $\nu_1$ denote the value of $\nu$ that minimizes the upper bound above, namely

$$\nu_1 = \left(\frac{T\epsilon_G^2}{\alpha} \cdot hL_{\text{eff}}^2\right)^{1/2} \cdot \left(\frac{T}{\tau} \cdot \alpha h R_G^2 R_{Y,\mathcal{C}}^2\right)^{-1/2}.$$

We set $\bar{\nu} = \min\{\nu_0, \nu_1\}$. For this value, we have that

$$\frac{T\epsilon_G^2}{\alpha} \cdot \frac{hL_{\text{eff}}^2}{\bar{\nu}} + \frac{T\bar{\nu}}{\tau} \cdot \left(\alpha h R_G^2 R_{Y,\mathcal{C}}^2\right) \leq \frac{T\epsilon_G^2}{\alpha} \cdot \frac{hL_{\text{eff}}^2}{\nu_0} + \frac{T\epsilon_G^2}{\alpha} \cdot \frac{hL_{\text{eff}}^2}{\nu_1} + \frac{T\nu_1}{\tau} \cdot \left(\alpha h R_G^2 R_{Y,\mathcal{C}}^2\right)$$

$$\leq \frac{T\epsilon_G^2}{\alpha} \cdot \frac{hL_{\text{eff}}^2}{\nu_0} + 2\sqrt{\frac{T^2\epsilon_G^2 h^2 L_{\text{eff}}^2 R_G^2 R_{Y,\mathcal{C}}^2}{\tau}}.$$

$$\leq \frac{T\epsilon_G^2}{\alpha} \cdot \frac{hL_{\text{eff}}^2}{\nu_0} + 2(T\epsilon_G)^{2/3}\sqrt{h}L_{\text{eff}} R_G R_{Y,\mathcal{C}}$$

$$\lesssim \frac{T\epsilon_G^2}{\alpha\sqrt{\kappa}} \cdot (1 + R_Y)hL_{\text{eff}}^2 + (T\epsilon_G)^{2/3}hL_{\text{eff}} R_G R_{Y,\mathcal{C}}.$$

Combining with the above,

$$\overline{\text{MemoryReg}}_T\left(\frac{\bar{\nu}}{4\alpha}; z_\star\right) \lesssim \frac{T\epsilon_G^2}{\alpha\sqrt{\kappa}} \cdot \underbrace{\left((1 + R_Y)hL_{\text{eff}}^2 + \beta^2\sqrt{\kappa}(c_v^2 + R_{Y,\mathcal{C}} + R_{Y,\mathcal{C}}^2) + \alpha^2\sqrt{\kappa}D^2\right)}_{:=C'_{\text{mid}}}.$$

$$+ (T\epsilon_G)^{2/3}\mathfrak{L} \cdot \underbrace{\left(hL_{\text{eff}} R_G R_{Y,\mathcal{C}} + \alpha D^2 + \alpha(1 + R_Y)R_{Y,\mathcal{C}} R_G^2 \cdot dL_{\text{eff}}\right)}_{:=C'_{\text{hi}}}$$

$$+ \frac{\mathfrak{L}}{\alpha\sqrt{\kappa}} \underbrace{(1 + R_Y)^2 R_G h^2 \cdot dL_{\text{eff}}^2}_{:=C_{\text{low}}} + \alpha\lambda D^2$$

where we use $C'_{\text{hi}}, C'_{\text{mid}}$ as intermediate constants that we simplify as follows. Recalling the

$$C'_{\text{mid}} = (1 + R_Y)hL_{\text{eff}}^2 + \beta^2\sqrt{\kappa}(c_v^2 + R_{Y,\mathcal{C}} + R_{Y,\mathcal{C}}^2) + \alpha^2\sqrt{\kappa}D^2$$

$$\leq (1 + \tfrac{\beta^2}{L^2})(1 + R_Y)hL_{\text{eff}}^2 + \beta^2\sqrt{\kappa}c_v^2 + \alpha^2\sqrt{\kappa}D^2 := C_{\text{mid}}$$

$$C'_{\text{hi}} = hL_{\text{eff}} R_G R_{Y,\mathcal{C}} + \alpha D^2 + \alpha(1 + R_Y)R_{Y,\mathcal{C}} R_G^2 \cdot dL_{\text{eff}}$$

$$\leq (1 + R_Y)L_{\text{eff}} R_G^2 R_{Y,\mathcal{C}}(h + d) + \alpha D^2 := C_{\text{hi}}$$

Note that the constant $C_{\text{hi}}, C_{\text{low}}, C_{\text{mid}}$ coincided with those in Definition G.1. Thus, writing our regret bound compactly, we have

$$\overline{\text{MemoryReg}}_T\left(\frac{\bar{\nu}}{4\alpha}; z_\star\right) \lesssim C_{\text{hi}}(T\epsilon_G)^{2/3}\mathfrak{L} + \frac{C_{\text{mid}}(T\epsilon_G^2)}{\alpha\sqrt{\kappa}} + \frac{C_{\text{low}}\mathfrak{L}}{\alpha\sqrt{\kappa}} + \alpha\lambda D^2.$$

Finally, let us expose $\bar{\nu}$. Recall we set $\bar{\nu} = \min\{\nu_0, \nu_1\}$, with $\nu_0 = \frac{\sqrt{\kappa}}{4(1+R_Y)}$, and

$$\nu_1 = \left(\frac{T\epsilon_G^2}{\alpha} \cdot hL_{\text{eff}}^2\right)^{1/2} \cdot \left(\frac{T}{\tau} \cdot \alpha h R_G^2 R_{Y,\mathcal{C}}^2\right)^{-1/2}.$$

$$= \frac{L_{\text{eff}}\epsilon_G\sqrt{\tau}}{\alpha R_G R_{Y,\mathcal{C}}} = \frac{L_{\text{eff}}(T\epsilon_G^4)^{1/3}}{\alpha R_G R_{Y,\mathcal{C}}},$$

finally yielding

$$\bar{\nu} = \min\left\{\frac{L_{\text{eff}}(T\epsilon_G^4)^{1/3}}{\alpha R_G R_{Y,\mathcal{C}}}, \frac{\sqrt{\kappa}}{4(1 + R_Y)}\right\},$$

To conclude, we paramaterize $\bar{\nu}' = \frac{\bar{\nu}}{4\eta}$. Since $\eta = \frac{3}{\alpha}$, we take

$$\bar{\nu}' = \frac{\alpha\sqrt{\kappa}}{48(1 + R_Y)} \min\left\{\frac{4(1 + R_Y)L_{\text{eff}}(T\epsilon_G^4)^{1/3}}{\alpha\sqrt{\kappa}R_G R_{Y,\mathcal{C}}}, 1\right\}$$

$$\geq \frac{\alpha\sqrt{\kappa}}{48(1 + R_Y)} \min\left\{\frac{4(1 + R_Y)L_{\text{eff}}(T\epsilon_G^4)^{1/3}}{R_G R_{Y,\mathcal{C}}}, 1\right\}$$

$$\geq \frac{\alpha\sqrt{\kappa}}{48(1 + R_Y)} \min\left\{4(1 + R_Y)(T\epsilon_G^4)^{1/3}, 1\right\} := \nu_\star$$

where in the last line we use $L \geq 1$ to bound $L_{\text{eff}} \geq R_G R_{Y,\mathcal{C}}$. Thus, taking $\nu_\star$ to be the above lower bound on $\bar{\nu}'$ concludes. $\square$

## H Ommited Proofs from Appendix F

### H.1 Proof of Proposition F.8

*Proof of Proposition F.8.* Let $v \in \mathbb{R}^{d_u}$, with $\|v\| = 1$, and let $u_s = \mathbf{Y}_s v$ for $s \in \{1 - h, 2 - h, \dots, t\}$, and set $u_s = 0$ for $s \le t - h$ and $s > t$. From Fact I.2, which shows that $\|v + w\|_2^2 \ge \frac{1}{2}\|v\|^2 - \|w\|^2$, we have

$$v^\top \sum_{s=1}^{t} \mathbf{H}_s^\top \mathbf{H}_s v := \sum_{s=1}^{t} \|\mathbf{H}_s v\|_2^2 = \sum_{s=1}^{t} \left\| \sum_{i=0}^{h} G^{[i]} \mathbf{Y}_{s-i} v \right\|_2^2$$

$$= \sum_{s=1}^{t} \left\| \sum_{i=0}^{h} G^{[i]} u_{s-i} \right\|_2^2$$

$$\ge \sum_{s=1-h}^{t+h} \left\| \sum_{i=0}^{h} G^{[i]} u_{s-i} \right\|_2^2 - 2h R_G^2 R_Y^2.$$

$$\overset{(\text{Fact I.2})}{\ge} \frac{1}{2} \sum_{s=1-h}^{t+h} \left\| \sum_{i=0}^{\infty} G^{[i]} u_{s-i} \right\|_2^2 - \sum_{s=1-h}^{t+h} \left\| \sum_{i>h}^{\infty} G^{[i]} u_{s-i} \right\|_2^2 - 2h R_G^2 R_Y^2$$

$$\ge \frac{1}{2} \sum_{s=1-h}^{t+h} \left\| \sum_{i=0}^{\infty} G^{[i]} u_{s-i} \right\|_2^2 - \sum_{s=1-h}^{t+h} \psi_G(h+1)^2 R_Y^2 - 2h R_G^2 R_Y^2$$

$$= \frac{1}{2} \sum_{s=1-h}^{t+h} \left\| \sum_{i=0}^{\infty} G^{[i]} u_{s-i} \right\|_2^2 - \underbrace{\left( t\psi_G(h+1)^2 + 4h R_G^2 \right) R_Y^2}_{:= \gamma_{t;h}},$$

where we use $\psi_G(h+1) \le \psi_G(0) = R_G^2$ in the last line. Moreover, setting $\tilde{u}_s = u_{s-h}$,

$$\sum_{s=1-h}^{t+h} \left\| \sum_{i=0}^{\infty} G^{[i]} u_{s-i} \right\|_2^2 = \sum_{s=1}^{t+2h} \left\| \sum_{i=0}^{\infty} G^{[i]} \tilde{u}_{s-i} \right\|_2^2$$

$$\overset{(i)}{=} \sum_{s=1}^{\infty} \left\| \sum_{i=0}^{s} G^{[i]} \tilde{u} \right\|_2^2$$

$$\overset{(ii)}{\ge} \kappa_0 \sum_{s=1}^{\infty} \|\tilde{u}_s\|_2^2$$

$$= \kappa_0 \sum_{s=1}^{\infty} \|u_{s-h}\|_2^2$$

where $(i)$ uses that we have $\tilde{u}_s = 0$ for $s \le 0$ and for $s \ge t + 2h$, and $(ii)$ invokes Definition 2.2. Combining the two displays, we have

$$v^\top \sum_{s=1}^{t} \mathbf{H}_s^\top \mathbf{H}_s v \ge \frac{\kappa_0}{2} \sum_{s=1}^{\infty} \|u_{s-h}\|_2^2 - \gamma_{t;h}$$

$$\ge \frac{\kappa_0}{2} \sum_{s=1}^{t+h} \|\mathbf{Y}_{s-h} v\|_2^2 - \gamma_{t;h}$$

$$= v^\top \left( \frac{\kappa_0}{2} \sum_{s=1-h}^{t} \mathbf{Y}_s^\top \mathbf{Y}_s - \gamma_{t;h} I \right) v,$$

where the last line uses $\|v\| = 1$. Finally, defining $c_{\psi;t} := \max\{1, \frac{t\psi_G(h+1)^2}{h R_G^2}\}$, we have $\gamma_{t;h} = R_Y^2 \left( t\psi_G(h+1)^2 + 4h R_G^2 \right) \le R_Y^2 \left( h c_{\psi;t} R_G^2 + 4h R_G^2 \right) \le 5h R_H^2 c_{\psi;t}$, yielding the desired bound. $\qquad \square$

## H.2  Proof of Lemma F.4

Let $z_\star \in \arg\min_{z \in \mathcal{C}} \sum_{t=1}^{T} f_t(z)$. Following the standard analysis of Online Newton Step (e.g. Hazan [18, Chapter 4] with $\gamma \leftarrow 1/\eta$), one has

$$\sum_{t=1}^{T} \nabla_t(\mathbf{z}_t - z_\star) \leq \frac{\eta}{2} \sum_{t=1}^{T} \nabla_t^\top \Lambda_t^{-1} \nabla_t + \frac{1}{2\eta} \sum_{t=1}^{T} (\mathbf{z}_t - z_\star)^\top (\Lambda_t - \Lambda_{t-1})(\mathbf{z}_t - z_\star) + \frac{1}{2\eta}(\mathbf{z}_1 - z_\star)^\top \Lambda_0 (\mathbf{z}_1 - z_\star)$$

The last term is at most $\frac{\lambda}{2\eta} D^2$. Moreover, since $\Lambda_t - \Lambda_{t-1} = \mathbf{H}_t \mathbf{H}_t^\top$,

$$\sum_{t=1}^{T} \nabla_t(\mathbf{z}_t - z_\star) - \frac{1}{2\eta} \|\mathbf{H}_t(\mathbf{z}_t - z_\star)\|_2^2 \leq \lambda D^2 + \frac{\eta}{2} \sum_{t=1}^{T} \nabla_t^\top \Lambda_t^{-1} \nabla_t.$$

Finally, for $\eta \geq \frac{1}{\alpha}$, we recognize that $\nabla_t(\mathbf{z}_t - z_\star) - \frac{1}{2\eta}\|\mathbf{H}_t(\mathbf{z}_t - z_\star)\|_2^2 \geq \nabla_t(\mathbf{z}_t - z_\star) - \frac{\alpha}{2}\|\mathbf{H}_t(\mathbf{z}_t - z_\star)\|_2^2 \geq f_t(\mathbf{z}_t) - f_t(z_\star)$ by Lemma F.3. Thus,

$$\sum_{t=1}^{T} f_t(\mathbf{z}_t) - f_t(z_\star) \leq \lambda D^2 + \frac{\eta}{2} \sum_{t=1}^{T} \nabla_t^\top \Lambda_t^{-1} \nabla_t,$$

as needed.  □

## H.3  Proof of Lemma F.6

We have $F_t(\mathbf{z}_t, \ldots, \mathbf{z}_{t-h}) - f_t(\mathbf{z}_t) = F_t(\mathbf{z}_t, \ldots, \mathbf{z}_{t-h}) - F_t(\mathbf{z}_t, \ldots \mathbf{z}_t)$. Therefore Taylor's theorem, there exists some $\mu \in [0, 1]$ such that, for $\bar{\mathbf{z}}_{t-i} = \mu \mathbf{z}_{t-i} + (1-\mu)\bar{\mathbf{z}}_t$,

$$F_t(\mathbf{z}_t, \ldots, \mathbf{z}_{t-h}) - f_t(\mathbf{z}_t) = (\nabla F_t(\bar{\mathbf{z}}_t, \ldots, \bar{\mathbf{z}}_{t-h}))^\top (0, \mathbf{z}_{t-1} - \mathbf{z}_t, \mathbf{z}_{t-2} - \mathbf{z}_t, \ldots, \mathbf{z}_{t-h} - \mathbf{z}_t).$$

By the Chain Rule, we then have

$$|F_t(\mathbf{z}_t, \ldots, \mathbf{z}_{t-h}) - f_t(\mathbf{z}_t)| = \left| \nabla \ell(\mathbf{v}_t + \sum_{i=0}^{h} G^{[i]} \mathbf{Y}_{t-i} \bar{\mathbf{z}}_t)^\top \left( \sum_{i=1}^{h} G^{[i]} \mathbf{Y}_{t-i}(\mathbf{z}_{t-i} - \mathbf{z}_t) \right) \right|$$

$$\leq \|\nabla \ell(\mathbf{v}_t + \sum_{i=0}^{h} G^{[i]} \mathbf{Y}_{t-i} \bar{\mathbf{z}}_t)\|_2 \cdot R_G \cdot \max_{i \in \{1, \ldots, h\}} \|\mathbf{Y}_{t-i}(\mathbf{z}_{t-i} - \mathbf{z}_t)\|_2.$$

Analogous to the Lemma F.2, we have $\|\nabla \ell(\mathbf{v}_t + \sum_{i=0}^{h} G^{[i]} \mathbf{Y}_{t-i} \bar{\mathbf{z}}_t)\|_2 \leq L_{\text{eff}}$, concluding the first part of the proof. For the second display, we have

$$\sum_{t=1}^{T} F_t(\mathbf{z}_t, \ldots, \mathbf{z}_{t-h}) - f_t(\mathbf{z}_t) \leq L_{\text{eff}} R_G \sum_{t=1}^{T} \max_{i \in \{1, \ldots, h\}} \|\mathbf{Y}_{t-i}(\mathbf{z}_t - \mathbf{z}_{t-i})\|_2$$

$$\leq L_{\text{eff}} R_G \sum_{t=1}^{T} \sum_{i=1}^{h} \|\mathbf{Y}_{t-i}(\mathbf{z}_t - \mathbf{z}_{t-i})\|_2$$

$$\leq L_{\text{eff}} R_G \sum_{t=1}^{T} \sum_{i=1}^{h} \sum_{j=1}^{i-1} \|\mathbf{Y}_{t-i}(\mathbf{z}_{t-j+1} - \mathbf{z}_{t-j})\|_2$$

$$= L_{\text{eff}} R_G \sum_{s=1-h}^{T} \sum_{i=1}^{h} \sum_{j=1}^{i-1} \|\mathbf{Y}_s(\mathbf{z}_{s+i-j+1} - \mathbf{z}_{s+i-j})\|_2$$

$$\leq h L_{\text{eff}} R_G \sum_{s=1-h}^{T} \sum_{i=1}^{h-1} \|\mathbf{Y}_s(\mathbf{z}_{s+i+1} - \mathbf{z}_{s+i})\|_2 \cdot \mathbb{I}_{s+i+1 \leq t}.$$

Finally, since $\mathbf{z}_t - \mathbf{z}_{t-1} = 0$ for $t \leq 1$, the above indicator $\mathbb{I}_{s+i+1 \leq t}$ can be replaced with $\mathbb{I}_{2 \leq s+i+1 \leq t} = \mathbb{I}_{1 \leq s+i \leq t-1}$, completing the proof.  □

## H.4   Proof of Lemma F.7

For $t \leq 0$, $\|\mathbf{Y}_s(\mathbf{z}_{t+1} - \mathbf{z}_t)\|_2 = 0$. Otherwise, we have

$$
\begin{aligned}
\|\mathbf{Y}_s(z_t - z_{t-1})\|_2 &= \|\mathbf{Y}_s \Lambda_t^{-1/2} \Lambda_t^{1/2}(\mathbf{z}_{t+1} - \mathbf{z}_t)\|_2 \\
&\leq \|\mathbf{Y}_s \Lambda_t^{-1/2}\|_{\text{op}} \cdot \|\Lambda_t^{1/2}(\mathbf{z}_{t+1} - \mathbf{z}_t)\|_2 \\
&\overset{(i)}{\leq} \|\mathbf{Y}_s \Lambda_t^{-1/2}\|_{\text{op}} \cdot \|\Lambda_t^{1/2}(\widetilde{\mathbf{z}}_{t+1} - \mathbf{z}_t)\|_2 \\
&= \|\mathbf{Y}_s \Lambda_t^{-1/2}\|_{\text{op}} \|\Lambda_t^{1/2} \cdot \eta \Lambda_t^{-1} \nabla_t\|_2 \\
&= \eta \sqrt{\|\mathbf{Y}_s \Lambda_t^{-1/2}\|_{\text{op}}^2 \|\Lambda_t^{-1/2} \nabla_t\|_2^2}, \quad\quad\quad\quad\quad (\text{H.1})
\end{aligned}
$$

where $(i)$ follows from the Pythagorean theorem, using that $\mathbf{z}_{t+1}$ is projected in the $\Lambda_t$-norm. Finally, we can crudely bound $\|\mathbf{Y}_s \Lambda_t^{-1} \mathbf{Y}_s\|_{\text{op}} \leq \text{tr}(\mathbf{Y}_s \Lambda_t^{-1} \mathbf{Y}_s)$. Since we consider indices $t \geq s$, we have $\text{tr}(\mathbf{Y}_s \Lambda_t^{-1} \mathbf{Y}_s) \leq \text{tr}(\mathbf{Y}_s \Lambda_s^{-1} \mathbf{Y}_s)$, where we have the understanding that $\Lambda_s = \Lambda_1$ for $s \leq 0$. Thus, we see that for $t > 0$,

$$
\|\mathbf{Y}_s(\mathbf{z}_{t+1} - \mathbf{z}_t)\|_2 \leq \eta \text{tr}(\mathbf{Y}_s \Lambda_s^{-1} \mathbf{Y}_s)^{1/2} \text{tr}(\nabla_t^\top \Lambda_t^{-1} \nabla_t))^{1/2}
$$

Thus, from Lemma F.6 and by Cauchy Schwartz,

$$
\begin{aligned}
\text{MoveDiff}_T &\leq h L_{\text{eff}} R_G \sum_{i=1}^{h-1} \sum_{s=1-h}^{T} \|\mathbf{Y}_s(\mathbf{z}_{s+i+1} - \mathbf{z}_{s+i})\|_2 \mathbb{I}_{1 \leq s+i \leq t-1} \\
&\leq \eta h L_{\text{eff}} R_G \cdot \sum_{i=1}^{h-1} \sqrt{\sum_{s=1-h}^{T} \mathbb{I}_{1 \leq s+i \leq t-1} \cdot \text{tr}(\mathbf{Y}_s \Lambda_s^{-1} \mathbf{Y}_s)} \sqrt{\sum_{s=1-h}^{T} \mathbb{I}_{1 \leq s+i \leq t-1} \cdot \text{tr}(\nabla_{s+i}^\top \Lambda_{s+i}^{-1} \nabla_{s+i})} \\
&\leq \eta h^2 L_{\text{eff}} R_G \cdot \sqrt{\sum_{s=1-h}^{T} \text{tr}(\mathbf{Y}_s \Lambda_h^{-1} \mathbf{Y}_s)} \sqrt{\sum_{s=1}^{T} \text{tr}(\nabla_t^\top \Lambda_t^{-1} \nabla_t)},
\end{aligned}
$$

as needed.                                                                                              $\square$

# I   Ommited Proofs from Appendix G

## I.1   Useful Facts for Analysis

We begin by listing some useful elementary facts:

**Fact I.1.** *For all $t \geq 1$ and all $z \in \mathcal{C}$, we have $\|\mathbf{H}_t - \widehat{\mathbf{H}}_t\|_{\text{op}} \leq \epsilon_G R_Y$ and $\|(\mathbf{H}_t - \widehat{\mathbf{H}}_t)z\|_{\text{op}} \leq \epsilon_G R_{Y,\mathcal{C}}$*

*Proof.* $\|\mathbf{H}_t - \widehat{\mathbf{H}}_t\|_{\text{op}} = \|\sum_{i=0}^{h}(G_\star^{[i]} - \widehat{G}^{[i]})\mathbf{Y}\|_{\text{op}} \leq \|R_Y\|_{\text{op}} \sum_{i=0}^{h} \|G_\star^{[i]} - \widehat{G}^{[i]}\|_{\text{op}} \leq \epsilon_G R_Y$. The second bound is similar.                                                                        $\square$

**Fact I.2.** *Given two vectors $v, w \in \mathbb{R}^m$, $\|v + w\|_2^2 \geq \frac{1}{2}\|v\|^2 - \|w\|^2$.*

*Proof.* $\|v + w\|_2^2 = \|v\|^2 + \|w\|^2 + 2\langle v, w \rangle \geq \|v\|^2 + \|w\|^2 - 2\|v\|\|w\| \geq \|v\|^2 + \|w\|^2 - \frac{1}{2}\|v\|^2 - 2\|w\|^2 = \frac{\|v\|^2}{2} - \|w\|^2$, as needed.                                                                        $\square$

**Fact I.3.** $\|a\|^2 \leq \|b\|^2 + (\|a\| + \|b\|)\|b - a\|$

*Proof.* $\|a\|_2^2 = \langle a, a \rangle = \langle b - a, a \rangle + \langle b, a \rangle = \langle b - a, a + \langle b, a - b \rangle + \|b\|^2$. The bound now follows form Cauchy-Schwartz                                                                        $\square$

## I.2 Proof of Lemma G.1

Let $z_\star \in \mathcal{C}$ be an arbitrary comparator point. Analogus to the proof of Lemma F.4,

$$\sum_{t=1}^{T} \widehat{f}_t(\mathbf{z}_t) - \widehat{f}_t(z_\star) \leq \sum_{t=1}^{T} \widehat{\nabla}_t^\top (\mathbf{z}_t - z_\star) - \frac{\alpha}{2} \|\widehat{\mathbf{H}}_t(\mathbf{z}_t - z_\star)\|_2^2 \tag{I.1}$$

One the other hand, the standard inequality obtained from applying Semi-ONS to the $(\widehat{f}_t)$-sequence (see, for analogy, page 58 of [18]), we obtain

$$\widehat{\nabla}_t^\top (\mathbf{z}_t - z_\star) \leq \frac{\eta}{2} \|\widehat{\nabla}\|_{\widehat{\Lambda}_t^{-1}}^2 + \frac{2}{\eta} \|\mathbf{z}_t - z_\star\|_{\widehat{\Lambda}_t}^2 - \frac{2}{\eta} \|\mathbf{z}_{t+1} - z_\star\|_{\widehat{\Lambda}_t}^2.$$

Summing up over $t$ and telescoping

$$\sum_{t=1}^{T} \widehat{\nabla}_t^\top (\mathbf{z}_t - z_\star) \leq \frac{\eta}{2} \sum_{t=h+1}^{T} \|\widehat{\nabla}\|_{\widehat{\Lambda}_t^{-1}}^2 + \sum_{t=1}^{T} \frac{1}{2\eta} \|\mathbf{z}_t - z_\star\|_{\widehat{\Lambda}_t - \widehat{\Lambda}_{t-1}}^2 + \frac{1}{2\eta} \|\mathbf{z}_h - z_\star\|_{\widehat{\Lambda}_h}^2$$

$$= \frac{\eta}{2} \sum_{t=1}^{T} \|\widehat{\nabla}\|_{\widehat{\Lambda}_t^{-1}}^2 + \frac{1}{2\eta} \sum_{t=1}^{T} \|\widehat{\mathbf{H}}_t(\mathbf{z}_t - z_\star)\|^2 + \frac{\lambda D^2}{2\eta}, \tag{I.2}$$

where we use $\widehat{\Lambda}_t - \widehat{\Lambda}_{t-1} = \widehat{\mathbf{H}}_t^\top \widehat{\mathbf{H}}_t$ and $\widehat{\Lambda}_0 = \lambda I$. Thus, introducing $\mathbf{err}_t := \nabla \widehat{f}_t(z) - \nabla f(\mathbf{z}_t)$ and combining (I.1) and (I.2),

$$\sum_{t=1}^{T} f_t(\mathbf{z}_t) - f_t(z_\star) \leq \sum_{t=1}^{T} \mathbf{err}_t^\top (\mathbf{z}_t - z_\star) + \frac{1}{2\eta} \sum_{t=1}^{T} (\|\widehat{\mathbf{H}}_t(\mathbf{z}_t - z_\star)\|^2 - \eta\alpha \|\mathbf{H}_t(\mathbf{z}_t - z_\star)\|^2)$$

$$+ \frac{\eta}{2} \sum_{t=1}^{T} \|\widehat{\nabla}\|_{\widehat{\Lambda}_t^{-1}}^2 + \frac{\lambda D^2}{2\eta}$$

Plugging in $\boldsymbol{\delta}_t = \mathbf{z}_t - z_\star$ concludes the proof, and re-iterating the proof of Proposition F.1 concludes the proof.

## I.3 Proof of Lemma G.2

First, we can bound $\|\widehat{\mathbf{H}}_t \boldsymbol{\delta}_t\|^2 \leq 2\|\mathbf{H}_t \boldsymbol{\delta}_t\|^2 + 2\|(\mathbf{H}_t - \widehat{\mathbf{H}}_t)\boldsymbol{\delta}_t\|^2$, and

$$\|(\mathbf{H}_t - \widehat{\mathbf{H}}_t)\boldsymbol{\delta}_t\| \leq \|(\mathbf{H}_t - \widehat{\mathbf{H}}_t)\mathbf{z}_t\| + \|(\mathbf{H}_t - \widehat{\mathbf{H}}_t)z_\star\| \leq 2R_{Y,\mathcal{C}}\epsilon_G$$

by Fact I.1. Taking $\eta \geq \frac{3}{\alpha}$, we find then that

$$\|\widehat{\mathbf{H}}_t \boldsymbol{\delta}_t\|^2 - \eta\alpha \|\mathbf{H}_t \boldsymbol{\delta}_t\|^2 \leq 2\|\mathbf{H}_t \boldsymbol{\delta}_t\|^2 + 8R_{Y,\mathcal{C}}^2 \epsilon_G^2 - 3\|\mathbf{H}_t \boldsymbol{\delta}_t\|^2 = -\|\mathbf{H}_t \boldsymbol{\delta}_t\|^2 + 8R_{Y,\mathcal{C}}^2 \epsilon_G^2.$$

The second statement of the lemma follows by substitution into Lemma G.1. $\qquad\square$

## I.4 Proof of Lemma G.3

We have the bound

$$\mathbf{err}_t := \nabla \widehat{f}_t(z) - \nabla f(\mathbf{z}_t)$$

$$= \widehat{\mathbf{H}}_t^\top \nabla \ell_t(\widehat{\mathbf{v}}_t + \widehat{\mathbf{H}}_t \mathbf{z}_t) - \mathbf{H}_t \nabla \ell_t(\mathbf{v}_t^\star + \mathbf{H}_t \mathbf{z}_t)$$

$$= (\widehat{\mathbf{H}}_t - \mathbf{H}_t)^\top \nabla \ell_t(\widehat{\mathbf{v}}_t + \widehat{\mathbf{H}}_t \mathbf{z}_t) + \mathbf{H}_t \left( \nabla \ell_t(\widehat{\mathbf{v}}_t + \widehat{\mathbf{H}}_t \mathbf{z}_t) - \nabla \ell_t(\mathbf{v}_t^\star + \mathbf{H}_t \mathbf{z}_t) \right).$$

Defining

$$g_{t,1} := \nabla \ell_t(\widehat{\mathbf{v}}_t + \widehat{\mathbf{H}}_t \mathbf{z}_t)$$

$$g_{t,2} := \left( \nabla \ell_t(\widehat{\mathbf{v}}_t + \widehat{\mathbf{H}}_t \mathbf{z}_t) - \nabla \ell_t(\mathbf{v}_t^\star + \mathbf{H}_t \mathbf{z}_t) \right)$$

We have that $\|g_{t,1}\|_2 \leq L_{\mathrm{eff}}$ by analogy to Lemma F.2. Moreover, since $\beta$-smoothness implies that the gradients are $\beta$-Lipschitz, and by invoking Fact I.1, we have

$$\left( \nabla \ell_t(\widehat{\mathbf{v}}_t + \widehat{\mathbf{H}}_t \mathbf{z}_t) - \nabla \ell_t(\mathbf{v}_t^\star + \mathbf{H}_t \mathbf{z}_t) \right) \leq \beta \|(\widehat{\mathbf{v}}_t + \widehat{\mathbf{H}}_t \mathbf{z}_t) - (\mathbf{v}_t^\star + \mathbf{H}_t \mathbf{z}_t)\| \leq \beta(c_v \epsilon_G + 2\epsilon_G R_{Y,\mathcal{C}}).$$

$\qquad\square$

## I.5 Proof of Lemma G.4

Recall that from Lemma G.2, we have the bound

$$\sum_{t=1}^{T} f_t(\mathbf{z}_t) - f_t(z_\star) \le \sum_{t=1}^{T} \mathbf{err}_t^\top \boldsymbol{\delta}_t - \frac{1}{2\eta} \sum_{t=1}^{T} \|\mathbf{H}_t \boldsymbol{\delta}_t\|^2 + \frac{4}{\eta} T R_{Y,\mathcal{C}}^2 \epsilon_G^2 + \widehat{\mathrm{Reg}}_T. \qquad (\mathrm{I.3})$$

Let us now bound the sum $\sum_{t=1}^{T} \mathbf{err}_t^\top \boldsymbol{\delta}_t$ via Lemma G.3. The lemma ensures $\mathbf{err}_t = (\widehat{\mathbf{H}}_t - \mathbf{H}_t)^\top g_{1,t} + \mathbf{H}_t^\top g_{2,t}$. where $\|g_{1,t}\|_2 \le L_{\mathrm{eff}}$ and $\|g_{2,t}\| \le \beta \epsilon_G(c_v + 2R_{Y,\mathcal{C}})$. The contribution of the term including $g_{2,t}$ is easily adressed:

$$(\mathbf{H}_t^\top g_{2,t})^\top \boldsymbol{\delta}_t \le \|g_{2,t}\|_2 \|\mathbf{H}_t \boldsymbol{\delta}_t\|_2 \le \beta \epsilon_G (c_v + 2R_{Y,\mathcal{C}}) \|\mathbf{H}_t \boldsymbol{\delta}_t\|_2 \le \eta \beta^2 \epsilon_G^2 (c_v + 2R_{Y,\mathcal{C}})^2 + \frac{1}{4\eta} \|\mathbf{H}_t \boldsymbol{\delta}_t\|_2,$$

by the AM-GM inequality. Next, we handle the term $(\widehat{\mathbf{H}}_t - \mathbf{H}_t)^\top g_{1,t}$. First we bound

$$((\widehat{\mathbf{H}}_t - \mathbf{H}_t)^\top g_{1,t})^\top \boldsymbol{\delta}_t \le \|g_{1,t}\| \|(\widehat{\mathbf{H}}_t - \mathbf{H}_t) \boldsymbol{\delta}_t\| \le L_{\mathrm{eff}} \|(\widehat{\mathbf{H}}_t - \mathbf{H}_t) \boldsymbol{\delta}_t\|.$$

Plugging into Eq. (I.3) gives

$$\sum_{t=1}^{T} f_t(\mathbf{z}_t) - f_t(z_\star) \le \sum_{t=1}^{T} L_{\mathrm{eff}} \|(\widehat{\mathbf{H}}_t - \mathbf{H}_t) \boldsymbol{\delta}_t\| - \frac{1}{4\eta} \sum_{t=1}^{T} \|\mathbf{H}_t \boldsymbol{\delta}_t\|^2$$
$$+ T \left( \eta \beta^2 (c_v + 2\epsilon_G R_{Y,\mathcal{C}})^2 + \frac{4R_{Y,\mathcal{C}}^2}{\eta} \right) \epsilon_G^2 + \widehat{\mathrm{Reg}}_T. \qquad (\mathrm{I.4})$$

For arbitrary sequences $\mathbf{H}_t, \widehat{\mathbf{H}}_t$, there is no obvious way to cancel the terms $L_{\mathrm{eff}} \|(\widehat{\mathbf{H}}_t - \mathbf{H}_t) \boldsymbol{\delta}_t\|$ and $-\|\mathbf{H}_t \boldsymbol{\delta}_t\|^2$ to achieve a $\mathcal{O}(T\epsilon_G^2)$-error dependence. However, there is additional structure we can leverage. We can observe that

$$\|(\widehat{\mathbf{H}}_t - \mathbf{H}_t) \boldsymbol{\delta}_t\|_2^2 = \left\| \sum_{i=0}^{h} (\widehat{G}^{[i]} - G_\star)^{[i]} \mathbf{Y}_{t-i} \boldsymbol{\delta}_t \right\|_2^2 \le \epsilon_G \max_{i \in [0:h]} \|\mathbf{Y}_{t-i} \boldsymbol{\delta}_t\|^2.$$

Hence, by AMG-GM, we have that for any $\nu > 0$,

$$L_{\mathrm{eff}} \|(\widehat{\mathbf{H}}_t - \mathbf{H}_t) \boldsymbol{\delta}_t\| \le \nu^{-1}(h+1) \eta L_{\mathrm{eff}}^2 \epsilon_G^2 + \frac{\nu}{4(h+1)\eta} \max_{i \in [0:h]} \|\mathbf{Y}_{t-i} \boldsymbol{\delta}_t\|^2.$$

Together with Eq. (I.4), the above display implies

$$\sum_{t=1}^{T} f_t(\mathbf{z}_t) - f_t(z_\star) \le \frac{1}{4\eta} \sum_{t=1}^{T} \left( \frac{\nu}{h+1} \sum_{i=0}^{h} \|\mathbf{Y}_{t-i} \boldsymbol{\delta}_t\|^2 - \|\mathbf{H}_t \boldsymbol{\delta}_t\|^2 \right) + T\epsilon_G^2 \cdot \mathrm{ERR}(\nu) + \widehat{\mathrm{Reg}}_T,$$

where $\mathrm{ERR}(\nu) := \left( \frac{\eta(h+1)L_{\mathrm{eff}}^2}{\nu} + \eta\beta^2(c_v + 2R_{Y,\mathcal{C}})^2 + \frac{4R_{Y,\mathcal{C}}}{\eta} \right)$. $\qquad \square$

## I.6 Proof of Lemma G.5

Fix a block length $\tau \in \mathbb{N}$, and recall the index $k_j = (j-1)\tau$, and $j_{\max}$ as the largest $j$ such that $j_{\max}\tau \leq T$. We bound

$$\sum_{t=1}^{T} \|\mathbf{Y}_{t-i}\boldsymbol{\delta}_t\|_2^2$$

$$= \sum_{j=1}^{j_{\max}} \sum_{s=1}^{\tau} \|\mathbf{Y}_{k_j+s-i}\boldsymbol{\delta}_{k_j+s}\|_2^2 + \sum_{s=1+\tau(j_{\max}-1)}^{T} \|\mathbf{Y}_{i-h}\boldsymbol{\delta}_t\|_2^2$$

$$\leq 4\tau R_{Y,\mathcal{C}} + \sum_{j=1}^{j_{\max}} \sum_{s=1}^{\tau} \|\mathbf{Y}_{k_j+s-i}\boldsymbol{\delta}_{k_j+s}\|_2^2$$

$$\overset{(i)}{\leq} 4\tau R_{Y,\mathcal{C}} + \sum_{j=1}^{j_{\max}} \sum_{s=1}^{\tau} \|\mathbf{Y}_{k_j+s-i}\boldsymbol{\delta}_{k_j}\|_2^2 + (\|\mathbf{Y}_{k_j+s-i}\boldsymbol{\delta}_{k_j+1}\|_2 + \|\mathbf{Y}_{k_j+s-i}\boldsymbol{\delta}_{k_j+s}\|_2)\|\mathbf{Y}_{k_j+s-i}(\boldsymbol{\delta}_{k_j+s} - \boldsymbol{\delta}_{k_j+1})\|_2$$

$$\overset{(ii)}{\leq} 4\tau R_{Y,\mathcal{C}} + \sum_{j=1}^{j_{\max}} \sum_{s=1}^{\tau} \|\mathbf{Y}_{k_j+s-i}\boldsymbol{\delta}_{k_j+1}\|_2^2 + 4R_{Y,\mathcal{C}} \sum_{j=1}^{j_{\max}} \sum_{s=1}^{\tau} \|\mathbf{Y}_{k_j+s-i}(\boldsymbol{\delta}_{k_j+s} - \boldsymbol{\delta}_{k_j+1})\|_2, \quad \text{(I.5)}$$

Where $(i)$ uses the inequality $\|a\|^2 \leq \|b\|^2 + (\|a\| + \|b\|)\|b - a\|$ from Fact I.3, and where $(ii)$ uses the $\|\mathbf{Y}_s(\boldsymbol{\delta}_t)\| \leq \|\mathbf{Y}_s z_\star\| + \|\mathbf{Y}_s \mathbf{z}_t\| \leq 2R_{Y,\mathcal{C}}$.

Next, recalling $\boldsymbol{\delta}_t := \mathbf{z}_t - z_\star$, we develop

$$\sum_{j=1}^{j_{\max}} \sum_{s=1}^{\tau} \|\mathbf{Y}_{k_j+s-i}(\boldsymbol{\delta}_{k_j+s} - \boldsymbol{\delta}_{k_j+1})\|_2 = \sum_{j=1}^{j_{\max}} \sum_{s=2}^{\tau} \|\mathbf{Y}_{k_j+s-i}(\mathbf{z}_{k_j+s} - \mathbf{z}_{k_j+1})\|_2$$

$$\leq \sum_{j=1}^{j_{\max}} \sum_{s=2}^{\tau} \sum_{s'=0}^{s-2} \|\mathbf{Y}_{k_j+s-i}(\mathbf{z}_{k_j+s-s'} - \mathbf{z}_{k_j-s'-1})\|_2$$

$$\leq \sum_{j=1}^{j_{\max}} \sum_{s=2}^{\tau} \sum_{s'=0}^{\tau'-1} \|\mathbf{Y}_{k_j+s-i}(\mathbf{z}_{k_j+s-s'} - \mathbf{z}_{k_j-s'-1})\|_2$$

$$\leq \sum_{t=1}^{T} \sum_{s'=0}^{\tau-1} \|\mathbf{Y}_{t-i}(\mathbf{z}_{t-s'} - \mathbf{z}_{t-s'-1})\|_2,$$

where above we use the convention $\mathbf{z}_t = 0$ for $t \leq 1$, and that the induces $k_j + s$ range over a subset of $t \in [T]$. Relabeling $s'$ with $s$, and combining with Eq. (I.5) this finally yields

$$\sum_{t=1}^{T} \|\mathbf{Y}_{t-i}\boldsymbol{\delta}_t\|_2^2 \geq 4\tau R_{Y,\mathcal{C}} + \sum_{j=1}^{j_{\max}} \sum_{s=1}^{\tau} \|\mathbf{Y}_{k_j+s-i}\boldsymbol{\delta}_{k_j}\|_2^2 + 4R_{Y,\mathcal{C}} \sum_{t=1}^{T} \sum_{s=0}^{\tau-1} \|\mathbf{Y}_{t-i}(\mathbf{z}_{t-s} - \mathbf{z}_{t-s-1})\|_2.$$

Following similar steps (but using Fact I.1 to bound $\|\mathbf{H}_t z\| \leq R_G R_{Y,\mathcal{C}}$), we obtain

$$\sum_{t=1}^{T} \|\mathbf{H}_t \boldsymbol{\delta}_t\|_2^2 \geq \sum_{j=1}^{j_{\max}} \sum_{s=1}^{\tau} \|\mathbf{H}_{k_j+s}\boldsymbol{\delta}_{k_j}\|_2^2 - 4R_{Y,\mathcal{C}} R_G \sum_{t=1}^{T} \sum_{s=0}^{\tau-1} \|\mathbf{H}_t(\mathbf{z}_{t-s} - \mathbf{z}_{t-s-1})\|_2,$$

$\square$

## I.7 Proof of Lemma G.6

Recall our convention $\widehat{\Lambda}_s = \widehat{\Lambda}_1$ and $\Lambda_s = \Lambda_1$ for $s \le 1$. For any $\mu \in (0,1]$, we have the bound

$$
\begin{aligned}
\widehat{\Lambda}_{t-\tau} = \lambda I + \sum_{s=1}^{t-\tau} \widehat{\mathbf{H}}_s^\top \widehat{\mathbf{H}}_s \; &\succeq \; \lambda I + \mu \sum_{s=1}^{t-\tau} \widehat{\mathbf{H}}_s^\top \widehat{\mathbf{H}}_s \\
&\succeq \; (\lambda - \mu\tau R_H^2)I + \mu \sum_{s=1}^{t} \widehat{\mathbf{H}}_s^\top \widehat{\mathbf{H}}_s \\
&\succeq \; (\lambda - \mu\tau R_H^2)I + \sum_{s=1}^{t} \frac{\mu}{2}\mathbf{H}_s^\top \mathbf{H}_s - \mu(\widehat{\mathbf{H}}_s - \mathbf{H}_s)^\top(\widehat{\mathbf{H}}_s - \mathbf{H}_s),
\end{aligned}
$$

where the last step follows from Fact I.2. We can crudely bound $(\widehat{\mathbf{H}}_s - \mathbf{H}_s)^\top(\widehat{\mathbf{H}}_s - \mathbf{H}_s) \preceq \|\widehat{\mathbf{H}}_s - \mathbf{H}_s\|^2 I \preceq R_Y^2 \epsilon_G^2 I$ via Fact I.1, giving

$$
\widehat{\Lambda}_{t-\tau} \succeq (\lambda - \mu\tau R_H^2 - \mu R_Y^2 t\epsilon_G^2)I + \frac{\mu}{2}\sum_{s=1}^{t} \mathbf{H}_s^\top \mathbf{H}_s.
$$

Bounding $t \le T$, and taking $\mu = \min\{1, \frac{\lambda}{2(\tau R_H^2 + R_Y^2 \epsilon_G^2 T)}\}$, we obtain

$$
\widehat{\Lambda}_{t-\tau} \succeq \frac{\lambda}{2} + \frac{\mu}{2}\sum_{s=1}^{t} \mathbf{H}_s^\top \mathbf{H}_s \succeq \frac{\mu}{2}\Lambda_t
$$

Thus, for any upper bound $c_\Lambda \ge \sqrt{\frac{2}{\mu}}$

$$
\widehat{\Lambda}_{t-\tau}^{-1} \preceq \frac{2}{\mu}\Lambda_t^{-1} \preceq c_\Lambda^2 \Lambda_t^{-1}. \tag{I.6}
$$

Finally, we can bound

$$
\begin{aligned}
\sqrt{\frac{2}{\mu}} &= \sqrt{\max\{2, \frac{4(\tau R_H^2 + R_Y^2 \epsilon_G^2 T)}{\lambda}\}} \\
&= \sqrt{\max\{2, 4R_Y^2 \frac{\tau R_G^2 + \epsilon_G^2 T}{\lambda}\}} \\
&\overset{(i)}{\ge} \sqrt{\max\{2, 4c_\lambda^{-1} R_Y^2 (1 + \frac{\tau R_G^2}{\lambda})} \\
&\le 2(1 + R_Y) + 2c_\lambda^{-\frac{1}{2}} R_Y \sqrt{\frac{\tau R_G^2}{\lambda}} := c_\Lambda,
\end{aligned}
$$

where we use that $\lambda \ge c_\lambda T\epsilon_G^2$ in $(i)$. This verifies that $c_\Lambda$ in the lemma is an upper bound on $\sqrt{2/\mu}$, and the lemma now follows from Eq. (I.6). $\square$

## I.8 Proof of Lemma G.7

Let $\tau \in \mathbb{N}$ denote our blocking parameter. Again, adopt the convention $\widehat{\Lambda}_s = \widehat{\Lambda}_1$ and $\Lambda_s = \Lambda_1$ for $s \le 0$, and let $c_\Lambda$ be such from Lemma G.5, which ensures that, for all $t$,

$$
\widehat{\Lambda}_{t-\tau}^{-1} \preceq c_\Lambda^2 \Lambda_t^{-1}. \tag{I.7}
$$

Then, any for $s \in \{0, \ldots, \tau - 1\}$ such that $s \leq t - 1$ any $\mu > 0$, we have

$$
\begin{aligned}
\|\mathbf{Y}_{t-i}(\mathbf{z}_{t-s} - \mathbf{z}_{t-s-1})\|_2 &\leq \|\mathbf{Y}_{t-i}\widehat{\Lambda}_{t-s-1}^{-\frac{1}{2}}\|_{\mathrm{op}}\|\widehat{\Lambda}_{t-s-1}^{\frac{1}{2}}(\mathbf{z}_{t-s} - \mathbf{z}_{t-s-1})\|_2 \\
&\leq \|\mathbf{Y}_{t-i}\widehat{\Lambda}_{t-\tau-i}^{\frac{1}{2}}\|_{\mathrm{op}}\|\widehat{\Lambda}_{t-s-1}^{\frac{1}{2}}(\mathbf{z}_{t-s} - \mathbf{z}_{t-s-1})\|_2 \\
&\leq \|\mathbf{Y}_{t-i}\widehat{\Lambda}_{t-\tau-i}^{-\frac{1}{2}}\|_{\mathrm{op}}\|\widehat{\Lambda}_{t-s-1}^{\frac{1}{2}}\hat{\nabla}_{t-s-1}\|_2 \qquad \text{(Projection Step)} \\
&\leq \sqrt{\mathrm{tr}(\mathbf{Y}_{t-i}\widehat{\Lambda}_{t-\tau-i}^{-1}\mathbf{Y}_{t-i}) \cdot \|\hat{\nabla}_{t-s-1}\|_{\widehat{\Lambda}_{t-s-1}}^2} \\
&\leq c_\Lambda \sqrt{\mathrm{tr}(\mathbf{Y}_{t-i}\widehat{\Lambda}_{t-i}^{-1}\mathbf{Y}_{t-i}) \cdot \|\hat{\nabla}_{t-s-1}\|_{\widehat{\Lambda}_{t-s-1}}^2}. \qquad \text{(Eq. (I.7))}
\end{aligned}
$$

Note that the above expression does not depend on $\tau$. Thus, since $\mathbf{z}_{t-s} - \mathbf{z}_{t-s-1} = 0$ for $s > t - 1$ (recall here we assume $\mathbf{z}_i = \mathbf{z}_1$ for $i \leq 1$), an application of Cauchy Schwartz yields

$$
\begin{aligned}
\sum_{t=1}^{T}\sum_{s=0}^{\tau-1}\|\mathbf{Y}_{t-i}(\mathbf{z}_{t-s} - \mathbf{z}_{t-s-1})\|_2 &\leq \tau c_\Lambda \left(\sum_{t=s+1}^{T}\mathrm{tr}(\mathbf{Y}_{t-i}\Lambda_{t-i}^{-1}\mathbf{Y}_{t-i})\right)^{\frac{1}{2}}\left(\sum_{t=s+1}^{T}\|\hat{\nabla}_{t-s}\|_{\Lambda_{t-s}}^2\right)^{\frac{1}{2}} \\
&\leq \tau c_\Lambda \left(\sum_{t=1}^{T}\mathrm{tr}(\mathbf{Y}_{t-i}\Lambda_{t-i}^{-1}\mathbf{Y}_{t-i})\right)^{\frac{1}{2}}\left(\sum_{t=1}^{T}\|\hat{\nabla}_{t}\|_{\widehat{\Lambda}_t}^2\right)^{\frac{1}{2}} \\
&\leq \tau c_\Lambda \left(\sum_{t=1-h}^{T}\mathrm{tr}(\mathbf{Y}_t\Lambda_t^{-1}\mathbf{Y}_t)\right)^{\frac{1}{2}}\left(\sum_{t=1}^{T}\|\hat{\nabla}_{t}\|_{\widehat{\Lambda}_t}^2\right)^{\frac{1}{2}} \qquad \text{(I.8)}
\end{aligned}
$$

Arguing as in the proof of Proposition F.1, and using $\lambda \geq hR_G^2 \geq 1$,

$$
\sum_{t=1}^{T}\|\hat{\nabla}_t\|_{\widehat{\Lambda}_t}^2 \leq L_{\mathrm{eff}}^2 \sum_{t=1}^{T}\mathrm{tr}\left(\widehat{\mathbf{H}}_t\widehat{\Lambda}_t^{-1}\widehat{\mathbf{H}}_t\right)^{\frac{1}{2}} \leq dL_{\mathrm{eff}}^2\log(1 + \frac{TR_H^2}{\lambda}) \leq dL_{\mathrm{eff}}^2 \cdot \mathfrak{L}. \qquad \text{(I.9)}
$$

We now develop a simple claim, which is a consequence of Proposition F.8:

**Claim I.4.** *Recall* $c_{\psi;t} := \max\{1, \frac{t\psi_G(h+1)^2}{hR_G^2}\}$, *and set* $\mu_0 = \min\{1, \frac{\lambda}{10hR_H^2 c_{\psi;T}}\}$. *We have*

$$
\sum_{t=1-h}^{T}\mathrm{tr}(\mathbf{Y}_t^\top\Lambda_t^{-1}\mathbf{Y}_t) \leq \frac{2d}{\mu_0\kappa}\mathfrak{L}.
$$

*Proof of Claim I.4.* From Proposition F.8, we have the bound

$$
\sum_{s=1}^{t}\mathbf{H}_s^\top\mathbf{H}_s \succeq \frac{\kappa}{2}\sum_{s=1-h}^{t}\mathbf{Y}_s^\top\mathbf{Y}_s - 5hR_H^2 c_{\psi;t}I.
$$

Thus, for any $\mu_0 = \min\{1, (10hR_H^2 c_{\psi;T})^{-1}\} \leq 1$,

$$
\begin{aligned}
\Lambda_t = \lambda I + \sum_{s=1}^{t}\mathbf{H}_t\mathbf{H}_t^\top &\geq \lambda I + \mu_0\sum_{s=1}^{t}\mathbf{H}_t\mathbf{H}_t^\top \\
&= \lambda I + \mu_0\left(\frac{\kappa}{2}\sum_{s=1-h}^{t}\mathbf{Y}_s\mathbf{Y}_s^\top - 5hR_H^2 c_{\psi;T}\right) \succeq \frac{\lambda}{2}I + \frac{\mu_0\kappa}{2}\sum_{s=1-h}^{t}\mathbf{Y}_s\mathbf{Y}_s^\top.
\end{aligned}
$$

Hence, from the log-det potential bound of Lemma F.5, the bounds $\mu_0, \kappa \leq 1$ and $R_H = R_GR_Y$

$$
\sum_{s=1-h}^{T}\mathrm{tr}(\mathbf{Y}_s^\top\Lambda_s^{-1}\mathbf{Y}_s) \leq \frac{2d}{\mu_0\kappa}\log(1 + \frac{\mu_0\kappa TR_Y^2}{\lambda}) \leq \frac{2d}{\mu_0\kappa}\log(1 + \frac{TR_H^2}{\lambda}) = \frac{2d}{\mu_0\kappa}\mathfrak{L}.
$$

$\square$

To apply the above, let us simplify our expression for $\mu_0$. Recall that

$$\mu_0 = \min\left\{1, \frac{\lambda}{10hR_H^2 c_{\psi;T}}\right\}, \quad c_{\psi;T} := \max\left\{1, \frac{T\psi_G(h+1)^2}{hR_G^2}\right\} \leq (1 + T\epsilon_G^2/hR_G^2),$$

where we note that $\epsilon_G = \|\widehat{G} - G\|_{\ell_1,\mathrm{op}} \geq \sum_{i>h}\|G^{[i]}\|_{\mathrm{op}} \geq \psi_G(h+1)$, since $\widehat{G}^{[i]} = 0$ for $i > h$. Using the bounds $R_H/R_G = R_Y$ and $\lambda \geq c_\lambda(T\epsilon_G^2 + hR_G^2)$ for $c_\lambda \in (0,1]$,

$$\begin{aligned}
\mu_0^{-1} &\leq 1 + \frac{10hR_H^2 c_{\psi;T}}{\lambda}\\
&\leq 1 + \frac{10hR_H^2(1 + T\epsilon_G^2/hR_G^2)}{\lambda}\\
&= 1 + \frac{10R_Y^2(hR_G^2 + R_Y^2 T\epsilon_G^2/h)}{\lambda} \leq 1 + c_\lambda^{-1}10R_Y^2.
\end{aligned}$$

Together with Claim I.4, we obtain

$$\sum_{t=1-h}^{T}\mathrm{tr}(\mathbf{Y}_t\Lambda_t^{-1}\mathbf{Y}_t) \leq \frac{2d}{\mu_0\kappa}\mathfrak{L} \leq \frac{2d(1 + 10R_Y^2)}{\kappa}\cdot\mathfrak{L}. \tag{I.10}$$

Thus, putting together Equations (I.8), (I.9), and (I.10),

$$\sum_{t=1}^{T}\sum_{s=0}^{\tau-1}\|\mathbf{Y}_{t-i}(\mathbf{z}_{t-s} - \mathbf{z}_{t-s-1})\|_2 \leq \tau c_\lambda c_\lambda^{-\frac{1}{2}}\cdot L_{\mathrm{eff}}d\sqrt{\frac{2(1 + 10R_Y^2)}{\kappa}}\mathfrak{L},$$

which is the first inequality of the lemma. For the second inequality, we establish the following analogue of Eq. (I.8):

$$\sum_{t=1}^{T}\sum_{s=0}^{\tau-1}\|\mathbf{H}_t(\mathbf{z}_{t-s} - \mathbf{z}_{t-s-1})\|_2 \leq \tau c_\lambda\cdot\left(\sum_{t=1}^{T}\mathrm{tr}(\mathbf{H}_t\Lambda_t^{-1}\mathbf{H}_t)\right)^{\frac{1}{2}}\left(\sum_{t=1}^{T}\|\hat{\nabla}_t\|_{\widehat{\Lambda}_t}^2\right)^{\frac{1}{2}}.$$

Again, we bound $\sum_{t=1}^{T}\|\hat{\nabla}_t\|_{\widehat{\Lambda}_t}^2 \leq dL_{\mathrm{eff}}^2\cdot\mathfrak{L}$ as in Eq. (I.9). Moreover, from Eq. (F.2), we can bound $\sum_{t=1}^{T}\mathrm{tr}(\mathbf{H}_t\Lambda_t^{-1}\mathbf{H}_t) \leq d\mathfrak{L}$. Thus,

$$\sum_{t=1}^{T}\sum_{s=0}^{\tau-1}\|\mathbf{H}_t(\mathbf{z}_{t-s} - \mathbf{z}_{t-s-1})\|_2 \leq \tau dL_{\mathrm{eff}}c_\lambda^{-\frac{1}{2}}c_\lambda\mathfrak{L},$$

which is precisely the second inequality of the lemma.

$\square$

## I.9 Proof of Lemma G.8

We state a slighlty sharper variant of Proposition F.8, which considers directions limited to $\delta \in \mathcal{C} - \mathcal{C}$:

**Claim I.5.** *Set* $c_{\psi;t} := \max\{1, \frac{t\psi_G(h+1)^2}{hR_G^2}\}$. *let* $\delta = z - z'$ *for some* $z, z' \in \mathcal{C}$. *Then,*

$$\delta^\top\left(\sum_{s=1}^{T}\mathbf{H}_t\mathbf{H}_t\right)\delta \geq \frac{\kappa}{2}\delta^\top\left(\sum_{s=1-h}^{T}\mathbf{H}_t\mathbf{H}_t\right)\delta - 20hR_{Y,\mathcal{C}}^2 R_G^2 c_{\psi;t}.$$

*Proof.* The proof is analogous to Proposition F.8, but instead, the remainder term need only account for directiong $z - z'$ for $z, z' \in \mathcal{C}$. This replaces the factor of $R_Y$ one would obtain with a factor of $\max_{t,t'}\|\mathbf{Y}_t\delta_{t'}\| \leq 2R_{Y,\mathcal{C}}$, yielding a remainder temr of $20hR_{Y,\mathcal{C}}^2 R_G^2 c_{\psi;t}$ instead of $5hR_{\mathbf{y}}^2 R_G^2 c_{\psi;t}$ in the original proposition. $\square$

Let us now turn to the proof of our lemma. From Claim I.5, we have

$$\sum_{s=1}^{\tau}\|\mathbf{H}_{k_j}\boldsymbol{\delta}_{k_j+1}\|_2^2 = \boldsymbol{\delta}_{k_j+1}^{\top}\left(\sum_{s=1}^{\tau}\mathbf{H}_{k_j+s}^{\top}\mathbf{H}_{k_j+s}\right)\boldsymbol{\delta}_{k_j+1}$$

$$\geq \frac{\kappa}{2}\boldsymbol{\delta}_{k_j+1}^{\top}\left(\sum_{s=1-h}^{\tau}\mathbf{Y}_{k_j+s}^{\top}\mathbf{Y}_{k_j+s}\right)\boldsymbol{\delta}_{k_j+1} - 20hc_{\psi;\tau}R_G^2R_{Y,\mathcal{C}}^2$$

Moreover, for any $i \in [h]$, we have

$$\sum_{s=1}^{\tau}\sum_{i=0}^{h}\|\mathbf{Y}_{k_j+s-i}\boldsymbol{\delta}_{k_j+1}\|_2^2 = \boldsymbol{\delta}_{k_j+1}^{\top}\left(\sum_{s=1}^{\tau}\mathbf{Y}_{k_j+s-i}^{\top}\mathbf{Y}_{k_j+s-i}\right)\boldsymbol{\delta}_{k_j+1}$$

$$\leq \boldsymbol{\delta}_{k_j+1}^{\top}\left(\sum_{s=1-h}^{\tau}\mathbf{Y}_{k_j+s}^{\top}\mathbf{Y}_{k_j+s}\right)\boldsymbol{\delta}_{k_j+1}.$$

Thus, for $\nu \leq \frac{\kappa}{4}$, we have

$$\sum_{s=1}^{\tau}\sum_{i=0}^{h}\nu(h^{-1}+\mathbb{I}_{i=0})\|\mathbf{Y}_{k_j+s-i}\boldsymbol{\delta}_{k_j}\|_2^2 \leq 2\nu\boldsymbol{\delta}_{k_j}^{\top}\left(\sum_{s=1-h}^{\tau}\mathbf{Y}_{k_j+s}^{\top}\mathbf{Y}_{k_j+s}\right)\boldsymbol{\delta}_{k_j}$$

$$\leq \frac{\kappa}{2}\boldsymbol{\delta}_{k_j}^{\top}\left(\sum_{s=1-h}^{\tau}\mathbf{Y}_{k_j+s}^{\top}\mathbf{Y}_{k_j+s}\right)\boldsymbol{\delta}_{k_j}$$

$$\leq \sum_{s=1}^{\tau}\|\mathbf{H}_{k_j}\boldsymbol{\delta}_{k_j}\|_2^2 + 20hc_{\psi;\tau}R_G^2R_{Y,\mathcal{C}}^2.$$

Hence, rearranging, we have

$$\mathrm{Reg}_{\mathrm{cancel}} := \sum_{j=1}^{j_{\max}}\sum_{s=1}^{\tau}\left(\sum_{i=0}^{h}\left(\nu(1+h\mathbb{I}_{i=0})\|\mathbf{Y}_{k_j+s-i}\boldsymbol{\delta}_{k_j}\|_2^2\right) - \|\mathbf{H}_{k_j+s}\boldsymbol{\delta}_{k_j}\|_2^2\right)$$

$$\leq j_{\max}20hc_{\psi;\tau}R_G^2R_{Y,\mathcal{C}}^2$$

$$\leq \frac{T}{\tau}20hc_{\psi;\tau}R_G^2R_{Y,\mathcal{C}}^2.$$

Finally, let us simplify the dependence on $c_{\psi;\tau}$. We have

$$\frac{c_{\psi;\tau}}{\tau} = \max\{\tau^{-1}, \frac{\psi_G(h+1)^2}{hR_G^2}\} \leq \frac{c_{\psi;\tau}}{\tau} = \max\{\tau^{-1}, \frac{\epsilon_G^2}{hR_G^2}\} \leq \frac{1}{\tau} + \frac{\epsilon_G^2}{hR_G^2}.$$

Together with $\nu \leq \frac{\kappa}{4}$, this gives

$$\mathrm{Reg}_{\mathrm{cancel}} \leq \frac{20\nu h}{\tau}Tc_{\psi;\tau}R_G^2R_{Y,\mathcal{C}}^2 \leq \frac{20\nu h}{\tau}TR_G^2R_{Y,\mathcal{C}}^2 + 20\nu T\epsilon_G^2R_{Y,\mathcal{C}}^2$$

$$\leq \frac{20T}{\tau}\cdot\nu hR_G^2R_{Y,\mathcal{C}}^2 + 5T\epsilon_G^2\cdot\kappa R_{Y,\mathcal{C}}^2.$$

$\square$

## I.10 Proof of Lemma G.9

From Eq. (G.15), we bound

$$\overline{\mathrm{OcoReg}}_T\left(\frac{\nu}{4\eta}; z_\star\right) \leq \frac{1}{4\eta}\mathrm{Reg}_{\mathrm{block}} + T\epsilon_G^2\,\mathrm{ERR}(\nu) + \widehat{\mathrm{Reg}}_T,$$

where from Eq. (G.14) we have

$$\mathrm{Reg}_{\mathrm{block}} \leq 8\tau\cdot\nu R_{Y,\mathcal{C}} + 8\nu R_{Y,\mathcal{C}}\left(\max_{i\in[h]}\mathrm{Reg}_{Y,\mathrm{move},i}\right) + 4R_{Y,\mathcal{C}}R_{\pi_0}\cdot\mathrm{Reg}_{H,\mathrm{move}} + \mathrm{Reg}_{\mathrm{cancel}}.$$

Let us develop the above bound on $\text{Reg}_{\text{block}}$. From Lemma G.7, we have

$$\text{Reg}_{Y,\text{move},i} \le \tau c_\Lambda c_\lambda^{-\frac{1}{2}} \cdot dL_{\text{eff}}\sqrt{\frac{2(1+10R_Y^2)}{\kappa}}\mathfrak{L}, \quad \text{and} \quad \text{Reg}_{H,\text{move}} \le \tau c_\Lambda c_\lambda^{-\frac{1}{2}} \cdot dL_{\text{eff}}\mathfrak{L},$$

and from Lemma G.8, we have $\text{Reg}_{\text{cancel}} \le \frac{20T}{\tau}\cdot \nu h R_G^2 R_{Y,\mathcal{C}}^2 + 5T\epsilon_G^2 \cdot \kappa R_{Y,\mathcal{C}}^2.$. Thus, using followed by

$$\text{Reg}_{\text{block}} \le 8\tau\cdot\nu R_{Y,\mathcal{C}} + 8\nu R_{Y,\mathcal{C}}\left(\max_{i\in[h]}\text{Reg}_{Y,\text{move},i}\right) + 4R_{Y,\mathcal{C}}R_{\pi_0}\cdot\text{Reg}_{H,\text{move}} + \text{Reg}_{\text{cancel}}$$

$$\overset{(i)}{\le} 8\tau c_\Lambda c_\lambda^{-\frac{1}{2}}R_{Y,\mathcal{C}}\left(\nu + \nu dL_{\text{eff}}\sqrt{\frac{2(1+10R_Y^2)}{\kappa}}\mathfrak{L} + dR_G L_{\text{eff}}\mathfrak{L}\right) + \frac{20T}{\tau}\cdot\nu h R_G^2 R_{Y,\mathcal{C}}^2 + 5T\epsilon_G^2\cdot\kappa R_{Y,\mathcal{C}}^2$$

$$\overset{(ii)}{\le} 8\tau c_\Lambda c_\lambda^{-\frac{1}{2}}R_{Y,\mathcal{C}}dL_{\text{eff}}\mathfrak{L}\left(\nu\sqrt{\frac{2(1+10R_Y^2)}{\kappa}} + 2R_G\right) + \frac{20T}{\tau}\cdot\nu h R_G^2 R_{Y,\mathcal{C}}^2 + 5T\epsilon_G^2\cdot\kappa R_{Y,\mathcal{C}}^2$$

$$\lesssim \tau c_\Lambda c_\lambda^{-\frac{1}{2}}R_{Y,\mathcal{C}}dL_{\text{eff}}\mathfrak{L}\left(\nu\sqrt{\frac{2(1+R_Y^2)}{\kappa}} + R_G\right) + \frac{T}{\tau}\cdot\nu h R_G^2 R_{Y,\mathcal{C}}^2 + T\epsilon_G^2\cdot\kappa R_{Y,\mathcal{C}}^2,$$

where $(i)$ uses the above bounds together with $c_\Lambda c_\lambda^{-\frac{1}{2}} \ge 1$ (see Lemma G.6) , and $(ii)$ uses $\nu \le 1 \le L_{\text{eff}}$ and $dR_G\mathfrak{L} \ge 1$, and where the last line disposes of constants. Using $R_G \ge 1$, and the assumption $\nu \le \frac{\sqrt{\kappa}}{4(1+R_Y)}$, the above is at most

$$\text{Reg}_{\text{block}} \lesssim \tau c_\lambda^{-\frac{1}{2}}c_\Lambda R_{Y,\mathcal{C}}R_G dL_{\text{eff}}\mathfrak{L} + \frac{T}{\tau}\cdot\nu h R_G^2 R_{Y,\mathcal{C}}^2 + T\epsilon_G^2\cdot\kappa R_{Y,\mathcal{C}}^2,$$

Next, using $\lambda \ge c_\lambda\tau$, we have from Lemma G.6,

$$c_\Lambda = 2(1+R_Y) + 2R_Y\sqrt{\frac{\tau R_G^2}{\lambda}} \lesssim c_\lambda^{-\frac{1}{2}}(1+R_Y)R_G.$$

Thus, we obtain

$$\text{Reg}_{\text{block}} \lesssim c_\lambda^{-1}\tau(1+R_Y)\cdot R_{Y,\mathcal{C}}R_G^2\cdot dL_{\text{eff}}\mathfrak{L} + \frac{T}{\tau}\cdot\nu h R_G^2 R_{Y,\mathcal{C}}^2 + T\epsilon_G^2\cdot\kappa R_{Y,\mathcal{C}}^2,$$

Combining with $\eta = \frac{3}{\alpha}$, we have

$$\overline{\text{OcoReg}}_T\left(\frac{\nu}{4\eta}; z_\star\right)$$

$$\le \frac{1}{4\eta}\text{Reg}_{\text{block}} + T\epsilon_G^2\,\text{ERR}(\nu) + \widehat{\text{Reg}}_T$$

$$\lesssim c_\lambda^{-1}\tau\left(\alpha(1+R_Y)R_{Y,\mathcal{C}}R_G^2\cdot dL_{\text{eff}}\mathfrak{L}\right) + \frac{T}{\tau}\left(\alpha\nu h R_G^2 R_{Y,\mathcal{C}}^2\right) + T\epsilon_G^2\left(\alpha\kappa R_{Y,\mathcal{C}}^2 + \text{ERR}(\nu)\right) + \widehat{\text{Reg}}_T.$$

Finally, let us substitute in

$$\text{ERR}(\nu) := \frac{\eta(h+1)L_{\text{eff}}^2}{\nu} + \eta\beta^2(c_v + 2R_{Y,\mathcal{C}})^2 + \frac{4R_{Y,\mathcal{C}}}{\eta}$$

$$\lesssim \frac{hL_{\text{eff}}^2}{\alpha\nu} + \frac{1}{\alpha}\beta^2(c_v^2 + R_{Y,\mathcal{C}}^2) + \alpha R_{Y,\mathcal{C}}.$$

Since $\alpha \le \beta$ by necessitiy and $\kappa \le 1$, we have $\alpha \le \frac{\beta^2}{\alpha}$, so that

$$\text{ERR}(\nu) + \alpha\kappa R_{Y,\mathcal{C}}^2 \lesssim \frac{hL_{\text{eff}}^2}{\alpha\nu} + \frac{\beta^2}{\alpha}(c_v^2 + R_{Y,\mathcal{C}} + R_{Y,\mathcal{C}}^2)$$

Altogether, combined with the bound $c_\lambda \le 1$, this yields

$$c_\lambda\overline{\text{OcoReg}}_T(\frac{\nu}{4\eta}; z_\star) \lesssim \frac{T\epsilon_G^2}{\alpha}\left(\frac{hL_{\text{eff}}^2}{\nu} + \beta^2(c_v^2 + R_{Y,\mathcal{C}} + R_{Y,\mathcal{C}}^2)\right) + \widehat{\text{Reg}}_T.$$

$$+ \frac{T\nu}{\tau}\left(\alpha h R_G^2 R_{Y,\mathcal{C}}^2\right) + \tau\cdot\left(\alpha(1+R_Y)R_{Y,\mathcal{C}}R_G^2\cdot dL_{\text{eff}}\mathfrak{L}\right),$$

as needed.

$\square$.

### I.11  Proof of Lemma G.10

Consider $\text{MoveDiff}_T := \sum_{t=1}^{T} F_t(\mathbf{z}_{t:t-h}) - f_t(\mathbf{z}_t)$. The decomposition Lemma F.6 holds verbatim, and by appropriately modifying Lemma F.7 to use the fact that the iterates are based on $\widehat{\mathbf{H}}_t$, $\widehat{\Lambda}_t$, we arive at.

$$\text{MoveDiff}_T \leq \eta h^2 L_{\text{eff}}^2 R_G \cdot \sqrt{\sum_{t=1-h}^{T} \text{tr}(\mathbf{Y}_t \widehat{\Lambda}_t^{-1} \mathbf{Y}_t)} \cdot \sqrt{\sum_{t=1}^{T} \text{tr}(\widehat{\mathbf{H}}_t^\top \widehat{\Lambda}_t^{-1} \widehat{\mathbf{H}}_t)}.$$

As in Eq. (F.2), we bound

$$\sum_{t=1}^{T} \text{tr}(\widehat{\mathbf{H}}_t^\top \widehat{\Lambda}_t^{-1} \widehat{\mathbf{H}}_t) \leq d \log(1 + \frac{TR_H^2}{\lambda}) \leq d\, \mathfrak{L},$$

where we take $\lambda \geq 1$ and use $\mathfrak{L} = \log(1+TR_H^2/\lambda)$ from Eq. (G.6). Moreover, applying Lemma G.6 with $\tau = 0$, we have that $\widehat{\Lambda}_t^{-1} \preceq 4(1 + R_Y)^2 \Lambda_t^{-1}$, giving

$$\sum_{t=1-h}^{T} \text{tr}(\mathbf{Y}_t \widehat{\Lambda}_t^{-1} \mathbf{Y}_t) \leq 4(1+R_Y)^2 \sum_{t=1-h}^{T} \text{tr}(\mathbf{Y}_t \Lambda_t^{-1} \mathbf{Y}_t) \leq 4(1+R_Y)^2 \frac{2d(1+10R_Y^2)}{\kappa} \cdot \mathfrak{L}$$

where the last inequality uses Eq. (I.10). Thus,

$$\text{MoveDiff}_T \leq 9\eta(1 + R_Y) h^2 d L_{\text{eff}}^2 \mathfrak{L} R_G \cdot \sqrt{(1 + R_Y^2)/\kappa}$$
$$\leq 9\eta \kappa^{-\frac{1}{2}}(1 + R_Y)^2 R_G h^2 \cdot d L_{\text{eff}}^2 \mathfrak{L}$$

$\square$

## J  Lower and Upper Bounds on Euclidean Movement

### J.1  Proof of Theorem 2.3

Our construction is loosely based of of [5, Theorem 13].

Recall the lower bound set up $\mathcal{C} = [-1, 1]$, $f_t(z) = (\mathbf{v}_t - \epsilon z)^2$, and $\epsilon \leq 1$. Let $E$ be an epoch length to be selected, and suppose for simplicity that $k = T/E$ is an integer. Let $T_i := 1 + E \cdot (i - 1)$ denote the start of each epoch for $i \geq 1$. Let us define the distribution $\mathcal{D}$ over $\mathbf{v}_1, \ldots, \mathbf{v}_T$ via:

$$\mathbf{v}_t := \begin{cases} \overset{\text{i.i.d}}{\sim} \text{Unif}(\{-1, 1\}) & t = T_i \\ \mathbf{v}_{T_i} & t \in \{T_i + 1, \ldots, T_{i+1} - 1\} \end{cases}$$

Lastly, recall the definition:

$$\mu\text{-Reg}_T := \sum_{t=1}^{T} f_t(\mathbf{z}_t) - \inf_{z \in \mathcal{C}} \sum_{t=1}^{T} f_t(z) + \mu \sum_{t=1}^{T} |\mathbf{z}_{t-1} - \mathbf{z}_t|$$

Our key technical ingredient is the following lemma, which shows that if the regularizer is large enough, the optimal strategy is essentially to select $\mathbf{z}_t = \mathbf{z}_{T_i}$ within any given epoch $i$:

**Lemma J.1.** *For $\mu \geq 4E\epsilon$,*

$$\sum_{t=T_i+1}^{T_{i+1}-1} f_t(\mathbf{z}_t) + \mu|\mathbf{z}_t - \mathbf{z}_{t-1}| \geq (E - 1) f_t(\mathbf{v}_{T_i} - \mathbf{z}_{T_i}).$$

*Proof.* We can write

$$\sum_{t=T_i+1}^{T_{i+1}-1} f_t(\mathbf{z}_t) + \mu|\mathbf{z}_t - \mathbf{z}_{t-1}| = \sum_{t=T_i+1}^{T_{i+1}-1} f_{T_i}(\mathbf{z}_t) + \mu|\mathbf{z}_t - \mathbf{z}_{t-1}|$$

$$\geq \sum_{s=1}^{E-1} f_{T_i}(\mathbf{z}_t) + \mu \cdot \max_{t=T_i+1,\ldots,T_{i+1}-1} |\mathbf{z}_{T_i} - \mathbf{z}_t|$$

$$\geq \sum_{s=1}^{E-1} f_{T_i}(\mathbf{z}_t) + \underbrace{\frac{\mu}{E-1}|\mathbf{z}_{T_i} - \mathbf{z}_t|}_{:=g(\mathbf{z}_t)},$$

where the first inequality uses the triangle inequality, and the second replaces the maximum by the average. Define $\mu_0 = \frac{\mu}{2(E-1)\epsilon}$, and set $g(z) := f_{T_i}(\mathbf{z}_t) + \frac{\mu}{E-1}|\mathbf{z}_{T_i} - \mathbf{z}_t| = (\mathbf{v}_{T_i} - \epsilon z)^2 + 2\epsilon\mu_0|\mathbf{z}_{T_i} - \mathbf{z}_t|$. Then,

$$\partial g(z) = 2\epsilon\left(\epsilon z - \mathbf{v}_{T_i} + \mu_0\sigma(z)\right)$$

where $\sigma(z) = 1$ if $\mathbf{z}_{T_i} > z$, $-1$ if $\mathbf{z}_{T_i} < z$, and is in interval $[-1, 1]$ if $z = \mathbf{z}_{T_i}$. Now, if $\mu_0 \geq 2$, then, $|\epsilon z - \mathbf{v}_{T_i}| \leq \mu_0$, so that the first order optimality conditions are met by selecting $z^\star = \mathbf{z}_{T_i}$. This yields

$$g(z^\star) = (\mathbf{v}_{T_i} - \epsilon\mathbf{z}_{T_i})^2.$$

The bound follows. □

By summing within different epochs, the above lemma implies a simple lower bound on $\mu\text{-Reg}_T$:

$$\mu\text{-Reg}_T = \sum_{i=1}^{k}\sum_{t=T_i}^{T_{i-1}+1} f_t(\mathbf{z}_t) - f_t(z) + \mu\|\mathbf{z}_t - \mathbf{z}_{t-1}\|$$

$$\overset{(i)}{=} \sum_{i=1}^{k} f_{T_i}(\mathbf{z}_{T_i}) - Ef_{T_i}(z) + \mu\|\mathbf{z}_{T_i} - \mathbf{z}_{T_i-1}\| + \left(\sum_{t=T_i}^{T_{i-1}+1} f_t(\mathbf{z}_t) + \mu\|\mathbf{z}_t - \mathbf{z}_{t-1}\|\right)$$

$$\overset{(ii)}{\geq} \sum_{i=1}^{k} f_{T_i}(\mathbf{z}_{T_i}) - Ef_{T_i}(z) + \mu\|\mathbf{z}_{T_i} - \mathbf{z}_{T_i-1}\| + (E-1)f_{T_i}(\mathbf{z}_{T_i})$$

$$\geq \sum_{i=1}^{k} f_{T_i}(\mathbf{z}_{T_i}) - Ef_{T_i}(z) + (E-1)f_{T_i}(\mathbf{z}_{T_i})$$

$$= \sup_{z\in\mathcal{C}} E\left(\sum_{i=1}^{K} f_{T_i}(\mathbf{z}_{T_i}) - f_{T_i}(z)\right),$$

where $(i)$ uses that $f_t = f_{T_i}$ in epoch $i$ and $(ii)$ uses Lemma J.1. Crucially, the above quantity is scaled up by a factor of $E$, and the learner is forced to commit to a single iterate per epoch. Continuing with $f_{T_i}(z) = (\mathbf{v}_{T_i} - \epsilon z)^2$,

$$\mu\text{-Reg}_T \geq \sup_{z\in\mathcal{C}} E\left(\sum_{i=1}^{k}(\mathbf{v}_{T_i} - \epsilon\mathbf{z}_{T_i})^2 - (\mathbf{v}_{T_i} - \epsilon z)^2\right)$$

$$= \sup_{z\in\mathcal{C}} E\left(\sum_{i=1}^{k} -2\epsilon\mathbf{v}_{T_i}\mathbf{z}_{T_i} + \underbrace{\epsilon^2\mathbf{z}_{T_i}^2}_{\geq 0} + 2\epsilon z\mathbf{v}_{T_i} - \epsilon^2 \cdot \underbrace{z^2}_{\leq 1}\right)$$

$$\geq \sup_{z\in\mathcal{C}} E\left(\left(\sum_{i=1}^{k} -2\epsilon\mathbf{v}_{T_i}\mathbf{z}_{T_i} + 2\epsilon z\mathbf{v}_{T_i}\right) - k\epsilon^2\right).$$

Taking an expectation, and noting that $\mathbb{E}[\mathbf{v}_{T_i} \mathbf{z}_{T_i}] = 0$ by construction, we have that

$$\mathbb{E}[\mu\text{-Reg}_T] \geq E\left(2\epsilon\mathbb{E}\left[\sup_{z \in \mathcal{C}} z \sum_{i=1}^{k} \mathbf{v}_i\right] - k\epsilon^2\right)$$

$$= 2\epsilon E\left(\mathbb{E}\left|\sum_{i=1}^{k} \mathbf{v}_i\right| - \frac{k}{2}\epsilon\right)$$

$$\geq 2\epsilon E\left(c\sqrt{k} - \frac{k\epsilon}{2}\right),$$

where $c \leq 1$ is a universal constant. [8] Let us now tune the above bound. Select

- $k = \lfloor (8Tc/\mu)^{2/3} \rfloor$
- $\epsilon = \mu/4E$.

We first check that these parameters are valid:

**Claim J.2.** *For a universal constant $c_1$, it holds that if $\mu \leq c_1 T$, then $k \geq 1$ and $\epsilon \leq 1$.*

*Proof.* For $\mu \leq 8cT$, $k \geq 1$. Moreover,

$$\epsilon = \frac{\mu}{4E} = \frac{\mu k}{4T} \leq (8Tc/\mu)^{2/3}\frac{\mu}{T} = 4c^{2/3}(\mu/T).$$

Hence, for $\mu \leq T/4c^{2/3}$, the above is at most 1. Setting $c_1 = \min\{8c, 1/4c^{2/3}\}$ concludes. $\quad\square$

For the above choices, we have

$$\mathbb{E}[\mu\text{-Reg}_T] \geq 2\epsilon E\left(c\sqrt{k} - \frac{k\epsilon}{2}\right)$$

$$= \frac{\mu}{2}\left(c\sqrt{k} - \frac{k^2\mu}{8T}\right) \quad = \frac{c\sqrt{k}\mu}{4}\left(2 - \frac{k^{3/2}}{8Tc/\mu}\right)$$

$$\geq \frac{c\sqrt{k}\mu}{4} \quad \geq \frac{c\mu\lfloor (8Tc/\mu)^{2/3} \rfloor^{1/2}}{4}$$

$$\geq c_2\mu(T/\mu)^{1/3} = c_2(\mu^2 T)^{1/3},$$

for some universal constant $c_2$. Moreover, suppose that that $\mathbb{E}[\text{OcoReg}_T] \leq R$. Then, for $\mu \geq c_1 T$

$$c_2(\mu^2 T)^{1/3} \leq \mathbb{E}[\mu\text{-Reg}_T] \leq R + \mu\mathbb{E}[\text{EucCost}_T].$$

Rearranging, we have that if $c_2(\mu^2 T)^{1/3} \geq 2R$, $\mathbb{E}[\text{EucCost}_T] \geq \frac{c_2}{2}(T/\mu)^{1/3}$. For this to hold, we take $\mu = \sqrt{(2R/c_2)^3/T}$, yielding

$$\mathbb{E}[\text{EucCost}_T] \geq \frac{c_2}{2}(T \cdot (T/(2R/c_2)^3)^{1/3} = \frac{c_2}{2}(c_2 T/2R)^{1/2} \geq c_3\sqrt{T/R}. \tag{J.1}$$

Finally, we need to ensure that $\mu \leq c_1 T$, which hold for $(2R/c_2)^3/T \leq c_1^2 T^2$, i.e. for $R \leq c_4 T$ for a universal $c_4$.

## J.2 Matching Tradeoff via ONS

We now show that ONS mathces the tradeoff in Theorem 2.3 up to logarithmic factors, problem constants and dimension. To show this, we first check that OCOAM losses satisfy the general ONS regularity conditions. We say $f$ is $\tau$-exp concave if $\nabla^2 f \succeq \tau \cdot \nabla f(\nabla f)^\top$ [18]. The following is a direct consequence of Lemma F.2

**Lemma J.3.** *Let $f_t$ be an OCOAM loss with parameters bounded as in Definition 2.1, where $\ell$ satisfies Assumption 1. Then $f_t$ is $\frac{\alpha}{L_{\text{eff}}^2}$-exp concave, and $R_H L_{\text{eff}}$-Lipschitz on $\mathcal{C}$.*

We now show that ONS matches the optimal $(\mu^2 T)^{1/3}$ scaling up to dimension and logarithmic factors:

**Theorem J.1.** *Consider* ONS *on a sequence family of $G$-Lipschitz, $\tau$-exp concave functions on a convex set $\mathcal{C}$ of diameter $D$. Let define $R_0 = (GD + \tau^{-1}) \cdot d \log T$ be the standard upper bound (up-to-constants) on the regret of* ONS *[18]. Then, for any $\mu \in \mathbb{R}$, there exists a choice of regularization parameter $\lambda$ such that* ONS *with $\eta = 2 \max\{4GD, 1/\tau\}$ has:*

$$\mu\text{-Reg}_T \lesssim (R_0 D^2 \cdot T\mu^2)^{1/3} + R_0.$$

For the special case of OCoAM, the above guarantee can also be satisfied for by Semi-ONS(albeit with modififed dependence on problem parameters ).

Consider the ONS algorithm, with updates

$$\tilde{\mathbf{z}}_{t+1} = \mathbf{z}_t - \eta \Lambda_t^{-1} \nabla_t, \quad \mathbf{z}_{t+1} = \underset{z \in \mathcal{C}}{\arg\min} \|\tilde{\mathbf{z}}_{t+1} - z\|_{\Lambda_t}^2, \quad \Lambda_t := \lambda I + \sum_{s=1}^{t-1} \nabla_t \nabla_t^\top, \quad \nabla_t := \nabla f_t(\mathbf{z}_t) \tag{J.2}$$

Set $\eta = 2\max\{4GD, 1/\tau\}$, $\lambda \geq G^2$. From Hazan [18, Section 4.3], with the notation change $\eta \leftarrow 1/\gamma$, $\tau \leftarrow \alpha$ $\Lambda_t \leftarrow A_t$, and $\lambda \leftarrow \epsilon$, ONS has unary regret bouned by

$$\text{OcoReg}_T \leq \frac{\eta}{2} \sum_{t=1}^T \nabla_t \Lambda_t^{-1} \nabla_t + \frac{D^2 \lambda}{2\eta} \leq \frac{d\eta}{2} \log(1+T) + \frac{D^2 \lambda}{2\eta}.$$

Moreover, we can bound

$$\text{EucCost}_T = \sum_{t=1}^T \|\mathbf{z}_t - \mathbf{z}_{t-1}\| \overset{(i)}{\leq} \frac{1}{\sqrt{\lambda}} \sum_{t=1}^T \|\Lambda^{1/2}(\mathbf{z}_t - \mathbf{z}_{t-1})\|$$

$$\overset{(ii)}{\leq} \frac{1}{\sqrt{\lambda}} \sum_{t=1}^T \|\Lambda^{1/2}(\tilde{\mathbf{z}}_t - \mathbf{z}_t)\| \quad = \frac{1}{\sqrt{\lambda}} \sum_{t=1}^T \|\Lambda^{-1/2} \nabla_t\|$$

$$\overset{(iii)}{\leq} \frac{\eta}{\sqrt{\lambda}} \sqrt{T \sum_{t=1}^T \nabla_t^\top \Lambda_t^{-1} \nabla_t} \quad \overset{(iv)}{\leq} \frac{\eta}{\sqrt{\lambda}} \sqrt{Td \log(1+T)},$$

where $(i)$ uses $\Lambda_t \succeq \lambda$, $(ii)$ uses the Pythagorean theorem, $(iii)$ uses Cauchy-Schwartz, and $(iv)$ applies the log-determinant lemma as in Hazan [18, Section 4.3] with $\lambda \geq G^2$. Hence,

$$\mu\text{-Reg}_T \leq \frac{d\eta}{2} \log(1+T) + \frac{D^2 \lambda}{2\eta} + \frac{\eta\mu}{\sqrt{\lambda}} \sqrt{Td \log(1+T)}.$$

Set $\lambda_0$ to satisfy $\frac{D^2 \lambda_0}{2\eta} = \frac{\eta\mu}{\sqrt{\lambda_0}} \sqrt{Td \log(1+T)}$. Then,

$$\frac{D^2 \lambda_0}{2\eta} + \frac{\mu}{\sqrt{\lambda_0}} \sqrt{Td \log(1+T)} = \frac{D^2 \lambda_0}{\eta}$$

$$= \frac{D^2}{\eta} \cdot \left(\frac{2\eta^2}{D^2} \mu \sqrt{Td \log(1+T)}\right)^{2/3}$$

$$= \frac{D^2}{\eta} \cdot \left(\frac{2\eta^4}{D^4} \mu^2 Td \log(1+T)\right)^{1/3}$$

$$= \left(2D^2 \cdot \mu^2 T \cdot \eta d \log(1+T)\right)^{1/3}.$$

Setting $\lambda = G^2 \vee \lambda_0$ yields

$$\mu\text{-Reg}_T \leq \frac{d\eta}{2} \log(1+T) + \frac{G^2 D^2}{2\eta} + \frac{d\eta}{2} \log(1+T) + \frac{D^2 \lambda_0}{2\eta} + \frac{\mu}{\sqrt{\lambda_0}} \sqrt{Td \log(1+T)}$$

$$\leq \frac{d\eta}{2} \log(1+T) + \frac{G^2 D^2}{2\eta} + \left(2D^2 \cdot \mu^2 T \cdot \eta d \log(1+T)\right)^{1/3}.$$

Subsititing in $\eta = 2\max\{4GD, 1/\tau\}$, and defining $R_0 = \max\{GD, 1/\tau\} \cdot d \log(1+T)$ gives that the above is at most

$$\mu\text{-Reg}_T \lesssim (R_0 D^2 \cdot T\mu^2)^{1/3} + R_0.$$