[Reviews · NeurIPS 2020]

Review 1

Summary and Contributions: This paper provides the first sublinear-regret algorithms in the non-stochastic online control setting with truly adversarial disturbance and costs and partial state observation. With known dynamics, the regret is logarithmic, and with unknown dynamics, the regret becomes \sqrt{T}. The main contribution of this paper is to prove that it is possible to achieve the same order of regret in the adversarial setting as in the stochastic one. Another contribution of this paper is from the development and analysis of the general-purpose optimization subroutine (Semi-ONS).

Strengths: This paper solidly and thoroughly studies the full generality of the non-stochastic control setting, which is definitely related to our community's interests. The theoretical results are clearly novel and some of the techniques are also novel (e.g., Semi-ONS). This paper potentially sheds light on the fundamental limits in online control.

Weaknesses: a) The biggest issue of this paper is that the presentation and organization need a few rounds of tunning, and it is (almost) impossible to understand this paper without thoroughly reading previous work and appendix. Notations and definitions are too dense. Also, there are only very weak connections between the technique introductions in Section 1.1 and the later sections. Especially, I miss the intuition behind DRC-ONS. Moreover, this paper spent more than one page on policy regret for Semi-ONS, which I feel should be studied in another parallel paper. Honestly, even if with one more page in the camera-ready stage, I doubt that this paper can include all the technical contents currently involved, in a clear manner. b) In my opinion, the claims "non-stochastic control is as easy as stochastic" is a bit misleading. The benchmark in the stochastic setting has totally different performance compared to the benchmark in the non-stochastic setting. In particular, one big reason why the non-stochastic setting can achieve the same order of regret as the stochastic setting is, the non-stochastic setting has a much weaker benchmark. In the stochastic setting, linear policy class is proved to be globally optimal, which means as long as some algorithm has sublinear regret, the average cost difference between it and the globally optimal policy quickly vanishes. However, in the non-stochastic setting, the globally optimal policy is not necessarily linear and static, which means sublinear regret doesn't guarantee global performance. Even if we generalize state to internal state as in this paper, the benchmark is still not global. Some recent works (https://arxiv.org/abs/2002.05318, https://arxiv.org/abs/2002.02574) show that there is at least a linear regret between the offline best linear policy and the true offline global optimal in the adversarial setting. For example, in LQ tracking problem, the cost function will be (x_t-d_t)^T Q (x_t-d_t) + u_t^T R u_t, where {d_t} is some desired trajectory to track. Obviously, in this case, simple linear policy u = -Kx will perform poorly - we have to consider feedforward term for d in the policy. In summary, in terms of static regret minimization with linear policy class, stochastic and non-stochastic settings indeed have the same order of regret. However, it doesn't mean the non-stochastic setting is as easy as the stochastic one. c) I feel the technical part needs more discussion about the relationship between previous work. For example, this paper almost has the same setting as [24], but generalize the notion of the adversary. Also, the algorithm DRC-ONS just replaces the optimization subroutine of DRC-GD in [24]. I miss the discussion about this relationship, especially why DRC-ONS is critical to handle the true adversary. d) It's hard to tell the proposed algorithm DRC-ONS is practical or not... It would be great to comment on the complexity of it.

Correctness: I didn't check each line in the 40-page appendix, but most of the claims are reasonable. The discussion about the relationship between non-stochastic and stochastic control is a bit misleading (see the weakness part).

Clarity: In general, this paper is too dense and it is (almost) impossible to fully understand without reading previous works.

Relation to Prior Work: In general yes, and this paper provides the first result in the full adversarial setting. However, I miss more comparisons from previous contributions (see the weakness part).

Reproducibility: Yes

Additional Feedback: After rebuttal: After reading all the reviews and the response. I would like to stay with my current evaluation (score: 6). Clearly I see the contribution and novelty from this paper and it seems the authors do have a plan to improve presentation and organization. However, I think this change is beyond NeurIPS camera-ready wiggle zoom and I doubt 9 pages are enough to motivate and convince non-experts. I feel like even only presenting the novel online optimization algorithm Semi-ONS might be a better idea. Presenting with the current structure in NeurIPS may not have enough value to non-experts.


Review 2

Summary and Contributions: This paper builds upon Simchowitz [24], improving the regret bounds of non-stochastic control. (See Thm 3.1, 3.2.) One of its main contributions is the use of an online Newton step, as in algorithm table.1 and 2.

Strengths: I can see the incremental contribution over Simchowitz [24]. The improved rate results in this paper may have the potential of being relevant to researchers working on the adaptive "non-stochastic" control setting.

Weaknesses: The writing and organization should be vastly improved before publication. The current paper is hard to read even for readers knowledgable in control. It contains numerous typos (see a non-exhaustive list below). I have a hard time seeing researchers outside "non-stochastic" control understand the paper.

Correctness: Yes, as far as I can tell from the manuscript.

Clarity: See below.

Relation to Prior Work: Yes. However, it did not refer to any adaptive control references which are relevant to online LQR.

Reproducibility: Yes

Additional Feedback: ============= Update after rebuttal: I thank the reviewers' response and willingness to improve the presentation. I have increased my rating and adjusted my confidence score. ============= ## Some suggestions To help improve the paper's readability, I recommend starting the paper by motivating the setting, instead of directly diving into the non-stochastic control results. For example, since the paper is categorized as control theory, how is the paper's setting relevant to control applications? In particular, abstract should be more "abstract" and appeal to a wider audience, not discussing related work directly. In what practical control applications of control does the non-stochasticity arises, that current robust/stochastic/adaptive control cannot address? I recommend detailing the assumptions on the disturbances when the system model is introduced in eq 1.1, e.g., are they bounded? By what? (and other assumptions of the system.) There are mentioning of various versions (e.g., stochastic noise, bounded noise) in various places in the paper. System model is crucial to control designs. When introducing "crucial identity" such as 2.1, it helps to provide an intuitive explanation of what it means and why it holds. The paper contains no numerical example. This is usually okay for a pure math/theory paper. However, this paper proposed the contribution of using an online Newton step. Therefore, I recommend at least a simple numerical example (e.g., LQ control) to demonstrate this numerical method. Adding a conclusion is also recommended for improving the readability. --- ## Some questions - Can the authors please concisely explain what the difference between non-stochastic control and robust control is? Is it that the loss for non-stochastic control is "adversarially chosen"? Note: While there is the common saying that classical robust control is pessimistic since it's worst-case, but adversarial is a synonym for worst-case. Merely saying "adversarially chosen" is not informative since if the adversary is infinitely powerful, there can be no hope for control. To what extent is noise/loss in this paper adversarial? Please explain it when using the phrase "adversarially chosen" in the paper. - In what practical control applications the non-stochasticity arises, that current robust and stochastic control cannot address? (adding this to the paper will readers understand the significance of the author's work. The description in "broader impact" is vague. It is hard to understand what "purely stochastic models" refer to. A more concrete example would help. ) In conclusion, while I can see the incremental progress made over Simchowitz [24] and that the improved rate results in this paper may have the potential of being relevant to researchers working on the adaptive "non-stochastic" control setting, the current manuscript - lacks a simple numerical evaluation of the online Newton step - must improve its organization/writing - has too many typos and incoherent sentences hence is not ready for publication. --- PS. Please proofread the paper once to catch other typos. --- ### Non-exhaustive list of typos 46: Generalizating? 50: observer --> observe 55: incoherent sentence 98: first half sentence incoherent 79: show --> shows 104: depart --> depart from 186: exogenous output --> exogenous input 207: show that for that ??? 209: algorithm with attains??? 228: is a is a??? 253: section title incoherent


Review 3

Summary and Contributions: This paper considers the problem of controlling a partially observed linear dynamical systems when the disturbances in the system are non-stochastic and the costs are strongly convex. The authors propose an adaptive control algorithm called DRC-ONS. The control at each time is determined via a combination of linear feedback term and an exogenous input. The exogenous input is a linear combination of the past $m$ observations under the stabilizing control policy where the weighting matrices are obtained via an online semi-newton procedure. The authors show that the regret of their algorithm over a time horizon $T$ is of the order $O(poly(log T))$ when the dynamics are known and of the order $O(\sqrt{T})$ when the dynamics are unknown.

Strengths: The proposed algorithm seems novel and non-trivial. The paper has rigorous theoretical guarantees for the regret of the proposed scheme. Moreover, the paper gets the state of the art regret bounds for their particular setting of non-stochastic control problem.

Weaknesses: One of the limitation of the main result is that the memory ($m$) of the DRC controller needs to scale as O(log T). Therefore, as $T$ grows the algorithm requires larger memory for the regret bound to hold true. Also, since the algorithm involves a second order newton step it needs to carry out a matrix inversion of $m*d_y*d_u$ dimensional square matrix. This makes the computational complexity of each step of the algorithm at least O( (log T)^3 ).

Correctness: The theoretical claims in the paper seem correct to me.

Clarity: I found the paper hard to read. It was hard to keep track of the notation and the lack of a notation section made it harder to recall the definitions. There are a few typos and grammatical errors that need to be fixed.

Relation to Prior Work: The paper discusses the prior work and its contributions. However, it will be nice if the authors can expand upon the similarities and the differences between this work and reference [24] since lot of analysis and results seem to have been taken from [24]

Reproducibility: Yes

Additional Feedback: Following are some comments and questions: 1) Reference [24] has a similar adaptive control algorithm but uses gradient descent for the finding the weighting matrices $M_t$. Is there any fundamental reason why gradient descent will not work for the fully adversarial setting? 2) Should the first equation in eq 2.1 include $u^{ex}_{t-i}$ instead of $u^{ex}_{t}$? 3) In assumption 2, how is $G_K^i$ dependent on n? 4) What choice of set $\mathcal{C}$ is used in Algorithm 2 for the control problem? What is the complexity of solving the minimization in step 8 of algorithm 2?


Review 4

Summary and Contributions: This work considers online control of linear dynamical systems under adversarial noise. For unknown systems, it obtains the optimal sqrt(T) regret and polylog(T) regret in known systems. This result extends the prior work which achieves the same order regret in a semi-adversarial setting, where there exists stochastic perturbation.

Strengths: -The technical results are sound and significantly non-trivial. There are various results and generalizations presented in the main text and mainly in the Appendix. -The regret results are optimal in terms of time.

Weaknesses: -The presentation of paper could be improved in some places. Due to heaviness of the technical content and the limited number of pages, some of the concepts are not well-discussed. This may bring up the question whether this work is suitable for 8 (if accepted 9) pages conference publication since most of the material is deferred to the appendix. In some parts, the paper seems inaccessible. -The lack of experimental study. It would be immensely valuable to provide a numerical study for the derived results to show the applicability of the proposed algorithms.

Correctness: I went over the general flow and statements of the proofs. They seem sound.

Clarity: Overall paper is well-written and tries to provide intuitive explanations. However, due to space constraint the explanations are limited and somewhat inaccessible and mostly deferred to Appendix which makes it hard to follow the arguments. For example, the discussion of movement cost and f-regret in line 130-138 rather obfuscates the message and the flow of the paper. Moreover, there are some typos in various places of the draft. Due to technical content, the notation is a bit overloaded.

Relation to Prior Work: The related work is very well discussed. As the authors mention at footnote 3, it would be useful to bring the discussion in Appendix B.3 to the main text if accepted.

Reproducibility: Yes

Additional Feedback: This work fills one of the gaps in recent online control results in literature and generalizes the prior work to a more difficult setting. I believe the authors have a deep understanding of system identification, control theory and online optimization. Hence, all of their results are derived with due diligence. My main concern is the accessibility of the results and techniques by the non-experts due to space constraints and lack of exact explanations. For example, the proof sketches on page 8 try to explain the steps in detail but without looking at the proofs in detail, I think they do not give much intuition to the reader. The fact that most of the results are deferred to Appendix makes the main text somewhat inaccessible. Another point is the lack of experiments. Even though the work is mainly theoretical, I believe that providing proper numerical examples are important. One can then judge the validity of the results and more importantly the limitations of the claims. Technical questions: - In line 191 the lower bound on R_M is missing. What is it suppose to be? In Appendix (D.8) states that R_M >= 2R_* and this fact is denoted as overparameterization. Does this mean we need prior information of R_* and construct \mathcal{M} for the algorithm such that the condition above holds or do we need to \mathcal{M} to be given a prior? Because as far as I understand the technical result for unknown system heavily relies on this fact. - lambda is set to be O(sqrt(T)). If I'm not mistaken this lambda is the regularization term. Does this mean we need the knowledge of time horizon a priori? I assume that this comes into play in the proof. How important is this choice? -The algorithm follows an explore-and-commit strategy for the unknown system. Would that be possible to have the same algorithm in more adaptive sense? - As far as I understood, the invertibility modulus exists for fixed stabilizing controller and Remark D.2 states that there are not general lower bounds for dynamic ones (please correct me if I'm wrong). Does this mean the result may not work for some LQG setting? ========================================================== After rebuttal: Thanks for your responses and clarifications. There are many points brought up by the reviewers that still need to be clarified in the manuscript before publication. Due to space constraints, it doesn't seem feasible to achieve the clarity and include the required discussions/numerical examples in the main text. Even though there is significant contribution, the fully accessible version of the manuscript requires significant changes and would not seem to be able to fit NeurIPS page constraints. Thus, I lower down my score to 6.

[Author Response · NeurIPS 2020]

Dear reviewers, we are deeply grateful for all your helpful feedback. Your reviews will be invaluable in revising the manuscript, regardless of its acceptance. We first address common concerns, and then particular reviewers. **Organization & Accessibility:** All reviewers expressed concerns about the accessibility of the paper to non-experts, and pointed out that the structure contributed to the obfuscation. The intent of the "our techniques" section was to explain, at an intuitive level, the main technical challenges we overcome. However, to the reader unfamiliar with the control-to-policy-regret reduction [3,12,24] , we now recognize this may be confusing. Moreover, the paper requires background from both the control theory community *and* the online learning community; and whereas are main results pertain to the former, are techniques and technical novelty relate to the latter. To better modularize the paper, we plan to restructure Section 2 to describe *only* the problem of attaining low policy regret for online convex optimization with memory. This requires limited notational overheard compared to the control setting. The section will both state and analyze our online algorithm optimization algorithm Semi-ONS. The "our techniques" section can then be woven into a complete proof, as the proof of logarithmic regret of Semi-ONS is quite direct. The full proof of $\sqrt{T}$ regret for approximate system knowledge will remain deferred to the appendix, but the sketch will be lengthened. Notably, this section will be accessible to online learning theorists without control knowledge, and will highlight the novelty of our techniques. Section 3 will apply Section 2 to the control problem. Having seen the OCO-with-memory formalism, the control-oriented reader will now understand the motivation for expressing the problem in the way we do. With the additional page of space, we will state formal black-box reductions from control-to-OCO-with-memory from the Appendix. The will allow readers unfamiliar with past work to understand how our innovations in online learning translate to control, even if they wish to skip the details of those regret bounds. The reader familiar with relevant prior work can skip the reduction. **Clarity, Typos, Experiments:** We apologize for the numerous grammatical errors, and mis-spliced sentences. The manuscript went through many rounds of restructuring, and we may not have caught all errors that arose as a consequence. We will be sure to adress all typos in the final version. Regarding experiments: past work on non-stochastic control does not include experiments, and we followed this convention. Nevertheless, we will attempt to include a simple demonstration in the final manuscript comparing Newton to Gradient-Based methods. Note that we propose Newton as an *learning procedure*, not simply an optimization subroutine. **Reviewer 1:** *(a)* See above. *(b)* We agree that, from a perspective of optimal control, online non-stochastic control is much harder than stochastic , as the tracking problem and known lower bounds elucidate. We initially had a sentence to that effect, which must have been mistakenly removed. We will be sure to clarify this point in further revision. However, for the narrow definition of non-stochastic control introduced by [3], where regret to a fixed benchmark of LTI controllers is defined as the objective, our paper does indeed demonstrate that that the regret rates for this problem coincide with stochastic. Secondly, at multiple points throughout the paper, we stress that while our algorithm uses a static $K$ for the parametrization, our benchmark are linear *dynamic* controls with internal state, which may fare well in many tracking tasks (e.g. targets generated by an LDS). The arXiv of [24] describes numerous other classes of control policies compatible with the DRC formalism, which may be better suited to various tasks. *(c)* This is addressed briefly in lines 110-114, but can be expanded upon in the appendix. *(d)* the algorithm admits an efficient implementation of maintaining the matrix inverse via the Woodbury identity, we can discuss this the revision. **Reviewer 2:** *Typos:* This manuscript underwent several revisions, and we apologize for the numerous typos which remain. We will address all in the subsequent revision. *Motivation:* This paper is best categorized as at the intersection of reinforcement learning theory, control, and online learning, and in the absence of clear RL theory categories, we decided to list control as our subject area. We agree that the setting could be better motivated to a broader control; while connections to robustness are described in prior work, we shall be sure to reiterate them in the introduction to motivate our setting. *Contribution:* We dispute the claim that the contribution is an incremental improvement over [24]. Our main techniques resolve a standing question in online learning with memory, open since 2015: whether "fast" policy regret is obtainable for non-strongly convex unary losses, as in possible in standard online learning (see discussion at 139). Moreover, our bounds demonstrate that the same results attained by a long list of online LQR papers [1,9,10,20,22] are attainable with adversarial noise. *Modeling Assumptions:* The modeling assumptions were stated formally in Section 3. We will include signposts to these conditions in the revision, and in the restructing, these assumptions will be placed at the beginning of the newly proposed control section. *Crucial Identity* We will be sure to expound upon the "crucial indenity in future revions". *Conclusion:* Appendix B.3 contains detailed concluding remarks. We will signpost to this more carefully from the main text. **Reviewer 3:** (1) See discussion at 139 for by GD fails (2) yes, good catch, (3) replace $i$ with $n$, (4) see Alg 3 for definition of how see $\mathcal{C}$ is set (the reference in the Alg 2 should be to Allg 3). This is shown to be efficient in [24], and can be made more practical with a slight relaxation. We shall discuss/clarify this further **Reviewer 4:** Should be $R_{\mathcal{M}} > 0$; For optimal parameter dependences, problem parameters must be known a-prior. However, the performance will degrade gracefully when parameters are misspecified; Unknown horizon can be adressed by the doubling trick (we can discuss this and resilience to parameter inaccuracies further in the appendix); adaptive exploration is challenging due to biases from closed loop control, but can be adressed by alternating rounds of explore and commit; yes, for certain partially observed systems, further assumptions may be required. However, one can show that if the assumptions of [24] hold (as in standard LQG), our algorithm naturally adapts of the induced strong convexity from semi-stochastic noise. Thus, we obtain the best of both worlds. We can sketch this as well.

[Meta-Review · NeurIPS 2020]

The reviewers liked the technical aspect of the paper. Perhaps the main concern is about the presentation style and the target audience, which at the moment is a relatively small crowd that are interested in both online learning and control. The reviewers made a number of useful suggestions for improving this aspect and the rebuttal also outlines a plan that seems reasonable to me. Please do make a serious effort to revise the paper for the final version. Otherwise, nice work!